# Dopaminergic mechanism underlying reward-encoding of punishment omission during reversal learning in *Drosophila*

Li Yan McCurdy [1,2], Preeti Sareen [2], Pasha A. Davoudian [3,4] & Michael N. Nitabach [2,3,5 ✉]

Animals form and update learned associations between otherwise neutral sensory cues and aversive outcomes (i.e., punishment) to predict and avoid danger in changing environments. When a cue later occurs without punishment, this unexpected omission of aversive outcome is encoded as reward via activation of reward-encoding dopaminergic neurons. How such activation occurs remains unknown. Using real-time in vivo functional imaging, optogenetics, behavioral analysis and synaptic reconstruction from electron microscopy data, we identify the neural circuit mechanism through which *Drosophila* reward-encoding dopaminergic neurons are activated when an olfactory cue is unexpectedly no longer paired with electric shock punishment. Reduced activation of punishment-encoding dopaminergic neurons relieves depression of olfactory synaptic inputs to cholinergic neurons. Synaptic excitation by these cholinergic neurons of reward-encoding dopaminergic neurons increases their odor response, thus decreasing aversiveness of the odor. These studies reveal how an excitatory cholinergic relay from punishment- to reward-encoding dopaminergic neurons encodes the absence of punishment as reward, revealing a general circuit motif for updating aversive memories that could be present in mammals.

[1] Interdepartmental Neuroscience Program, Yale University, New Haven, CT, USA. [2] Department of Cellular & Molecular Physiology, Yale University, New Haven, CT, USA. [3] Department of Neuroscience, Yale University, New Haven, CT, USA. [4] MD/PhD Program, Yale School of Medicine, Yale University, New Haven, CT, USA. [5] Department of Genetics, Yale University, New Haven, CT, USA. ✉email: michael.nitabach@yale.edu

Animals use associative learning to build internal models of their environments for guiding adaptive decision making. They also flexibly update learned associations when new or conflicting information arises. Reversal learning paradigms probe cognitive flexibility in the context of changing stimulus-outcome contingencies[1,2]. Reversal learning comprises two components: acquisition and reversal. During acquisition, the "conditioned stimulus+" (CS+) cue is paired with an unconditioned stimulus (e.g., a reward or punishment), while a different CS− cue is presented without, to form a specific association between CS+ and unconditioned stimulus. Later, during reversal, the stimulus-outcome contingencies are reversed: the unconditioned stimulus is omitted during CS+ presentation, and is instead delivered during CS−. Reversal learning requires cognitive flexibility in order for the internal representation of each cue to be first assigned and later updated as contingencies change.

Mammalian studies have identified the importance of dopaminergic neurons (DANs) for reversal learning. For example, manipulation of dopamine signaling in the nucleus accumbens, prefrontal cortex or striatum interferes with reversal learning[2–7]. These studies suggest a key role for DANs in encoding the unexpected omission of aversive outcomes that drives reversal learning. However, how this leads to changes in DAN activity remains unknown, as do the specific distinct roles possibly played by different subsets of DANs.

Here we use Drosophila flies as a model system to address these questions. Drosophila are capable of both forming and reversing cue-outcome associations[8–11], such as odor-shock associations, and have well-characterized neural circuitry and genetic tools for precise manipulation and visualization of neural activity at cellular resolution. The main brain structure underlying learning and memory in the fly is the mushroom body (MB)[10,12–16]. Olfactory cues are represented by the activity of sparse subpopulations of cholinergic Kenyon cell interneurons (KCs), which receive synaptic inputs from second-order olfactory projection neurons and innervate the neuropil of the MB[17,18]. MB output neurons (MBONs), comprising 21 subsets defined by the twenty anatomical MB compartments their dendrites innervate to receive KC synaptic inputs, project to downstream centers of the brain to promote approach or avoidance[19–24]. Approximately 130 modulatory DANs comprising 20 subsets distinguishable genetically and anatomically innervate the twenty compartments of the MB lobes[18,24–27]. MBONs and DANs are named based on the MB lobe compartments they innervate, by reference to the α, β, and γ lobes and contiguous position along each lobe[28,29]. DANs are anatomically classified as PPL1 or PAM DANs, which are viewed generally as encoding negative and positive reinforcement signals, respectively[16,27,30–38]. DANs secrete dopamine into the specific compartments they innervate, where they induce plasticity of KC-MBON synapses, thus modulating MBON odor responses and odor-evoked behavior[19,22,23,26,39–47].

In Drosophila, as in mammals, reward-encoding DANs are thought to underlie the extinction of aversive memories via the formation of a parallel association between the CS+ cue and the omission of punishment[48,49]. However, those studies are of extinction learning, which is identical to reversal learning except with no pairing of CS- with punishment; it is less clear whether a similar process occurs during reversal learning. One group found impairments in reversal learning when a pair of GABAergic neurons called APL that broadly innervates MBs was silenced[9,11]. A few other studies identified factors necessary for reversal learning, such as Rac, the small G-protein involved in cytoskeleton dynamics[50] and Dop1R2, a D1-like receptor[51], but it is not clear where in the MB their effects are necessary or how they are necessary.

While much is known about learning-induced changes in odor-evoked neural activity in MBONs[22,23,26,39–47], few studies have assessed changes in DAN odor responses during acquisition[27,37,52], and none during reversal. It thus remains unknown in any model organism how DAN signals are generated in response to the omission of unconditioned stimuli during reversal learning, or how the omission of punishment is encoded as rewarding.

Here we use a combination of functional imaging, optogenetics, and behavioral approaches in Drosophila to answer these key issues in the biology of reversal learning, specifically the neural mechanisms underlying the formation of the association between CS+ and shock omission. Using an unbiased approach, we establish that PAM-β′2a DANs encode shock omission during reversal as rewarding. This dopaminergic reward signal extinguishes CS+-shock association, and hence reduces CS+ avoidance, by depressing KC synapses onto avoidance-encoding MBON-γ5β′2a. We further demonstrate that approach-encoding MBON-γ2α′1 is an excitatory upstream element of PAM-β′2a which likely causes the changes in PAM-β′2a CS+ odor response during acquisition and reversal. Finally, we show that the absence of shock response by PPL1-γ2α′1 shock-responsive DANs during reversal learning when CS+ is presented without shock relieves synaptic depression of KC-MBON-γ2α′1 synapses to increase CS+ odor activation of MBON-γ2α′1. These studies reveal the underlying cellular and synaptic mechanisms through which DANs of opposite valence participate in an indirect relay from the γ2α′1 compartment to the β′2a compartment to encode omission of a negative outcome as a positive outcome.

## Results

**Reward encoding of shock omission by PAM-β′2a DAN during reversal learning.** To identify specific DAN subsets involved in reversal learning, we developed an experimental preparation for real-time recording of neural activity in genetically targeted neurons during aversive olfactory conditioning. We simultaneously expressed green GCaMP6f $Ca^{2+}$ indicator and red $Ca^{2+}$-insensitive tdTomato in neurons of interest for ratiometric visualization of $Ca^{2+}$ changes reflecting neural activity. Each fly is head-fixed for simultaneous delivery of odors and electric shocks while recording neural activity (Fig. 1a). During acquisition, flies receive alternating 5 s pulses of the conditioned stimulus odor paired with electric shock (CS+) and a control odor (CS−), repeated for five trials (Fig. 1b). During reversal, the contingencies are reversed for two trials: CS+ odor is presented without shock, and CS− odor is presented with shock. We validated this experimental setup in two ways: first, by conditioning single flies in this setup (without cuticle dissection or head-fixing) and testing their odor preference in a two-choice assay after the acquisition. Flies showed a preference for the CS- odor, demonstrating successful acquisition of the aversive memory (Supplementary Fig. 1a). Second, we used the conditioning protocol shown in Fig. 1b to train flies en masse, and tested them in the two-choice assay after acquisition or reversal; flies avoided the CS+ odor after the acquisition, and this avoidance was greatly reduced after two reversal trials (Fig. 1c).

Using this imaging approach, we systematically characterized dynamic changes in CS+ and CS− odor response during acquisition and reversal of each DAN subset targeted individually using a library of intersectional "split-GAL4" driver lines[18,53]. For this screen, we used 4-methyl-cyclohexanol as CS+ odor and 3-octanol as CS− odor. Several DAN subsets responded to the odors (Supplementary Fig. 1b, c). We identified two of seventeen DAN subsets that decrease CS+ odor response relative to CS− odor response during acquisition: PAM-β′2a and PAM-β′2m

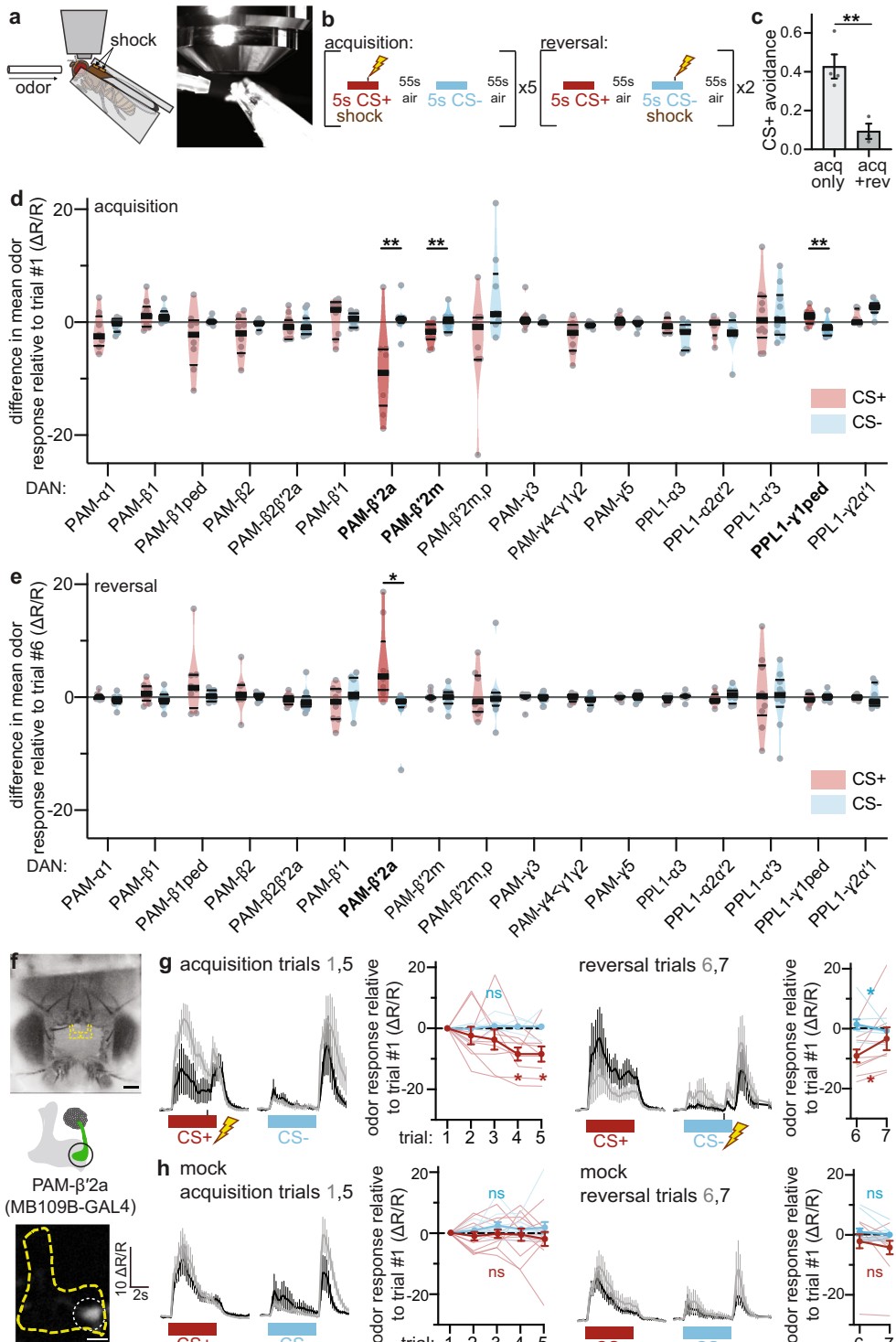

(Fig. 1d). PPL1-γ1ped increases CS+ odor response during acquisition, as previously observed[52]. No changes in CS− odor response occur in any DANs during acquisition. While PAM-β′2a DAN decreases during acquisition, it increases CS+ odor response during reversal (Fig. 1e). No other DANs change their odor responses during reversal; PAM-β′2m and PPL1-γ1ped retain their decreased and increased CS+ odor response, respectively.

PAM-β′2a significantly decreases its CS+ odor response only after three acquisition trials, yet significantly increases its CS+ odor response after a single reversal trial (Fig. 1f, g). This increase

in CS+ odor response during reversal does not completely return to or exceed the initial CS+ odor response, which is consistent with a weakening but not complete abolishing of negative valence associated with the CS+ odor after a single reversal trial. These changes were replicated in a separate group of flies using reciprocal odor identities: 3-octanol (OCT) and 4-methylcyclohexanol (MCH) as the CS+ and CS− odors, respectively (Supplementary Fig. 1d). We also found that PAM-β′2a decreases its CS− odor response after a single reversal trial, suggesting a higher learning rate during reversal than acquisition. Note that although the magnitudes of PAM-β′2a of naïve odor

**Fig. 1 PAM-β′2a DANs change CS+ odor response during aversive memory acquisition and reversal. a** Experimental setup. Flies are presented with odors and electric shocks while neural activity is recorded using GCaMP6f. **b** Training paradigm. During acquisition, CS+ odor (maroon) is paired with electric shock, CS− odor (blue) is not. During reversal, shock is omitted during CS+ and instead paired with CS−. **c** Behavioral validation. Flies conditioned *en masse* using training paradigm in (**b**) avoid CS+ after acquisition; avoidance is reduced in flies conditioned with acquisition and reversal. n = 4 and 3 groups of flies. Statistical comparison is by two-tailed unpaired *t*-test. **d** DAN imaging screen. The difference in mean $Ca^{2+}$ response to CS+ and CS− on last versus first acquisition trial. Dots represent data from single flies. Violin plots show range, median (thick line) and quartiles (thin lines). Change in PAM-β′2a, PAM-β′2 m, and PPL1-γ1ped CS+ response is significantly different relative to CS−. n = 8,9,8,8,11,8,9,11,9,11,8,9,8,8,10,9,7 flies. Statistical comparison by two-tailed paired *t*-test or Wilcoxon matched-pairs signed-rank test. **e** Same as (**d**), except for reversal. Change in PAM-β′2a CS+ response relative to CS- during reversal is significantly different. Sample sizes and statistical comparisons as in (**d**). **f** Top: fly head and dissection window; scale bar:100 μm. MBs outlined in yellow. Middle: Schematic of PAM-β′2a (green), MB in gray. Bottom: Fluorescence image of PAM-β′2a. Scale bar: 20 μm. ΔR/ R scale bar applies to mean fluorescence traces in (**g**, **h**). **g** Neural activity of PAM-β′2a in flies undergoing reversal learning. For each phase, mean fluorescence traces of first (gray) and last (black) acquisition or reversal trial are on the left. On the right, line graphs show CS+ (maroon) and CS− (blue) responses relative to the first acquisition trial for individual flies (thin lines) and group means (thick lines). PAM-β′2a CS+ response decreases during acquisition and increases during reversal. n = 9 flies. Statistical comparison for acquisition and reversal is by repeated-measures two-way ANOVA with Dunnett's post-hoc test and two-tailed paired *t*-test, respectively. **h** Same as (**g**), but mock conditioning. PAM-β′2a odor response does not change. n = 11 flies. All error bars are mean ± SEM. Statistical comparisons as in (**g**), n.s. not significant, *p < 0.05, **p < 0.01.

responses to OCT and MCH are different, using either odor as CS+ led to a decrease in CS+ odor response during acquisition relative to the mock acquisition condition (Supplementary Fig. 1f). Importantly, no changes in odor response occur during mock acquisition or reversal, in which no shocks are delivered, indicating that changes in odor response are dependent on odor-shock pairing (Fig. 1h, Supplementary Fig. 1e). Although we did not see any change in odor response in PAM-γ5 using our particular split-GAL4 line (MB315C-GAL4), we tested an additional PAM-γ5 line which targets different PAM-γ5 neurons to those used in the original screen, as these particular neurons (PAM-γ5(fb)) were recently discovered to be involved in the extinction of aversive memories during extinction learning[49]. We found no changes in odor response during acquisition or reversal, suggesting distinct mechanisms underlying extinction versus reversal learning (Supplementary Fig. 2a–c).

In order to determine the causal relationship between changes in neural activity and behavior, we employed a behavioral reversal learning paradigm (Fig. 2a), in which groups of flies were trained with two acquisition trials and two reversal trials in a standard shock-odor delivery chamber and then tested for their odor preference in a four-quadrant two-odor choice chamber[19,54]. Wild-type flies that undergo acquisition exhibit a robust avoidance of the CS+ odor and greatly prefer the CS− odor, demonstrating that an aversive CS+ memory is formed (Fig. 2b). Flies that undergo one or two reversal trials after acquisition drastically reduce their CS+ avoidance (Fig. 2b). To verify that the decrease in CS+ avoidance observed during reversal is due to a reduction in CS+ avoidance in addition to the acquisition of the new CS− memory, we tested odor preference of CS+ relative to a naïve odor (BA) after acquisition and reversal. Flies reduced CS+ avoidance after reversal compared to after acquisition, demonstrating that an association between CS+ and shock omission has been formed (Fig. 2c). Although the experimental protocol used in the imaging and behavioral experiments differ slightly, our data demonstrate that both setups lead to behavioral change, specifically avoidance of CS+ after the acquisition, and a decrease in avoidance after the reversal (Fig. 1c, Supplementary Fig. 1a, Fig. 2a–c).

Because PAM DANs are traditionally considered to encode reward[20,30,32,35,36,38] PAM-β′2a's increase in CS+ odor response relative to post-acquisition during reversal suggests that it could encode the rewarding value of shock omission. To test this, we first determined the reinforcing properties associated with PAM-β′2a activity. Optogenetic activation of PAM-β′2a using Chrimson red-light-gated ion channel[54] during de novo odor presentation ("CS+") decreases "CS+" odor avoidance, relative to control

"empty-splitGAL4" flies not expressing Chrimson (Fig. 2d). Conversely, silencing PAM-β′2a by activating GtACR1 light-gated inhibitory anion channel[55] with green light during "CS+" increases "CS+" odor avoidance relative to control (Fig. 2e). Thus, despite the mildly negative and positive valence associated with red and green light, respectively, PAM-β′2a activity is rewarding, and the lack of activity is punishing.

We next optogenetically interrogated the functional role of PAM-β′2a during reversal learning. Optogenetic activation of PAM-β′2a during CS+ presentation in reversal trials (when the shock is omitted) decreases CS+ avoidance, relative to control (Fig. 2f). This indicates that PAM-β′2a activity during CS+ presentation in reversal trials is sufficient to enhance CS+ reversal by decreasing CS+ avoidance, consistent with reward encoding of shock omission. Conversely, silencing PAM-β′2a during CS+ presentation in reversal trials increases CS+ avoidance relative to control, reflecting impairment of CS+ reversal via preventing CS+ avoidance decrease during reversal (Fig. 2g). This indicates that PAM-β′2a activity during CS+ presentation in reversal trials is necessary to drive the formation of the association between CS+ and shock omission, again consistent with reward encoding of shock omission. Although manipulating PAM DANs other than PAM-β′2a could also lead to the changes in behavior observed in Fig. 2d–g, our imaging data demonstrate that PAM-β′2a is the only DAN that exhibits a relative change in CS+ odor response during reversal, making PAM-β′2a the likely teaching signal that drives reversal learning. Even though PAM-β′2a CS+ odor responses during reversal do not exceed its initial CS+ odor response, our optogenetic experiments suggest a linear relationship between PAM-β′2a activity levels and reward signaling. As such, this sub-baseline increase in CS+ odor response during reversal is likely an increase in rewarding signaling thus decreases avoidance of the CS+ odor. Given that optogenetic inhibition of PAM-β′2a fails to interfere with aversive shock-odor memory acquisition (Supplementary Fig. 3a), these imaging and functional optogenetic results indicate a role for PAM-β′2a in encoding the relative reward of punishment omission during reversal.

**PAM-β′2a likely decreases CS+ odor avoidance during reversal by depressing KC-to-MBON-γ5β′2a synapses during reversal learning.** How does the increase in PAM-β′2a CS+ odor response during reversal decrease CS+ avoidance? Since PAM-β′2a innervates the same MB compartments as MBON-γ5β′2a and MBON-β2β′2a[18], we focused our attention on these two MBONs, which encode negative and positive valence, respectively[19,22]. MBON-γ5β′2a increases its CS+ odor response during

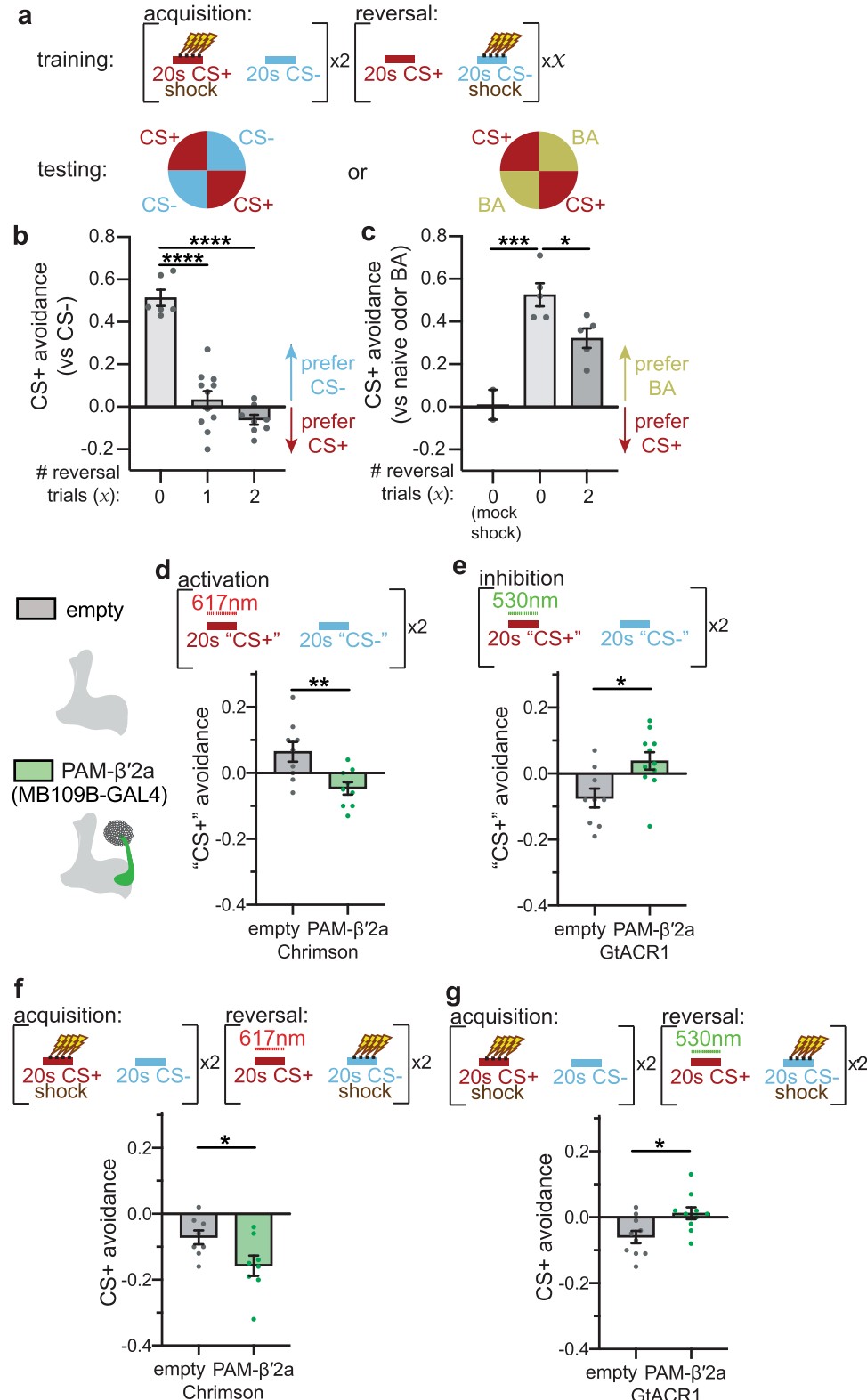

acquisition[22,40,48] and decreases CS+ odor response during reversal (Fig. 3a, b; Supplementary Fig. 4a). This is similar to the decrease in CS+ odor response during extinction learning observed in the previous studies[22,40,48]. These changes do not occur during mock conditioning (Fig. 3c). MBON-β2β′2a increases its response to both CS+ and CS− odors during acquisition and no changes in odor response are observed during reversal (Supplementary Fig. 4b). These imaging results suggest

that MBON-γ5β′2a is the downstream target through which PAM-β′2a reduces CS+ avoidance during reversal, as reduced MBON-γ5β′2a CS+ odor response would encode reduced negative valence of the CS+ odor.

We optogenetically interrogated whether avoidance-encoding MBON-γ5β′2a is involved in reversal. Optogenetic activation of MBON-γ5β′2a during CS+ presentation in reversal trials impairs CS+ reversal, reflected in a relative increase in CS+ avoidance

**Fig. 2 PAM-β'2a DANs encode reward of shock omission during CS+ presentation during reversal learning. a** Training and testing paradigm for behavioral experiments. During acquisition, CS+ is paired with electric shock, followed by CS− without reinforcement. After the acquisition, flies undergo zero, one or two reversal trials, where the shock is omitted during CS+ and presented with CS−. Flies are then placed in the quadrant arena to quantify odor preference. CS+ versus CS− are used in all behavioral figures other than Fig. 2c. **b** Flies avoid CS+ after acquisition; this avoidance is reduced for flies that undergo reversal trials after the acquisition. $n = 6$, 11 and 8 groups of flies. Statistical comparison is by one-way ANOVA with Dunnett's post-hoc test. **c** Flies that undergo reversal after acquisition exhibit reduced CS+ avoidance relative to flies that undergo acquisition only. $n = 2$, 5, and 5 groups of flies. Statistical comparison as in (**b**). **d** Left: Schematic of PAM-β'2a and empty-splitGAL4 driver line. Right: Experimental protocol for assessing valence associated with odor when paired with red-light neural activation ("617 nm") via Chrimson. Flies are exposed to "CS+" paired with red light, followed by "CS−" without light. Flies expressing Chrimson in PAM-β'2a increase preference for "CS+" odor, relative to control flies. $n = 9$ and 8 groups of flies. Statistical comparison is by two-tailed unpaired $t$-test. (**e**) Same as in (**d**), except using green-light ("530 nm") for neural inhibition via GtACR1. Flies expressing GtACR1 in PAM-β'2a decrease preference for "CS +" odor, relative to control flies. $n = 9$ and 11 groups of flies. Statistical comparison as in (**d**). **f** Reversal learning training paradigm. Same protocol as in Fig. 2a, with the addition of red light to activate neurons. Activating PAM-β'2a during CS+ reversal decreases CS+ avoidance relative to genetic controls, indicating enhanced reversal learning. $n = 8$ and 8 groups of flies. Statistical comparison as in (**d**). **g** Same protocol as (**f**), except using green light to silence neurons. Silencing PAM-β'2a during CS+ reversal impairs reversal learning. $n = 10$ and 10 groups of flies. Error bars are mean ± SEM. Statistical comparison as in (**d**), n.s. not significant, *$p < 0.05$, **$p < 0.01$, ***$p < 0.001$, ****$p < 0.0001$.

compared to control (Fig. 3d). This likely occurs by counteracting the decrease in CS+ odor response that normally occurs during reversal (see Fig. 3b). Conversely, optogenetic inhibition of MBON-γ5β'2a enhances CS+ reversal, reflected in a relative decrease in CS+ odor avoidance (Fig. 3e). Although manipulating other MBONs could also lead to the changes in behavior observed in Fig. 3d, e, our imaging data suggest that MBON-γ5β'2a is the likely postsynaptic partner of PAM-β'2a which changes its CS+ odor response to cause a reduction in CS+ avoidance. These results indicate that avoidance-encoding MBON-γ5β'2a plays a causal role in decreasing CS+ avoidance during reversal.

To test whether PAM-β'2a modulates MBON-γ5β'2a odor responses, we employed a combination of optogenetics and Ca$^{2+}$ imaging. De novo odor presentation of 4-methyl-cyclohexanol ("MCH") is paired with a 3 s pulse of red light to activate Chrimson-expressing PAM-β'2a in naïve flies, and odor-evoked responses are measured in GCaMP6m-expressing MBON-γ5β'2a before and after the odor-light pairing (Fig. 3f). PAM-β'2a activation during odor presentation decreases subsequent odor responses of MBON-γ5β'2a, consistent with sustained synaptic depression of KC-MBON-γ5β'2a synapses (Fig. 3g). MBON-γ5β'2a odor responses do not change in independent mock experiments where no light is delivered (Fig. 3h). We also quantified odor responses in MBON-β'2mp, which are unaffected by PAM-β'2a activation during prior odor presentation (Supplementary Fig. 4c). No significant effects of odor-light pairing on subsequent MBON-γ5β'2a odor responses occur in empty-splitGAL4 control flies (Supplementary Fig. 4d). Finally, acute activation of PAM-β'2a does not inhibit MBON-γ5β'2a (Supplementary Fig. 4e), indicating that the opposite changes in neural activity observed in PAM-β'2a and MBON-γ5β'2a are not due to acute inhibition of the latter by the former. These results demonstrate that PAM-β'2a activation during odor presentation depresses KC-MBON-γ5β'2a synapses persistently, thus decreasing MBON-γ5β'2a odor response. This is consistent with the previous observation that pairing neutral odor with broad, non-specific activation of PAMs decreases MBON-γ5β'2a odor response[42].

Taken together, the experiments presented thus far support a cellular and synaptic mechanism for the association between CS+ and shock omission during reversal learning (Fig. 3i): During acquisition, PAM-β'2a decreases CS+ odor response. When an electric shock is omitted during reversal, PAM-β'2a increases CS + odor response relative to post-acquisition, thus depressing KC-MBON-γ5β'2a synapses, decreasing MBON-γ5β'2a CS+ odor response, and thereby decreasing CS+ avoidance.

**MBON-γ2α'1 is upstream of PAM-β'2a and drives CS+ reversal induced by shock omission.** What causes the change in

PAM-β'2a CS+ odor response during acquisition and reversal? Since DAN dendrites receive extensive inputs from MBONs[18], we again took an unbiased screening approach to identify MBONs in addition to MBON-γ5β'2a that change their CS+ odor response during reversal learning.

Several of the MBON subsets responded to each odor (Supplementary Fig. 5a,b). Out of 17 MBON subsets screened, MBON-γ1ped and MBON-γ5β'2a showed a significant difference in changes in CS+ versus CS− odor response during acquisition (Fig. 4a), consistent with prior observations[22,23,26,40]. MBON-α'2, MBON-β'2mp, MBON-γ1ped, MBON-γ2α'1, and MBON-γ5β'2a changed CS+ odor response relative to CS- during reversal (Fig. 4b). MBON-α'2 increases CS+ odor response during acquisition and decreases during reversal (Fig. 4c). However, this finding did not generalize, as no changes in CS+ odor response were observed in the reciprocal odor experiment (Supplementary Fig. 6a). We thus excluded this MBON from further analyses, despite connectivity between MBON-α'2 and PAM-β'2a[18,49,56,57]. MBON-β'2mp increases CS+ odor response during acquisition and decreases during reversal (Fig. 4d). Increased CS+ odor response of MBON-β'2mp during acquisition has been observed previously[22]. MBON-γ1ped decreases CS− odor response during reversal, while its CS+ odor response is unchanged (Fig. 4e), the latter being consistent with prior findings from extinction learning[48]. MBON-γ2α'1 decreases CS+ odor response during acquisition and increases during reversal (Fig. 4f), consistent with prior observations[39]. No changes in CS- odor response occur during acquisition or reversal, except for MBON-γ1ped. No changes in CS+ odor response of MBON-β'2mp, MBON-γ1ped or MBON-γ2α'1 occur during mock reversal learning without shock, and these changes in CS+ odor responses also occur when the odors used as CS+ and CS− are switched (Supplementary Fig. 6b–d).

Our MBON imaging screen thus revealed two potential upstream partners of PAM-β'2a that could encode shock omission during reversal: MBON-γ2α'1 and MBON-β'2mp (Fig. 5a). This led to the hypothesis that manipulating their neural activity should alter reversal learning. Indeed, optogenetically activating Chrimson-expressing MBON-γ2α'1 with maximal red light exposure during CS+ delivery in reversal trials enhances CS+ reversal, reflected in decreased CS+ avoidance relative to empty-splitGAL4 control during subsequent preference testing (Fig. 5b). In contrast, activating MBON-β'2mp with lowest red light exposure impairs CS+ reversal, reflected in increased CS+ avoidance relative to control (Fig. 5b). These effects during reversal parallel previously reported approach- and avoidance-promoting behaviors driven by MBON-γ2α'1 and MBON-β'2mp, respectively[19]. Optogenetic inhibition of GtACR1-expressing

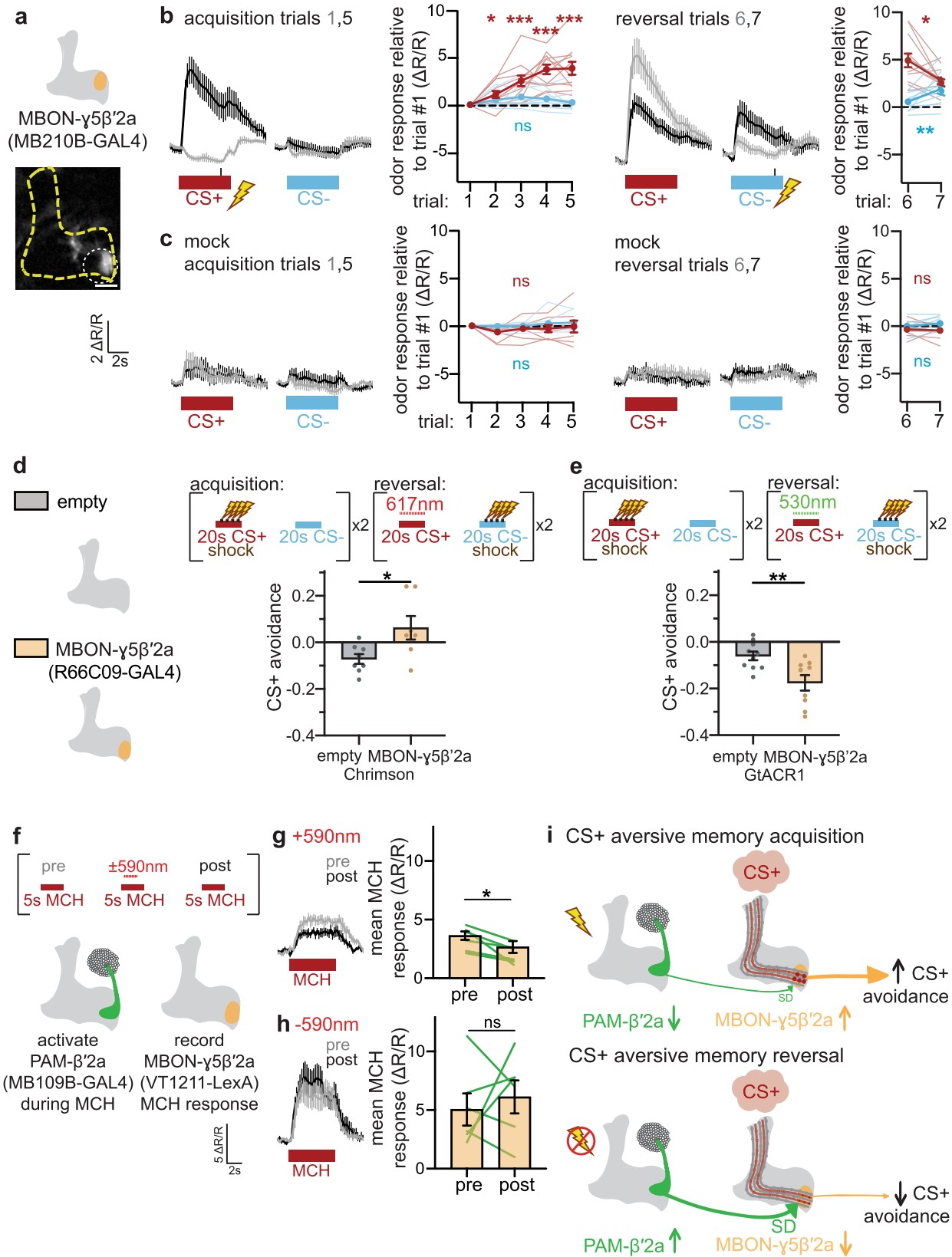

MBON-γ2α′1 or MBON-β′2mp with low green light exposure during CS+ delivery in reversal trials impairs and enhances CS+ reversal, respectively (Fig. 5c). These effects of optogenetic manipulation of MBON-γ2α′1 and MBON-β′2mp are again consistent with roles for both of these neurons in encoding shock omission during CS+ presentation in reversal trials. These

findings are also consistent with how altering MBON activity in the presence of an odor can have reinforcing effects that change corresponding behavior towards the odor[58–60]. Although manipulating other MBONs could also lead to the changes in behavior observed in Fig. 5b, c, our imaging data suggest that these two MBONs are the likely MBONs involved in driving reversal

**Fig. 3 PAM-β′2a DANs likely induce depression of KC-MBON-γ5β′2a synapses to decrease CS+ odor avoidance during reversal. a** Schematic and sample fluorescence image of MBON-γ5β′2a, an output neuron receiving dopaminergic input from PAM-β′2a. Orange region represents the dendrites of MBON-γ5β′2a. Sample fluorescent image is at 20×, scale bar is 20 µm. **b** MBON-γ5β′2a CS+ odor response increases during acquisition and decreases during reversal. $n = 12$ flies. Statistical comparison as in Fig. 1g. **c** No change in MBON-γ5β′2a odor response occurs during mock acquisition or reversal. $n = 8$ flies. Statistical comparison as in Fig. 1g. **d** Schematic of MBON-γ5β′2a and empty-splitGAL4 control used in behavioral experiments. Activating MBON-γ5β′2a during CS+ delivery in reversal impairs reversal learning. $n = 8$ and 7 independent groups of flies per genotype. Statistical comparison as in Fig. 2f. **e** Silencing MBON-γ5β′2a during CS+ delivery in reversal enhances reversal learning, $n = 10$ and 9 independent groups of flies per genotype. Statistical comparison as in Fig. 2f. **f** Optogenetic manipulation of DAN-mediated synaptic plasticity. Odor (MCH) is presented with red light ("590 nm" in red) to activate Chrimson in PAM-β′2a. Odor-evoked MBON-γ5β′2a neural responses to MCH are recorded before ("pre", gray) and after ("post", black) odor-light pairing. **g** Optogenetic activation of PAM-β′2a during MCH odor delivery decreases subsequent MCH odor response. $n = 6$ flies. Statistical comparison is by two-tailed paired $t$-test. **h** Odor responses in MBON-γ5β′2a do not change after mock odor-light pairing. $n = 6$ flies. Error bars are mean ± SEM. Statistical comparison as in (**g**), n.s. not significant, $*p < 0.05$, $**p < 0.01$, $***p < 0.001$. **i** Schematized circuit mechanisms involving PAM-β′2a and MBON-γ5β′2a during aversive memory acquisition and reversal. During acquisition, presentation of CS+ odor (pink cloud) and electric shock (lightning bolt) causes a decrease in PAM-β′2a (green) CS+ odor response; MBON-γ5β′2a (orange) CS + odor response increases via feedforward disinhibition by the γ1ped compartment. During reversal, unexpected shock omission during CS+ odor presentation increases PAM-β′2a CS+ odor response. This induces synaptic depression ("SD") at the KC-MBON-γ5β′2a synapse (green arrow pointing to small maroon circles), thus decreasing MBON-γ5β′2a (orange) CS+ odor response and reducing CS+ avoidance. For clarity, DANs and MBONs are depicted on the left and right hemispheres, respectively.

learning, given their changes in CS+ odor response during reversal.

The complex relationship between the degree of light exposure and behavioral effects in these optogenetic experiments could be due to (1) photobiophysics and membrane biophysics of Chrimson and GtACR1-expressed in MBON-γ2α′1 and MBON-β′2mp, (2) nonlinear network effects in the context of recurrent MB connectivity[18], and/or (3) superimposed non-linear intrinsic effects of red and green light on behavior. We assessed the effect of red light intensity on approach/avoidance to the light itself by flies expressing Chrimson in MBON-γ2α′1, MBON-β′2mp, and non-expressing empty-splitGAL4 control flies (Fig. 5d). Control flies avoid the lowest intensity red light and shift to the attraction at higher intensities. Flies expressing Chrimson in MBON-γ2α′1 shift from relative avoidance to relative attraction compared to control as light intensity increases (Fig. 5e). Conversely, flies expressing Chrimson in MBON-β′2mp shift from relative attraction to relative avoidance as light intensity increases (Fig. 5e). Although the light intensities used in these two experiments (Fig. 5b versus Fig. 5e) cannot be directly compared due to different materials used to diffuse the light, they provide evidence that opposing valences can be induced based on different light intensities.

These intensity-dependent differences in approach/avoidance to light in some respects parallel the intensity-dependent effects of light in the context of reversal learning (Fig. 5b,c). For example, activation of Chrimson-expressing MBON-γ2α′1 with higher red light dose during CS+ delivery in reversal trials decreases CS+ avoidance relative to control during subsequent testing (Fig. 5b), while lower doses of red light do not. Similarly, only the highest red light intensity drives approach of flies expressing Chrimson in MBON-γ2α′1, while lower intensities do not (Fig. 5e). For flies expressing Chrimson in MBON-β′2mp, only the lowest red light dose during CS+ delivery in reversal trials increases CS+ avoidance during subsequent testing, and not a higher dose (Fig. 5b). Similarly, red light preference of flies expressing Chrimson in MBON-β′2mp differs between the lowest and higher light intensities (Fig. 5c).

To distinguish potential roles of MBON-γ2α′1 and MBON-β′2mp in conveying shock omission during CS+ reversal trials to PAM-β′2a, we optogenetically activated each of these neurons individually using Chrimson while recording the neural activity of PAM-β′2a. Since MBON-γ2α′1 and MBON-β′2mp increase and decrease CS+ odor response during reversal, respectively, they would need to excite and inhibit PAM-β′2a, respectively, if involved in increasing PAM-β′2a CS+ odor response during

reversal (Fig. 5f). Indeed, optogenetic activation of MBON-γ2α′1 excites PAM-β′2a (Fig. 5g), supporting MBON-γ2α′1 as transmitting information of shock omission during CS+ presentation in reversal trials to PAM-β′2a. This is consistent with the prior observation that activating MBON-γ2α′1 excites PAMs, although use of a broad driver for GCaMP expression precluded subset identification[61]. Red light fails to elicit PAM-β′2a responses in control flies that do not express Chrimson (Supplementary Fig. 7a). Optogenetic activation of MBON-β′2mp and MBON-γ5β′2a (no MBON-β′2mp specific LexA driver exists) fails to elicit PAM-β′2a responses, and particularly, fails to inhibit PAM-β′2a (Fig. 5h), as would be required for an upstream neuron whose CS+ odor response decreases during reversal to transmit shock omission information to reward-encoding PAM-β′2a.

Taken together, the experiments presented thus far support the following model (Fig. 5i): During acquisition, decreased MBON-γ2α′1 CS+ odor response decreases excitation of PAM-β′2a, reflected in its decreased CS+ odor response. During reversal, shock omission increases MBON-γ2α′1 CS+ odor response, thus increasing PAM-β′2a CS+ odor response relative to post-acquisition levels. This increases persistent synaptic depression of KC-MBON-γ5β′2a synapses, thus persistently decreasing MBON-γ5β′2a CS+ odor response and, consequently, decreasing CS+ odor avoidance during subsequent testing.

**Shock-responsive PPL1-γ2α′1 conveys shock omission to avoidance-encoding MBON-γ5β′2a through MBON-γ2α′1-to-PAM-β′2a relay.** Shock-responsive PPL1-γ2α′1 is the only DAN that projects dopaminergic terminals to the same MB compartment innervated by the dendrites of MBON-γ2α′1[39]. This leads to the hypothesis that shock omission information enters the circuit via PPL1-γ2α′1, and is relayed to avoidance-encoding MBON-γ5β′2a via MBON-γ2α′1 and PAM-β′2a.

To test this hypothesis, we optogenetically interrogated whether reduced PPL1-γ2α′1 activity during shock omission is required for reversal learning. Substitution of direct optogenetic activation of PPL1-γ2α′1 for shock during CS+ presentation in reversal trials prevent CS+ reversal in flies expressing Chrimson in PPL1-γ2α′1, reflected in increased CS+ avoidance compared to control flies not expressing Chrimson (Fig. 6a). Optogenetic shock substitution targeted to other shock-responsive DAN subsets (Supplementary Fig. 8a, b) presynaptic to other MBONs that changed CS+ odor response during reversal (Fig. 4) fails to prevent CS+ reversal (Fig. 6a). Although manipulating PPL1 DANs other than PPL1-γ2α′1 could also lead to the changes in behavior observed in Fig. 6a, the extensive

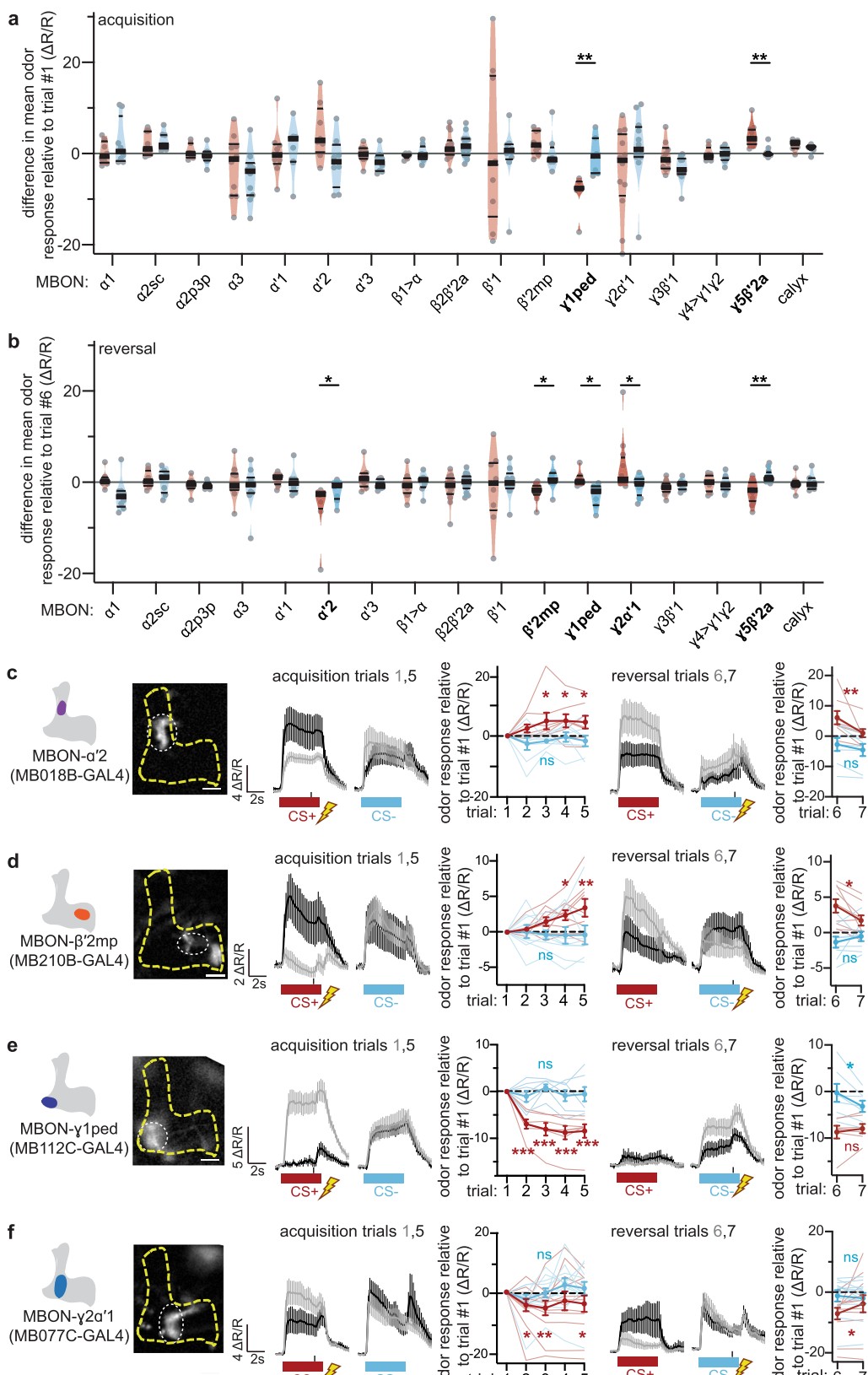

imaging and behavioral data presented thus far strongly suggest that PPL1-γ2α′1 is the most likely DAN responsible for driving reversal learning.

We also combined functional imaging and optogenetics by activating Chrimson-expressing PPL1-γ2α′1 during CS+ delivery in reversal trial #1 to mimic electric shock, and then measuring

CS+ odor response of GCaMP6m-expressing MBON-γ5β′2a during reversal trial #2 (Fig. 6b,c). In mock experiments in which no red light is presented, CS+ odor responses of MBON-γ5β′2a increase after the acquisition and then decrease after reversal (Fig. 6d), as in Fig. 3b. In contrast, when Chrimson-expressing PPL1-γ2α′1 is activated with red light during CS+ odor delivery

**Fig. 4 Five MBONs significantly change odor responses during reversal. a** MBON imaging screen: acquisition trials. The relative difference in mean $Ca^{2+}$ response of indicated MBONs to CS+ and CS− odors from last to first acquisition trials. Change in MBON-γ1ped and MBON-γ5β′2a CS+ odor response relative to CS− odor responses are significantly different. $n = 8, 8, 7, 10, 9, 8, 8, 8, 14, 9, 8, 7, 12, 10, 8, 12, 7$ flies per genotype. Description of violin plots and statistical comparison as in Fig. 1d. **b** MBON imaging screen: reversal trials. Changes in CS+ odor responses relative to CS− odor responses in five MBONs—α′2, β′2mp, γ1ped, γ2α′1, and γ5β′2a—are significantly different. Description of violin plots and statistical comparison as in Fig. 1d. **c** MBON-α′2 CS+ odor response increases during acquisition and decreases during reversal. $n = 8$ flies. Statistical comparison as in Fig. 1g. **d** MBON-β′2mp CS+ odor response increases during acquisition and decreases during reversal. $n = 9$ flies. Statistical comparison as in Fig. 1g. **e** MBON-γ1ped CS+ odor response decreases during acquisition, and its CS− odor response decreases during reversal. $n = 7$ flies. Statistical comparison as in Fig. 1g. Since its CS+ odor response does not change during reversal, it is excluded as a possible upstream modulator of PAM-β′2a. **f** MBON-γ2α′1 CS+ odor response decreases during acquisition and increases during reversal. $n = 12$ flies. Error bars are mean ± SEM. Statistical comparison as in Fig. 1g, n.s. not significant, $*p < 0.05$, $**p < 0.01$, $***p < 0.001$. All sample fluorescent images are at 20×, scale bar is 20 μm.

in reversal trial #1, the decrease in CS+ odor response in reversal trial #2 is eliminated (Fig. 6e; Supplementary Fig. 8c). This indicates that encoding of shock omission by PPL1-γ2α′1 is necessary for MBON-γ5β′2a CS+ odor response to decrease during reversal. Previous studies reveal that, like shock itself, direct optogenetic activation of PPL1-γ2α′1 during odor presentation changes subsequent MBON-γ2α′1 odor responses[39,41]. Our results indicate that PPL1-γ2α′1 encoding of shock is necessary during reversal learning to convey shock omission to avoidance-encoding MBON-γ5β′2a via an MBON-γ2α′1-to-PAM-β′2a indirect dopaminergic relay (Fig. 6f).

To assess synaptic connectivity potentially underlying the circuit we have presented multiple lines of functional evidence for, we took advantage of the recently published *Drosophila* hemibrain electron microscopy volume[57,62]. There are numerous synaptic connections from the single PPL1-γ2α′1 DAN onto each of the two MBON-γ2α′1 neurons (Fig. 6g). Each of the two MBON-γ2α′1 neurons forms an average of five synaptic connections onto the dendrites of each of the ten PAM-β′2a DANs, making the excitatory connection likely a monosynaptic one (Fig. 6h). MBON-γ1ped, MBON-γ3β′1, MBON-γ4γ5, MBON-γ5β′2a, and MBON-β′2mp each form substantially fewer synaptic connections onto PAM-β′2a than MBON-γ2α′1 (Supplementary Fig. 9a), and there are no direct connections from PPL1-γ2α′1 to PAM-β′2a (Supplementary Fig. 9b), providing further evidence that reversal learning occurs via the MBON-γ2α′1-PAM-β′2 relay. Finally, PAM-β′2a neuron forms dozens of synaptic connections onto MBON-γ5β′2a (Fig. 6i). These synaptic connections visualized directly via electron microscopy are consistent with our model, and suggest that the functional connections we have uncovered optogenetically are monosynaptic.

## Discussion

Here we combine functional imaging and optogenetic interrogation to provide extensive evidence for a lateral relay connecting the γ2α′1 MB microcircuit to the γ5β′2a microcircuit to encode punishment omission as a reward during reversal learning. While dopaminergic neurons have been generally implicated in encoding punishment omission as reward in the context of extinction learning, here we uncover not only the specific reward-encoding DANs involved in reversal learning, but also the mechanism through which these DANs are activated by punishment omission.

Our results support the following model for how shock omission during reversal learning reduces CS+ avoidance: During aversive memory acquisition, coincident activation of specific KCs by the CS+ odor and PPL1-γ1ped by electric shock depresses KC-MBON-γ1ped synapses, decreasing MBON-γ1ped CS+ odor response, and disinhibiting MBON-γ5β′2a[23,26,48]. In addition, coincident activation of specific KCs by the CS+ odor and PPL1-γ2α′1 by electric shock depresses KC-MBON-γ2α′

1 synapses, decreasing MBON-γ2α′1 CS+ odor response[39] and decreasing PAM-β′2a CS+ response. During reversal, the omission of electric shock during CS+ odor presentation decreases PPL1-γ2α′1 activation due to the lack of shock response; this coupled with the presence of shock during CS- odor presentation relieves depression of KC-MBON-γ2α′1 synapses that are activated by the CS+ odor. The consequent increase in MBON-γ2α′1 CS+ odor response is relayed by an excitatory synapse to PAM-β′2a, increasing specifically its CS+ odor response. This depresses KC-MBON-γ5β′2a synapses, decreases MBON-γ5β′2a CS+ odor response, and thereby decreases CS+ odor avoidance. Thus, synaptic plasticity in the γ2α′1 microcircuit directly responds to shock omission, and conveys this information to the γ5β′2a microcircuit via PAM-β′2a to extinguish the CS+-shock association. Since these changes in odor response is specific to the CS+ odor, odor specificity is achieved.

While PAM-β′2a has not previously been shown to encode shock omission as a reward, it has been previously implicated as encoding other rewards in other contexts, such as water-seeking[35], sugar-mediated appetitive learning[33], water[63] or ethanol[56] reinforcement, and food-mediated suppression of $CO_2$ avoidance[20]. Interestingly, in the case of ethanol reinforcement, PAM-β′2a likely acts through MBON-β2β′2a, not MBON-γ5β′2a[56], indicating that PAM-β′2a conveys reward information in different contexts via different downstream MBONs. It is important to note that the increase in PAM-β′2a CS+ odor response does not exceed the initial naïve odor response. This is consistent with our model in which PAM-β′2a encodes a relative reward signal, i.e., that shock omission is relatively more rewarding than the presence of shock, but the absence of aversive stimuli is not inherently rewarding in absolute terms. This also explains why a decrease in CS+ odor response occurs during acquisition (Fig. 1g), even though PAM-β′2a activity is not necessary for aversive memory acquisition (Supplementary Fig. 3a). We did not observe any reversal-related changes in CS+ odor response in PAM-β′2m, despite there being a decrease during acquisition. We hypothesize that PAM-β′2m may eventually change its CS+ odor response but only after additional reversal trials. PAM-β′2a would thus detect and signal unexpected reward as it occurs, and PAM-β′2m may eventually be recruited to signal reward as evidence accumulates. Given that PAM-β′2m receives dense innervation from MBON-γ2α′1[18], this may occur via a similar mechanism as the one proposed for PAM-β′2a.

The involvement of avoidance-encoding MBON-γ5β′2a in reversal learning is consistent with its similar role in extinction learning[48]. In both extinction and reversal learning, MBON-γ5β′2a CS+ odor response increases during acquisition and decreases when CS+ is no longer presented with shock. In the case of extinction learning, the authors identified that the increase in MBON-γ5β′2a CS+ odor response during acquisition causes an increase in PAM-γ5 CS+ odor response, which could then induce synaptic depression at KC-MBON-γ5β′2a synapses to reduce

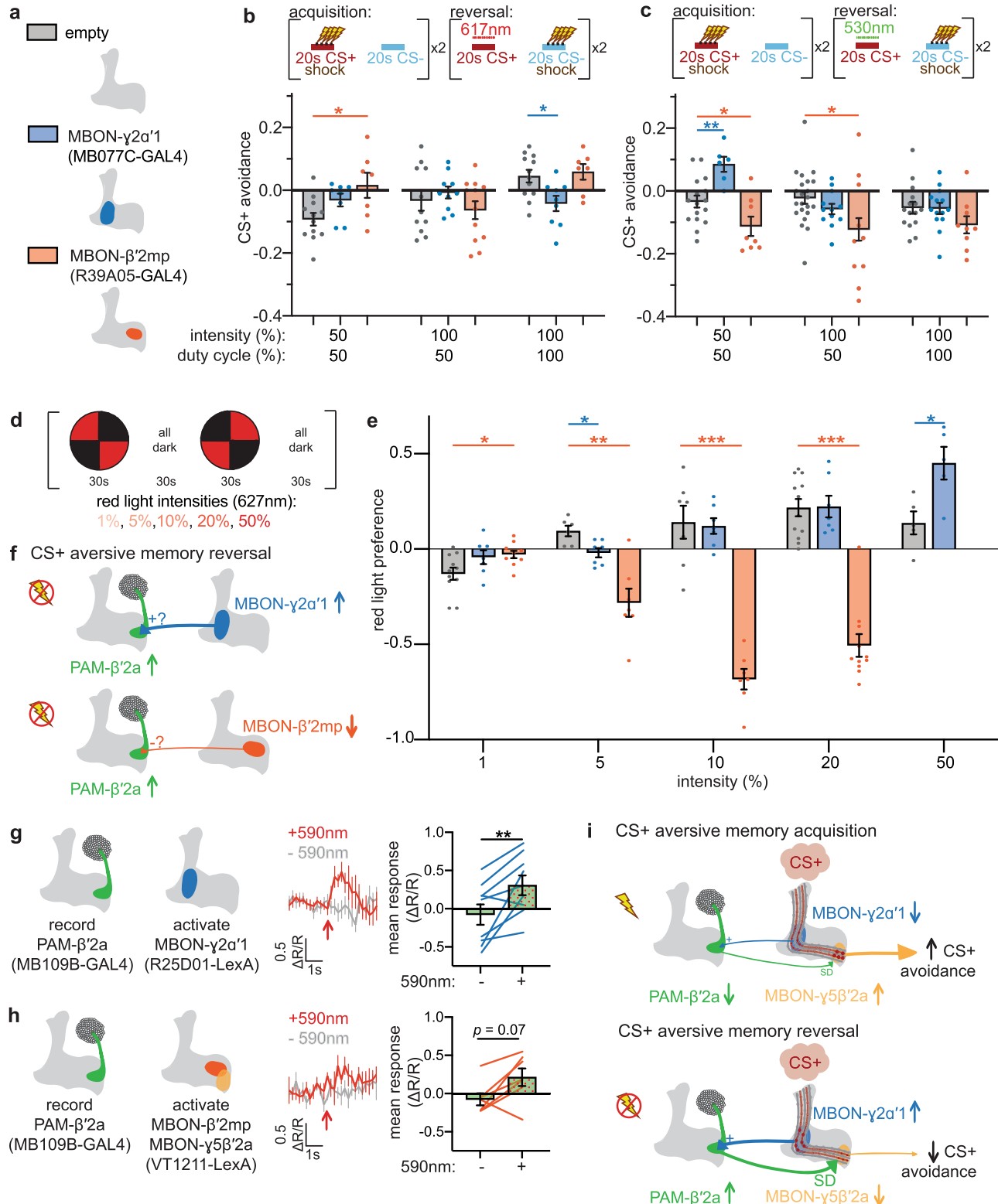

MBON-γ5β′2a CS+ odor response during extinction. More recent work has identified a specific subset of PAM-γ5 neurons, the lower commissure PAM-γ5(fb) neurons, as necessary for memory extinction, and uncovered recurrent connections from MBON-γ5β′2a to those PAM neurons as a likely mechanism underlying aversive memory extinction[49]. It is possible that distinct circuits are involved due to methodological differences between extinction learning performed by others and reversal

learning presented here. For example, flies undergo extinction and imaging 30 and 60 min after acquisition, respectively, whereas in our experiments reversal occurs immediately after acquisition. Interestingly, using our experimental setup, we did not observe any changes in odor response in PAM-γ5(fb) neurons during acquisition or reversal (Supplementary Fig. 2a–c), suggesting different PAM subsets are involved due to different kinds of learning. Likewise, unlike MBON-γ5β′2a which changes

**Fig. 5 MBON-γ2α′1 activation of PAM-β′2a encodes shock omission during reversal learning. a** Schematic of MBON-γ2α′1, MBON-β′2mp, and empty-splitGAL4 driver lines. **b** Activating MBON-γ2α′1 during CS+ reversal with the highest light dose enhances CS+ reversal, while activating MBON-β′2mp with the lowest dose impairs CS+ reversal. $n = 12,8,7$; 11,10,12; 12,9,7 independent groups of flies. Statistical comparison is by one-way ANOVA, with Dunnett's multiple comparisons test against empty-splitGAL4 control. **c** Silencing MBON-γ2α′1 during CS+ reversal at the lowest light dose impairs CS+ reversal, while silencing MBON-β′2mp at the lowest or medium-dose enhances CS+ reversal. $n = 16,6,8$; 22,12,10; 15,14,9 groups of flies. Statistical comparison as in (**b**). **d** Experimental protocol for assessing innate valence. Flies' preference for red light illuminated versus unilluminated quadrants is quantified. (**e**) Flies expressing Chrimson in MBON-γ2α′1 avoid red light quadrants at low light intensity relative to genetic control, and prefer red-light quadrants at the highest light intensity. Flies expressing Chrimson in MBON-β′2mp are relatively attracted to red light quadrants at the lowest light intensity, and avoid red light quadrants at higher intensities. $n = 11,6,10$; 6,7,7; 7,7,7; 12,7,11; 5,5 groups of flies. Statistical comparison is by two-way ANOVA with Dunnett's multiple comparisons test against empty-splitGAL4 control. **f** Proposed connectivity between potential upstream neurons and PAM-β′2a. For MBON-γ2α′1 or MBON-β′2mp to be the upstream neuron of PAM-β′2a, their activation would have to excite (+) or inhibit (−) PAM-β′2a, respectively. **g** Mean neural response in GCaMP6m-expressing PAM-β′2a when Chrimson-expressing MBON-γ2α′1 is activated ("+590 nm", red) and mock activated ("−590 nm", gray); red arrow indicates onset of red light. Bar graphs represent mean neural activity 1 s after red light onset. Activating MBON-γ2α′1 increases PAM-β′2a activity. $n = 9$ flies. Statistical comparison as in Fig. 3g. **h** Activating MBON-β′2mp and MBON-γ5β′2a does not significantly change PAM-β′2a activity. $n = 7$ flies. All error bars are mean ± SEM. Statistical comparison as in Fig. 3g, n.s. not significant, *$p < 0.05$, **$p < 0.01$, ***$p < 0.001$. **i** Proposed role of MBON-γ2α′1 and PAM-β′2a during aversive memory reversal. MBON-γ2α′1 CS+ odor response decreases and increases during memory acquisition and reversal, respectively, causing a corresponding decrease and increase in PAM-β′2a CS+ odor response via an excitatory connection (+).

its CS+ odor response both during extinction and reversal learning, we and others have uncovered some MBONs (e.g., MBON-γ2α′1 and MBON-β′2mp) which change their CS+ odor response during reversal learning and not extinction learning[39,48], providing additional evidence that distinct neural circuits underlie extinction and reversal learning.

Anatomical studies reveal both intra- and inter-compartment connections between DANs and MBONs[18,64], and functional studies suggest the involvement of inter-compartment connections in some forms of learning[65–67]. MBON-γ2α′1 makes connections with a variety of DAN subsets, and is involved in appetitive and aversive memories[39,68] and reconsolidation of appetitive memories[61]. PPL1-γ2α′1 has also been implicated in the acquisition and forgetting of aversive memories[25,51,69–71] and reconsolidation of CS- memory encoding absence of reward[61]. Here we demonstrate a role for MBON-γ2α′1 and PPL1-γ2α′1 in encoding shock omission as a reward during reversal learning, in addition to their known roles in aversive memory formation. Both MBON-γ2α′1 and PAM-β′2a respond to naïve odors, so it is likely that MBON-γ2α′1 contributes to the naïve odor response in PAM-β′2a, resulting in a particular valence associated with the odor. They both also decrease CS+ odor response during acquisition and increase during reversal. Our findings support the hypothesis that this relative increase in CS+ odor response during reversal skews the valence associated with the CS+ odor to decrease CS+ odor avoidance.

Our model does not completely rule out a potential contribution from MBON-β′2mp onto PAM-β′2a. Despite the fact that both MBON-β′2mp and MBON-γ5β′2a secrete glutamate[18,19], they could possibly cause inhibitory and excitatory responses in PAM-β′2a, respectively, leading to the non-significant increase in PAM-β′2a activity observed in Fig. 5h, the former possibly via glutamate-gated chloride channels or indirect connections. However, electron microscopy analyses revealed relatively few connections between MBON-β′2mp and PAM-β′2a, relative to the connections between MBON-γ2α′1 and PAM-β′2a (Supplementary Fig. 9a). Thus the direct excitatory connection between MBON-γ2α′1 and PAM-β′2a is the most parsimonious model.

Taken together, our findings support the model of parallel opposing memories, first demonstrated in *Drosophila* extinction learning[48], as the basis of reversal learning. In this model, the γ1ped microcircuit encodes the originally acquired aversive memory[23,26,48]. Indeed, we found that MBON-γ1ped does not increase its CS+ odor response during reversal (Fig. 4e), demonstrating the persistence of the original memory stored in

the γ1ped microcircuit. MBON-γ5β′2a on the other hand is responsible for forming the association between CS+ and shock omission via PAM-γ5(fb) in extinction learning, and is the last node in the circuit for forming the CS+-no shock memory via the γ2α′1-β′2 relay network we uncover here in reversal learning. MBON-γ5β′2a is poised to integrate the initial aversive memory with the new no-shock memory via the direct feed-forward inhibitory connection from MBON-γ1ped[48]. We propose that the γ1ped and γ2α′1-β′2 microcircuits are analogous to the roles of the amygdala and ventromedial prefrontal cortex (vmPFC) networks in fear reversal learning in mammals. The amygdala persistently encodes the originally acquired fear memory[72–74], while the vmPFC encodes omission of the aversive outcome during reversal as reward[74,75]. Interestingly, inhibiting vmPFC-projecting mesocortical ventral tegmental area (VTA) DANs during shock omission impairs extinction[76], analogous to our finding that inhibiting PAM-β′2a CS+ odor response during reversal impairs extinction of the aversive memory. PAM-β′2a odor responses can thus be thought of as a prediction error signal, which encodes the value associated with each cue by iteratively updating the value on a trial-by-trial basis[77,78]. Since a subset of VTA DANs have been found to respond to aversive stimuli such as electric shock[79], it would be interesting to see if these aversive-encoding DANs relay information about the absence or presence of punishment- to reward-encoding DANs which signal unexpected omission of punishment to the vmPFC. In sum, we have revealed how punishment-encoding DANs and reward-encoding DANs collaborate in an indirect dopaminergic relay circuit in the fly to encode shock omission as a reward, and thereby underlie formation of a CS+ reward memory during reversal learning. This circuit topology for encoding omission of punishment as a reward could be conserved across animal species, and suggests testable hypotheses for exploration in mammals.

## Methods

**Fly husbandry.** Fly stocks were cultured on standard cornmeal medium at 25 °C with a 12-h-light-dark cycle. Flies used for functional connectivity experiments were maintained on molasses medium. Flies used for optogenetic behavioral or functional connectivity experiments were kept in the dark using aluminum foil, and raised on a standard cornmeal medium supplemented with 0.4 mM all-*trans*-retinal (Cayman Chemical) at least two days before experiments. All experiments were conducted in the period of 1 h after lights-on and 1 h before lights-off (Zeitgeber time 1–11 h). For all behavioral experiments, mixed populations of male and female flies aged 3–10-days old were used. For all imaging and functional connectivity experiments, 5–8-day-old females were used. Please see Supplementary Information for a list of all genotypes used in all experiments.

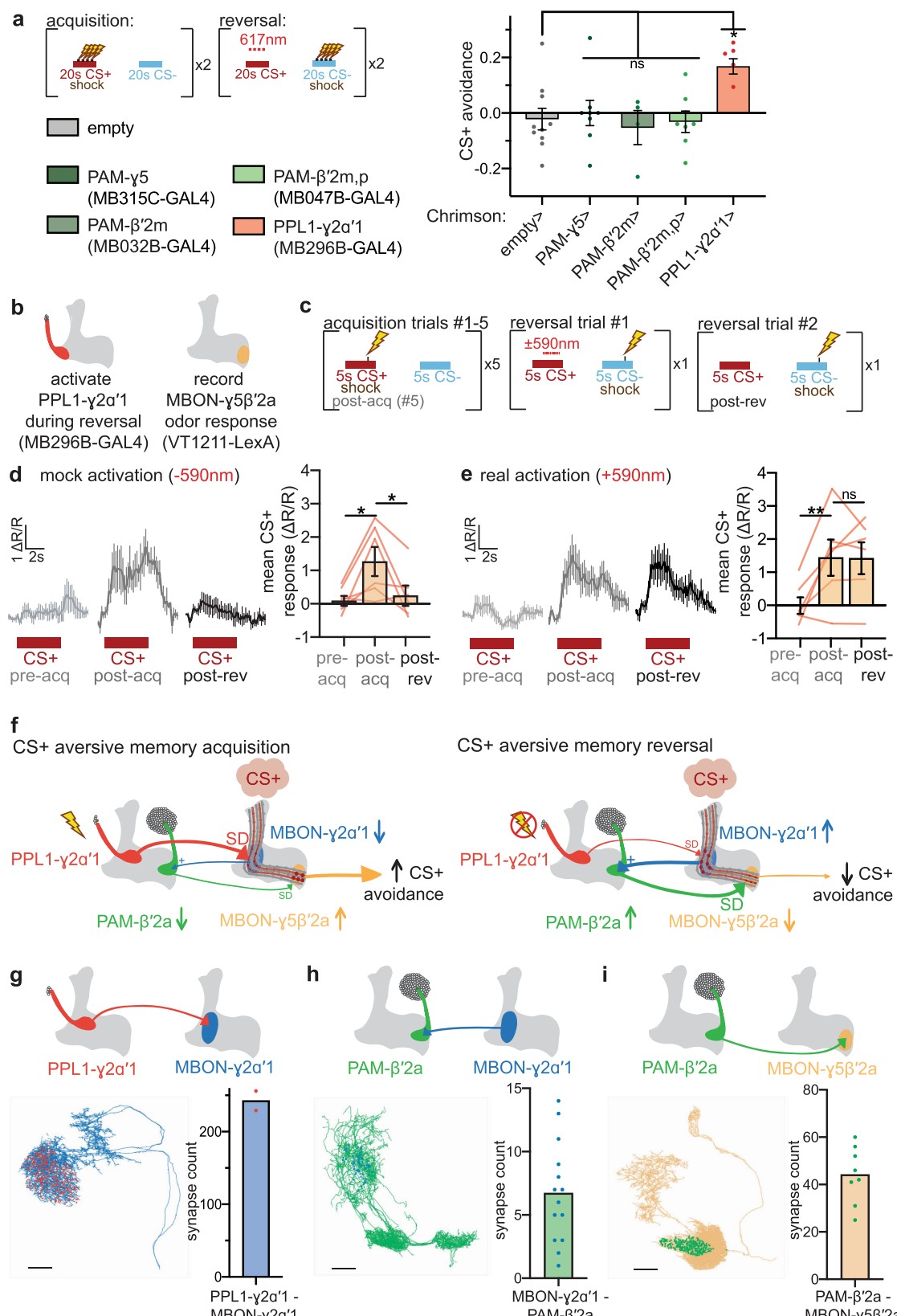

**Calcium imaging during classical conditioning**. Each 5–8-day-old female fly was aspirated without anesthesia into a custom-made chamber comprising a plastic slide, a 200 μL pipette tip, and some copper foil tape (Tapes Master). The pipette tip acted as a chamber for the fly, and the copper tape was connected via conductive epoxy (Atom Adhesives) to wires which connect to the power supply to deliver electric shocks. In this setup, the fly's head was accessible for dissection, its body was contained within the pipette tip, and the back of its thorax was in contact with the copper tape. A thin layer of conductive gel (Parker Labs) was applied to the copper tape prior to inserting the fly to improve conductivity. The fly was then head-fixed using two-component epoxy glue (Devcon); its proboscis was also glued, but its body and legs were free to move. After the glue dried (about 5 min), the cuticle above the head was dissected and air sacs were removed using a 30-gauge syringe needle (Covidien) and a pair of forceps (Fine Science Tools). The head capsule was then sealed with silicone adhesive Kwik-Sil (World

**Fig. 6 Shock-responsive PPL1-γ2α′1 dopaminergic neurons convey shock omission during reversal. a** Experimental paradigm for "shock-substitution" experiment. When Chrimson is expressed in PPL1-γ2α′1, but not other shock-responsive DANs, red-light pulses delivered during CS+ reversal trials mimicking electric shock impair reversal. $n = 10, 8, 4, 7$, and 5 independent groups of flies per genotype. Statistical comparison is by one-way ANOVA with Dunnett's post-hoc test. **b** Flies express red-light-activated Chrimson in PPL1-γ2α′1 neurons, and GCaMP6m in MBON-γ5β′2a. **c** Experimental paradigm for determining the effect of PPL1-γ2α′1 activation on reversal-induced decrease in MBON-γ5β′2a CS+ odor response. **d** MBON-γ5β′2a increases and decreases CS+ odor response during acquisition and reversal, respectively, when red-light activation of PPL1-γ2α′1 does not occur. $n = 6$ flies. Statistical comparison is by one-way ANOVA with Dunnett's post-hoc test against post-acquisition. **e** Optogenetic activation of PPL1-γ2α′1 DAN during CS+ presentation in first reversal trial prevents decrease in MBON-γ5β′2a CS+ odor response. $n = 6$ flies. Error bars are mean ± SEM. Statistical comparison as in (**d**), n.s. not significant, *$p < 0.05$, **$p < 0.01$. **f** Schematic of mechanism of how CS+ odor avoidance is reduced during reversal learning. See "Discussion" for full description. **g** Synapses between PPL1-γ2α′1 and MBON-γ2α′1. Left, blue neuron is EM-traced MBON-γ2α′1; red dots represent the location of individual synapses from PPL1-γ2α′1. Right: quantification of number of synapses from PPL1-γ2α′1 onto each MBON-γ2α′1 neuron. Bar indicates mean synapse count. Several synapses are formed between PPL1-γ2α′1 and the dendritic regions of both MBON-γ2α′1 neurons. $n = 2$ MBON-γ2α′1 neurons. Scale bar is 20 μm. **h** Synapses between MBON-γ2α′1 and PAM-β′2a. Green neuron is EM-traced PAM-β′2a neurons; blue dots represent synapses from MBON-γ2α′1. Bar indicates mean synapse count. Several synapses exist between the two MBON-γ2α′1 neurons and the dendritic region of ten PAM-β′2a neurons. $n = 20$, for the connections between two MBON-γ2α′1 neurons and ten PAM-β′2a neurons. Scale bar is 20 μm. **i** Synapses between PAM-β′2a and MBON-γ5β′2a. Orange neuron is EM-traced MBON-γ5β′2a; green dots represent synapses from PAM-β′2a. Bar indicates mean synapse count. Several synapses exist between each of the ten PAM-β′2a neurons and dendritic regions of MBON-γ5β′2a. $n = 10$ PAM-β′2a neurons synapsing onto one MBON-γ5β′2a neuron. Scale bar is 20 μm.

Precision Instruments). The fly was then placed in a humidified chamber to recover for 15 min.

The fly was then positioned under the microscope. The odor tube was placed about 5 mm away from the fly. The odorants used were 0.1% 3-octanol and 0.1% 4-methylcyclohexanol (Sigma Aldrich), controlled via three-way solenoids (Lee Company). The flow rate was kept constant at 1000 mL/min for each odor stream using mass flow controllers (Alicat Scientific). Before reaching the fly, the odor delivery line was split into two, one directed to the fly and the other directed to the photoionization detector (mini-PID, Aurora Scientific) to monitor odor delivery for each trial, for a final flow rate of 500 mL/min at each end. The fly was allowed to acclimate to the airflow for at least one minute before beginning the experiment.

Each trial lasted 1 min; during the first 20 s of each trial, blue and green lights were presented simultaneously to visualize fluorescence. The imaging experiment began with four such trials, for the animal to acclimate to the lights. For the actual experiment, alternating trials of 5 s of each odor was presented 5 s after the light onset. 4-methylcyclohexanol and 3-octanol were used as CS+ and CS− odors for the imaging screen, respectively. On trials where the electric shock was delivered, the shock was presented 4 s after odor onset for 100 ms at 120 V.

For experiments in which the single fly was later tested for odor preference (Supplementary Fig. 1a), no head-fixing or cuticle dissection was performed. Flies underwent acquisition conditioning protocol as above, were removed from the pipette tip, and allowed to recover for 5 min before being placed in the quadrant arena (described below).

**Calcium imaging specifications and analysis**. Imaging was performed on a Zeiss Axio Examiner upright microscope using a 20× air objective (Zeiss). Using a Colibri LED system (Zeiss), GCaMP was excited at 470 nm (blue) and tdTomato was excited at 555 nm (green). An optosplitter (Photometrics DV-2) was used to acquire fluorescence from both wavelengths to allow for ratiometric analysis for fluorescence, to control for movement artefacts. Images were acquired at 5 fps.

Data acquired from calcium imaging were analyzed using a combination of Zen (Zeiss) and custom MATLAB code (Mathworks). Since tdTomato was expressed in the same neurons that GCaMP was expressed in, regions of interest (ROIs) were manually drawn based on the tdTomato image and applied to the corresponding GCaMP frame. Mean pixel intensity within each ROI was extracted as a raw fluorescence value representing signal in each frame. The GCaMP signal was divided by the tdTomato signal to form the ratiometric fluorescence value, and then processed with a double-exponential fitting to compensate for any photobleaching. The resultant trace was converted into $\Delta R/R_0$ by using a baseline intensity ($R_0$), defined as mean intensity during the first 5 s of recording (prior to odor onset).

Mean odor response was defined as the mean $\Delta R/R_0$ during the first four seconds after odor onset ($t = 0$–4 s); this was so that the response would not be confounded by any potential electric shock response (which occurred 4 s after odor onset). Mean shock response was defined as the mean $\Delta R/R_0$ during 800 ms upon electric shock onset ($t = 4$–4.8 s). This again was to avoid confounding this response with any potential odor offset response that would occur at $t = 5$ s. In screen data (e.g., Fig. 1d), "difference in mean odor response" for acquisition is calculated as the subtraction of mean odor response on the first acquisition trial from mean odor response on the fifth acquisition trial. "Difference in mean odor response" for reversal (e.g., Fig. 1e) is calculated as the subtraction of mean odor response on the first reversal trial from the mean odor response on the second reversal trial. In both cases, a positive value indicates an increase in odor response relative to the first acquisition/reversal trial. In data for individual neurons (e.g., Fig. 1g), "odor response relative to trial #1" is the difference in mean odor response relative to the mean odor response on the first acquisition trial. Positive values indicate an increase in odor response.

**Behavioral classical conditioning assays**. Each group of 30–70 flies was aspirated into a training chamber (Con-Elektronik), comprising an electric shock system and an odor delivery system. The odorants used were 0.07% 3-octanol and 0.1% 4-methylcyclohexanol. The odors were delivered at a flow rate of 500 mL/min. Flies were allowed to acclimate in the training chamber for about a minute, and then training began. Odorants were presented for 20 s, with 20 s inter-trial interval in which clean air was presented. Depending on the protocol, electric shocks may have been delivered during odor presentation. These shocks were 120 V for 1.5 s, with an inter-trial interval of 3.5 s; four of them were delivered during each 20 s odor pulse. Within a minute after training, flies were aspirated into the testing arena[54] and allowed to walk around in the flat cylindrical arena with CS+ and CS− odors delivered to alternating quadrants at a flow rate of 100 mL/min each. The exception is in third odor experiments (Fig. 2c), in which 0.85% benzaldehyde was delivered to the quadrants instead of the CS− odor. Videography was performed under IR light using a camera (Flea3 USB 3.0 camera; Point Gray). CS+ avoidance was quantified as the number of flies in CS− quadrants minus the number of flies in CS+ quadrants, divided by a total number of flies, after being allowed to explore for 2 min; an avoidance index of 1 indicates that 100% of the flies chose CS− quadrants, and an avoidance index of 0 indicates that flies chose the two odors equally. Each odorant was alternately used as CS+ and CS− odors in separate groups of flies trained and tested contemporaneously and averaged to generate a reciprocally balanced preference score.

For the experiment mimicking the imaging protocol (Fig. 1c), the protocol depicted (Fig. 1b) is performed as presented, with the exception of using 1.5 s electric shocks instead of 100 ms. For experiments where single flies are tested for odor preference in the quadrant arena after conditioning under the microscope (Supplementary Fig. 1a), each fly's position in the quadrant arena is recorded for two minutes, and the percentage of time spent in CS+ versus CS− quadrants is quantified. For shock-substitution" experiments (Fig. 6a), the training protocol is identical to those used throughout the manuscript, with the exception of the pattern of red light activation. It is done to mimic electric shock stimuli during acquisition phase, i.e., four 1.5 s pulses with 3.5 s in between each pulse.

**Optogenetics in conjunction with behavioral classical conditioning assays**. The identical setup as described above was used, with the addition of a custom-made RGB LED board positioned ~3 cm below the training chamber. Because the electric shock training chamber was translucent, light was able to pass through and illuminate the chamber. The high-power LEDs (Luxeon) were red (617 nm, 166 lm) and green (530 nm, 125 lm), and were controlled together with the odor and electric shock stimuli via a microcontroller (Arduino Mega). Unless otherwise stated, red and green light was set to 50 and 100% max intensity, respectively; both were pulsed at 50 Hz with a duty cycle of 50%, i.e., 10 ms light on, 10 ms light off. A duty cycle of 100% indicates that the lights were constitutively on.

**Innate quadrant preference assay**. The choice assay was performed in the quadrant testing arena described above, but without any odor input. Approximately 30 flies were aspirated into the arena in the dark for one minute before the experiment commenced. For 30 s, two diagonal quadrants were illuminated with 627 nm red LEDs to activate Chrimson-containing neurons, after which, two quadrants were illuminated for 30 s. This was then repeated with the other two quadrants. Red light preference index was calculated as the number of flies in red light quadrants minus flies in dark quadrants, divided by the total number of flies; the values are then averaged to control for place preference. A preference index of 1 indicates that 100% of flies prefer red light quadrants, and a preference index of 0 indicates that flies chose the quadrants equally. This was then repeated with the same group of flies over a range of light intensities: 1, 5, 10, 20, and 50%. Note that these red light intensities

cannot be directly compared with the light intensities used in Fig. 5b. For a minority of data points, flies were only tested at one light intensity. Since there were no differences in mean preference between these two datasets, they were combined. Data for MBON-β′2 mp at the highest light intensity is not shown, because these flies exhibit locomotor deficits at this light intensity, so their red light preference would not accurately reflect valence associated with MBON activation.

**Functional connectivity experiments**. These experiments were performed similarly to the calcium imaging experiments described in the previous section, with the following modifications: After being placed under the microscope and focusing the field of view, each fly was given 5 min in darkness to recover. For direct stimulation experiments (Fig. 5g, h), activity was recorded for 20 s without any red light, using dim blue light (470 nm at 20% intensity). After 5 min, we recorded 10 s of baseline activity before stimulating with red light (590 nm at 50% intensity) for 500 ms, recording for a total of 20 s. For experiments looking at odor-evoked responses (Fig. 3f–h), a total of three odor pulses (of 0.01% 4-methylcyclohexanol) were delivered at 2 min intervals. Each 5 s odor pulse was preceded by 10 s of baseline recording (using dim blue light at 20% intensity), and recorded for 5 s after odor offset (20 s total per trial). During the second odor presentation, a red light pulse (590 nm) was delivered for 3 s at 8 Hz, beginning 0.5 s after odor onset. (Blue light was not presented during this trial.) For experiments looking at odor-evoked responses during stimulation (Supplementary Fig. 4e), two-odor pulses of MCH were presented at two-minute intervals. During the second odor pulse, red light (590 nm at 50% intensity) was delivered for 500 ms, 2 s after odor onset. Mean odor responses were calculated as mean $\Delta R/R$ for the 2 s duration after light offset. For experiments looking at opto-genetic stimulation after aversive memory acquisition (Fig. 6b–e), the protocol was identical to that used in the calcium imaging screen, except only recording (i.e., shining blue light) during the pre-acquisition CS+ trial, the fifth CS+ acquisition trial, and the second CS+ reversal trial. Optogenetic stimulation on the first reversal trial was done using a 590 nm red light at 50% intensity, for 3 s at 8 Hz beginning 1.5 s after CS+ onset. Mean odor responses were calculated as mean $\Delta R/R$ for the 5 s duration that the odor was presented.

**Anatomical synapse analysis**. Publicly available, open-source electron microscopy data were utilized for all analyses. Details regarding the generation and initial analysis of the primary data can be found in the original reports[57,62]. All further analyses described here were performed in R and Python. Synapse counts, synapse ROIs, synaptic locations, and neuronal morphology skeletons were assessed via the Python API for neuPrint[80]. Version 1.1 of hemibrain connectome data was utilized for all analyses. Neuronal skeletons were accessed and corresponding 3D images were visualized via the natverse R packages[81].

The primary dataset utilized in this analysis predominantly imaged one-half of a female *Drosophila* brain, as such we limited our analysis to this side of the data and included only neurons that were marked as fully traced to avoid biasing in our sampling. We first manually identified neurons in the neuPrint database that corresponded to neurons of interest. After identification, the number and ROI of synaptic contacts between neurons of interest were quantified using existing neuPrint API functions. Specific three-dimensional Euclidean coordinates of postsynaptic contacts of interest were generated and visualized on particular postsynaptic neurons in two-dimensions using existing neuPrint functions. Neuronal skeleton morphologies of interest were registered to the JRC2018F template brain and visualized in three-dimensions using the natverse.

**Statistics and reproducibility**. Experimental flies and genetic controls were tested at the same condition, and data were collected on at least two different days. Statistical analyses were performed in GraphPad Prism 8. All data were tested for normality using the D'Agostino and Pearson omnibus test. Each statistical test used in each figure is reported in the legends. Generally, normally distributed data were analyzed with a two-tailed $t$-test, one-way ANOVA or two-way ANOVA followed by Dunnett's post-hoc test where appropriate. For non-Gaussian distributed data, a Wilcoxon signed-rank test, two-tailed Mann-Whitney test, or a Kruskal-Wallis test was performed followed by Dunn's multiple comparisons test where appropriate. Since the variance in PAM-β′2a data for CS+ and CS− differed substantially, a Greenhouse-Geisser correction is applied before performing repeated-measures two-way ANOVA (Fig. 1g, h; Supplementary Fig. 1d, e). Most figures were generated using GraphPad Prism 8. All data in bar graphs are expressed as mean ± SEMs.

**Reporting summary**. Further information on research design is available in the Nature Research Reporting Summary linked to this article.

## Data availability
All data and statistics used in this manuscript are available in the Supplementary Information file and Source Data file. In addition, they are available from the corresponding author upon request. Source data are provided with this paper.

## Code availability
Code used to analyze calcium imaging data is available in the Supplementary Software file. In addition, it is available from the corresponding author upon request.

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

## Acknowledgements

We thank Damon A. Clark and Scott Waddell for reagents. We thank members of the Nitabach lab for technical support, advice, and comments. We thank Zunaira Arshad, Alexandru Buhimschi, Juliana Rioz Chen, Justin J. Choi, Michelle Benavidez Frausto, Madeleine Gharizan, Daniel A. Grubman, Katelyn Kang, Dawit Mengasha, Alice Oh, Natalie Dorzeide Orner, and Krishna Vali for conducting pilot behavioral experiments. L. Y.M. was supported by the National Science Foundation (NSF) Graduate Research Fellowship Program (GRFP), Yale University Cellular and Molecular Biology training grant from the National Institutes of Health (NIH) (T32GM007223), Gruber Science Fellowship, and National University of Singapore Overseas Graduate Scholarship. P.A.D. is supported by the Medical Scientist Training Grant, NIH (T32GM007205). Work in the laboratory of M.N.N. was supported by National Institute of Neurological Disease and Stroke (NINDS), NIH (R01NS091070).

## Author contributions

Designed study and experiments: L.Y.M. and M.N.N. Designed and created unpublished reagents: P.S. Performed experiments: L.Y.M. Data analysis and statistics: L.Y.M., P.D., and M.N.N. Wrote the paper: L.Y.M. and M.N.N.

## Competing interests

The authors declare no competing interests.
