## [Peer Review File · Nature Communications]

Reviewer #1 (Remarks to the Author):

McCurdy et al investigate the neural circuitry underlying aversive learning and reversal learning. By combining imaging, behavioral assays and optogenetic approaches the authors provide novel insights into how changes in contingencies of learned cues relate to learning dependent plasticity, to the activity of the dopaminergic system and to the valence coding output system within the mushroom body of fruit flies. The findings are of general importance and will raise interest beyond the fly learning and memory field. As positive as I am about the data presented, as much I disagree with many of the interpretation of the results and consider some of them for overstated. With all the respect for the work, the proposed model is not backed up by the data and therefore likely to be wrong. In fact, below I will challenge most of their claims point by point. I would like to emphasize that I do not question the quality of the data but its interpretation given the current state of experiments. Most importantly, the authors clearly under-appreciate the importance of their findings in relation to reversal learning. They neglect that, to my knowledge, they provide the first comprehensive data set on reversal learning related plasticity in the fly. Rather than solely focusing on the extinction-like omission of electric shock situation during reversal, the manuscript would massively improve if the authors would acknowledge that extinction and reversal learning are potentially different processes. Further, the authors should revisit their literature citations because some of them are inadequate. In am aware that, due to limited access to experimental set-ups in thess special times, it is hard to add additional data but I will have to point out where crucial controls are missing. I support the communication of the data but not in the current conceptual setting nor with the proposed model. The authors must substantially re-work the manuscript to incorporate the outlined criticisms. I will address the issues point by point, hoping that they will be helpful, and I am looking forward to reading the authors response.

First (1-4) I will go through the major claims one by one as they are stated in the manuscript, then address other major concerns followed by minor issues.

1) "We establish that PAM- β '2a DANs encode shock withdrawal during reversal as rewarding."

The data presented in Fig 1 C-Fii shows that the PAM β '2a responses to the CS+ decreases during aversive training and increases again during reversal learning. From the traces presented it becomes clear that the PAM β '2a responses during reversal return to the baseline of the first naïve-odor response (Fig 1 Fi and Fii). Thus, at no point in the entire protocol are the responses in these dopamine neurons elevated above the naïve odor response. Thus, the claim that PAM β '2a neurons encode the shock withdrawal and provides a rewarding teaching signal is not valid. In general, in all the imaging data, for all neurons investigated, the authors should visualize the effects of reversal trials 6 and 7 by normalizing and plotting it in the same panel then acq. trials 1-5 to avoid misinterpretations.

Further, the authors test if activating or inhibiting PAM β '2a neurons during the CS+ during the two reversal trials changes behavior. The observed results are consistent with their claim. However, there is an alternative explanation equally valid given the presented controls: it might be that PAM β '2a activity, independent of reversal learning, provides a reward-like teaching signal and that the absence of the activity assigns a negative value to the odor similar to other PAM neurons (PAM γ 3, Yamagata et al 2016). To control for this alternative explanation, one has to repeat the same reversal protocol, at least in terms of number of trials, to test if pairing an odor with the activity of the tested neurons changes behavior. The experiments presented by the authors don't do that, e.g. the use one instead of two trials in the activation experiment or block throughout single aversive training trial in the inhibition experiment which does not control if the inhibition of the neurons provides a teaching signal. Therefore, the presented controls are invalid to exclude that alteration of

PAM $\beta'2a$ activity during two trials generates a teaching signal. Without the correct controls the experimental results from the reversal learning are inconclusive.

Thus, since there is no increase in PAM $\beta'2a$ activity coinciding with the CS+ during shock withdrawal in reversal, and the behavioral experiments being inconclusive, the claim is not supported.

2) “This dopaminergic reward signal extinguishes CS+-shock association, and hence reduces CS+ avoidance, by depressing KC synapses onto avoidance-encoding MBON- $\gamma5\beta'2a$.”

Related to 1) it is not clear if the PAM $\beta'2a$ provides a reward like teaching signal during reversal. The authors show that activating PAM $\beta'2a$ artificially in coincidence with an odor leads to a depression of the MBON- $\gamma5\beta'2a$ drive for that odor. However, as outlined in 1) there is no increase in PAM $\beta'2a$ during reversal learning. Instead, the data shows that whenever PAM $\beta'2a$ is decreased MBON- $\gamma5\beta'2a$ responses are enhanced suggesting a direct modulation of KC-MBON- $\gamma5\beta'2a$ synapse efficiency by the activity of PAM $\beta'2a$ without plasticity. In fact, it has been frequently reported that PAM $\beta'2a$ activity can modulate odor driven behavior in a context or state dependent manner (Lewis et al 2015 Senapati et al 2019, Scalpen et al 2020). In fact, the state-dependent control involves the axis from $\gamma2\alpha'1$ MBON and PPL1 neuron similar to the finding of the authors and supports the idea of a cross-talk between the aversive/state controlling dopamine pathways and the MBON- $\gamma5\beta'2a$. The authors should consider these alternative interpretations of their findings. A valuable additional data set would be, to repeat the experiment presented in Figure 2G but look at the odor response during PAM $\beta'2a$ activation to test if its activity can directly inhibit the odor drive of the MBON.

3) “We further demonstrate that approach-encoding MBON- $\gamma2\alpha'1$ is an upstream element that activates PAM- $\beta'2a$ when shock is withdrawn”

I would suggest to the authors to rephrase the conclusion. They show that the MBON activates the PAM- $\beta'2$ which is supported by connectome data and by previously published data. Therefore, the authors demonstrated that activity in the MBON drives PAM- $\beta'2$ neurons. However, this is not exclusively during shock withdrawal. In fact, during withdrawal it drives the MBON similarly strong than during first naïve response.

4) “Finally, we show that PPL1- $\gamma2\alpha'1$ shock-responsive DANs relieve synaptic depression of KC-MBON- $\gamma2\alpha'1$ synapses as a consequence of shock withdrawal to increase CS+ odor activation of MBON- $\gamma2\alpha'1$ ”

The authors do not show that PPL1- $\gamma2\alpha'1$ relieve synaptic depression. Instead, the authors show that activating PPL1- $\gamma2\alpha'1$ neurons in coincidence with CS+ presentation during reversal phase boosts CS+ aversion. This is in line with previous findings that artificial activating PPL1- $\gamma2\alpha'1$ neurons can substitute for electric shock punishment and depress KC- $\gamma2\alpha'1$ MBON synapses (e.g. Berry et al 2018). Thus, assigning additional negative valence to the CS+ during the reversal leads to a stronger CS+ avoidance. These unsurprising findings does not allow the conclusion that PPL1- $\gamma2\alpha'1$ activity is reduced during reversal. In addition, the claim is not backed up by the imaging data presented in Figure 1 and there for is not valid.

5)The omission situation during a classical extinction procedure (trained cue in absence of the unconditioned stimulus/reinforcement) is similar to parts of the reversal learning when the contingency of the two odors switches. However, in the reversal learning the relationship between the stimuli is more complex and multiple different learning processes might take place. Thus, it cannot be concluded that extinction is an isolated and independent building block of reversal learning. Applied manipulations during the omission of shock in the reversal training might affect other aspects of reversal learning. The authors should avoid to claim that they show that the investigated circuits are involved in extinction. Further, it should be emphasized that omission learning during reversal training might substantially differ from extinction protocols. The presented work, for the first time, reveal a circuit mechanism related to reversal learning in the fly, that is very

exciting (see also Berry et al 2018 Fig 4). However, they should make the distinction to a classical extinction experiment very clear and avoid calling the reversal learning linked omission extinction to prevent confusion. I would encourage the authors to discuss the findings on reversal learning in comparison to the findings in extinction and emphasize the overlaps and the differences.

6) The authors write that extinction is the unlearning of an association. That statement is incorrect, classically extinction is regarded as a new learning but of an opposing memory, a CS-NoUs memory. There might be states in which extinction can lead to an unlearning, however, they are the exception rather than the rule.

7) Throughout the manuscript it is claimed that it is not known how the dopaminergic reward pathway is driven during extinction of memories, e.g.: "...but it remains unknown in any model organism or learning paradigm how DAN signals are generated in response to withdrawal of negative reinforcement, nor how the absence of negative reinforcement is encoded as positive reinforcement." However, in Felsenberg et al 2017, 2018 and Otto et al 2020 evidence is presented that the changes in the mushroom body output network implemented during initial learning drives the dopamine system during memory extinction. In particular for aversive extinction it is shown that MBONs, that are necessary for extinction learning and that have an learning induced enhanced CS+ odor drive during odor re-exposure (extinction training), form functional connections to those dopamine neurons that drive extinction learning. Thus, the combined conclusion of Felsenberg et al and Otto et al is that the dopaminergic teaching signal during extinction is generated by learning induced changes in the output pathways, in the fly. Given the cumulative evidence for this conclusion it appears odd that the authors call this conclusion speculative and the mechanisms as unknown. The authors have to change the text accordingly.

8) The authors seem to be unaware that Otto et al in 2020 present behavioral data that supports the conclusion that extinction of aversive memories is mediate by specific $\gamma 5$ PAM neurons, $\gamma 5n$ but not $\gamma 5b$ neurons or PAM $\beta'2a$ neurons. Instead the authors call the conclusion that $\gamma 5$ PAM neurons are involved in extinction speculative. Concerning their own findings, the authors should consider that the driver line they use MB315C-GAL4 does not label all $\gamma 5$ PAM neurons but in fact those neurons which are not involved in aversive memory extinction (Otto et al 2020). Thus, the authors should adjust the text accordingly and tune down their claims about $\gamma 5$ PAM neurons. It would massively strengthen the presented work if the authors would image the remaining $\gamma 5$ neurons using additional tools to make comparative claims between extinction learning and reversal learning.

8) The authors make assumptions about the behavioral assay they use. To improve clarity and the improve the justifications of their conclusions, the authors should test these assumptions to verify them or state clearly why they make these assumptions. The presented behavioral data includes no evidence that the avoidance to the CS+ is reduced after reversal training. All tests include a choice between CS- and CS+. Thus, the test result after reversal, more flies in the CS+, could result merely from an increased aversion to the CS-. Thus, the authors should test if the CS+ response is reduced after reversal training when compared to a naïve odor. Further, it would improve clarity if the authors could put their behavioral data from Sup-Fig 1D in the main figure and make it very clear that a CS+ of avoidance of zero actually reflects a balanced avoidance to CS- and CS+ rather than the absence of a CS+ memory.

9) The training paradigm used during imaging and behavioral experiments differs. This has to be mentioned clearly in the text and possible implications should be mentioned. As it is presented there is no evidence that the protocol under the scope would lead to a change in the behavior of the animal in the behavioral assay. I strongly recommend, if possible, adding behavioral data which, as closely as technically possible, reassemble the protocol used under the scope, e.g. trial numbers or stimulation length.

Minor points:

- a reinforcer is a stimulus that increases a certain behavior during operant conditioning, though more frequently used in the fly learning literature the authors might want consider to use the term unconditioned stimulus to avoid confusions.
- negative and positive reinforcement are not equivalent to electric shock punishment and food reward, e.g. negative reinforcement is strengthening a behavior by removing a unpleasant consequence during operant conditioning. Thus, the authors might want reconsider the usage of these terms.
- The authors should seriously double check their citations, e.g. Yamazaki et al 2018 and Berry et al 2018 do not use reward learning but are cited for PPL1 neurons being involved in the acquisition of appetitive memory. Further, they miss to cite Burke et al 2012, Huetteroth et al 2015 and Yamagata et al 2014 as references for PPL1 and PAM DANs and Perisse et al 2016 and Lewis et al 2015 for avoidance and approach coding output neurons.
- All figures with average traces should have scale bars.
- The range of the axis should be the same for the similar kind of experiments, at least within one figure.
- In Sup Fig 1, there seems to be an error in the color coding. If not, then the effects presented are not matching the findings presented in Fig 1. Since the experiments is done to verify that the effects can be observed with the reversed odor sequence the authors have to mention the differences more clearly. This issue needs clarification and could become a matter of major concern. Especially the reversal learning has different dynamics for the CS-. The authors have to mention these differences very clearly in the main text.
- The authors should consider to discuss the feed-forward inhibition from MBON $\gamma 1$ ped to the horizontal tip MBONS including MBON $\gamma 5\beta'2a$ in more depth.
- The authors should use the prime symbol rather than the apostrophe when describing the respective neurons.

Reviewer #2 (Remarks to the Author):

In this manuscript, McCurdy et al provide a circuit framework through which dopamine neurons encode the omission of aversive outcomes that drive reversal of cue-outcome contingencies in *Drosophila*. The authors present an excellent set of in vivo functional imaging, optogenetic manipulation and behavioral analysis along with synaptic reconstructions from electron microscopy data. They demonstrate that PAM- $\beta'2a$ dopaminergic neurons (DANs) encode shock withdrawal during reversal as potentially rewarding. This dopaminergic reward signal plays a role in the extinction of shock-paired conditioned stimulus (CS+) shock association thus reducing CS+ avoidance by depressing KC synapses onto avoidance-encoding MBON- $\gamma 5\beta'2a$. Further, they show approach-encoding MBON- $\gamma 2\alpha'1$ acts as an upstream modulator that activates PAM $\beta'2a$ when shock is withdrawn and PPL1- $\gamma 2\alpha'1$ shock-responsive DANs relieve synaptic depression of KC-MBON- $\gamma 2\alpha'1$ synapses to increase CS+ odor activation. Together, these findings provide a mechanistic circuit explanation through which dopamine neurons provide an indirect relay between compartments in a memory structure to encode withdrawal of a punishment outcome to an appetitive outcome during reversal learning. This is an important contribution to our understanding of the fundamental mechanisms underlying memory formation.

I'd like to commend the authors on putting together such a cohesive and interesting paper. The writing is easy to follow, and data analyzed and presented in a rigorous and aesthetically clear way. Appropriate controls are present and provided in the supplemental data. Furthermore, the authors eloquently discuss the current literature, clearly outline the new information their data adds to the field, and thoughtfully explain their data using comparisons to circuit motifs in other animals, but without over-reaching. I really enjoyed reading the paper and thank the authors for their beautiful work. Recommendations listed below are minor concerns and intended only as suggestions to improve the rigor and readability of this important piece of work.

1) Although extinction has historically been presented as unlearning of an association, the authors may want to consider updating their definition of extinction in the introduction. Recent data suggests that extinction is not the unlearning of an association, but occurs as a parallel memory trace. I recommend redefining in a way that is most consistent with the evidence.

2) In Figure 1B, the training protocol depicted is not accompanied by data demonstrating that memory is formed with this protocol. Could the authors cite other work that shows this protocol produces memory or provide this behavioral data?

3) In order to ensure other people could reproduce this work, please include more detail on how 'difference in mean odor response $\Delta R/R$ ' (as in Fig 1C) is calculated in the methods

4) Please provide justification in the methods or results for why the behavior protocols for the optogenetics experiments differ with respect to timing of the stimulus, inter-trial stimulus, and number of repetitions/trials from the Ca^{2+} imaging experiments.

5) According to FlyLight, R66C08 is not specific to just the MBON-y5B'2a neuron, which is highly relevant for the optogenetic experiments conducted in Figure 2D-F. Please list off-target expression as detailed in FlyLight in the table in the methods section. Similar information should be provided for R39A05, R25D01-LexA and VT1211-LexA, which all look like they express in behaviorally relevant areas outside of their respective MBON targets.

6) The authors might be interested in connectivity present between MBON a'2 and MBON B'2mp as this lends support to their functional data in Figure 3 (Aso et al 2014 closeness of fluorescent images in registered brains, Scaplen et al 2020 trans-tango connectivity, Xu et al 2020 bioRxiv latest annotation of EM hemibrain shows synapses).

7) Please include justification in the methods / results for not including data with optogenetic activation of MBON-B'2mp (R39A05) for Figure 4A,B 100% duty cycle and 100% intensity and for Figure 4E 50% intensity.

8) For Figure 5: Have the authors looked at whether there are connections between the PPL1-y2a'1 neuron and PAM B'2a neurons? If there are synapses between these neurons, the imaging and behavioral effects the authors see could potentially be attributed to these connections rather than going through the MBONs as the authors propose. I recommend either demonstrating the connections between these DANs don't exist (or aren't extensive depending on thresholding in EM dataset) or acknowledging this possibility in the discussion.

- 9) It would be helpful to see the p values for the not significant results if they are close to significance in the supplement (for example in Supplemental Figure 2B the right plot looks potentially interesting)
- 10) Supplemental Figure 3A-C the far right plots are confusing because only one ns is included when both blue and red are ns. I recommend either not including * when data is not significant or including both blue and red ns when data are not significant.
- 11) Supplemental Figure 5: it would be helpful if the plots were labelled acquisition and reversal.
- 12) The authors might want to update bioRxiv citations in case papers were published.

Reviewer #3 (Remarks to the Author):

Review on the research article „Cholinergic relay from punishment- to reward-encoding dopamine neurons signals punishment withdrawal as reward in *Drosophila*” by McCurdy et al for Nature Communications.

This article by McCurdy et al. investigates the role of a neuronal circuit consisting of four neuron types for the acquisition and the reversal of an associative odour-electric shock memory in *Drosophila*. Combining Ca-imaging and optogenetic activation of neurons, the authors describe this circuit in high detail and complexity, and suggest a 4-step pathway how electric shock information (or the omission of it) can tune this system to eventually make the animal avoiding or approaching the trained odour. Although this is a highly interesting research that provides important insights into the learning circuits of *Drosophila*, I have several major and minor concerns that have to be addressed before publication.

Major:

I. In the introduction, the authors try to define and extinction learning and reversal learning, and in the discussion they clearly separate between neurons involved in either of those learning types. In the result section, however, they mix up these phenomena, and sometimes use “reversal”, “extinction”, “shock omission” or “shock withdrawal” for the same observation. Sometimes they write “extinction that happens during reversal”. This makes it difficult to follow their arguments. I think one origin of this problem is that technically the authors perform a reversal learning paradigm, i.e. they swap the contingencies of odours and shock (1. phase A+/B, 2. phase A/B+), but they discuss and focus only on the effects of the “missing” shock for odour A, and thus on a phenomenon that is very similar to extinction learning. A complete account of reversal learning should include also what happens to odour B. I strongly advise to clarify the used terminology, and to re-consider what is the topic the authors want to investigate here. Specifically, I recommend not to use “withdrawal” because of its connotations of “taking away something” – that’s not what the authors do, they rather omit an expected shock.

II. The authors repeatedly activate or silence a given neuron only during the presentations of the previous CS+ in the reversal training phase. Based on the observed changes in odour avoidance they

conclude specific functions of these neurons for memory reversal. I want to offer an alternative perspective: all the effects the authors see in these experiments can be explained by rewarding or punishing properties of the neurons under study that may not be limited to reversal learning at all: PAM- $\beta'2a$: This neuron is even published as rewarding neuron, as the authors mention in the discussion. Nevertheless, the authors claim that this neuron is not generally rewarding based on an one-training-trial experiment (Fig. S1). In contrast, in the reversal learning experiment (Fig. 1), they use two training trials – this can make a big difference for memory scores (compare also Fig. S1D and S1F). Therefore, this control experiment is not sufficient to rule out a rewarding function of the neuron. Furthermore, Yamagata et al. 2016 (PLOS Biology) showed that silencing a DAN can have the opposite reinforcing function as compared to activating it. That means, in the experiment shown in Fig. 1J the authors may actually provide a virtual punishment by silencing PAM- $\beta'2a$, which would explain the result.

MBON- $\gamma5\beta'2a$ and MBON- $\gamma2\alpha'1$: König et al. 2019 (Biol Letters) demonstrated that also activating and silencing of MBONs can have reinforcing effects, with opposite signs for activating or silencing the neuron. That means also the authors' experiments (Fig. 2D-F, Fig. 4A-C) could be explained by reinforcing functions of the MBONs they use.

PPL1- $\gamma2\alpha'1$: here the authors' data suggest that it signals electric shock punishments: it's activated by shock, and animals behave as if there would have been a shock when the neuron was activated. However, the authors conclude that it is "uniquely necessary" for "encoding of withdrawal of shock". This specific role is not supported by any data – the most parsimonious explanation is that this neuron is active when there is shock, and not active when there is no shock; activating it simulates a shock punishment, and therefore no reversal is observed.

IF the authors want to make the claim that the effects they see in these experiments are specific for memory reversal, they need to do proper control experiments by activating or silencing every neuron under study with the very same parameters as in their reversal experiment – just omitting the first training phase. Alternatively, the authors should discuss the alternative explanations that I mentioned, and tone down their conclusions from the behavioural experiments.

III. In Fig. 5B-E, the authors activate the PPL1- $\gamma2\alpha'1$, the first neuron in the chain, and image MBON- $\gamma5\beta'2a$, the last neuron in the chain. First, they want to show that without activating PPL1- $\gamma2\alpha'1$ odour responses of MBON- $\gamma5\beta'2a$ increase upon acquisition, and decrease after reversal (Fig. 5D). This result is based on a sample size of 3, and not statistical significant. Even if an analogous experiment has been done before (with another genotype!), this sample size is just too low to allow any meaningful conclusions. Then, they show that with activation of MBON- $\gamma5\beta'2a$, the reduction of odour response does not occur (Fig. 5E). Here, the decisive statistical test is missing – between "post-acq" and "post-rev". The authors should add the respective test in both Fig. 5D and 5E. Furthermore, they should increase the sample size for the experiment of Fig. 5D, or find another, statistically valid way to make their point.

Minor:

1. For the screen results (e.g. Fig. 1C-D), the black lines and individual data points are difficult to discriminate. Also, what is the meaning of the stippled vertical line in Fig. 1D?
2. For many of the Ca-imaging experiments, the authors always use one given odour chemical as CS+, and the other one as CS-. Only for some of the experiments, they performed the reciprocal experiment as supplement. This is problematic as any differential effects for CS+ and CS- can be

confounded by differential responses of the neuron to the particular chemical substance rather than to the experimental manipulation. This problem gets particularly obvious for PAM- β '2a (Fig. 1/S1) because this neuron responds drastically different to the two odours. The authors should clearly state and discuss this potential confounding effect.

3. I would suggest adding a display like in Fig. 5D-E to all Ca-Imaging results, i.e. the changes of odour responses after acquisition and reversal with a common reference. This would make clear that all the changes the authors observe during acquisition are reversed back to baseline after reversal. The way the figures 1-4 are designed at the moment obscures this observation by using a different reference for acquisition and reversal.

Along the same lines, I suggest that the authors discuss this observation more clearly because it is actually a very interesting finding. Extinction is often regarded not as the "un-learning" of a previous memory, but as a new, additional memory that suppresses the first one. The observation that in the circuit under study the original memory trace seems to be lost during reversal is therefore very interesting – and seems to be in contrast to what happens in other circuits: MBON- γ 1ped, for example, does not restore the reduced CS+ response after reversal, but rather seems to keep the original memory trace (Fig. 3E).

4. I feel that the experiment in Fig. S1D deserves to be emphasized much more. This is the single experiment in the whole study showing that the reversal learning paradigm the authors use is actually working as intended, and it is not even mentioned with a single sentence in the main text! The authors should at least clearly refer to this result, or may even consider moving it to Figure 1.

5. I have a hard time to follow the rationale and conclusions of Fig. 4A-E:

First, I would ask the authors to explain the term "duty cycle" that is unknown to me and is also not used anywhere else in the study.

Second, the authors conclude from Fig. 4B that activation of the two MBONs have opposite effects on odour avoidance. However, contemplating the lowest intensity, for both neurons a reduction of avoidance is observed (although not significant in one case); for the highest intensity, this effect seems to be reversed for one of the MBONs – yet this is solely due to a change in the control group, whereas the experimental group behaves the same. Also, the other MBON is not tested under these conditions, so we don't know which effects it would have here.

Third, I understand that Fig. 4E demonstrates that light intensities can matter for behavior. But why do the authors use completely different intensity levels here (1-50% versus 50-100% in the previous experiment)?

The authors should state more clearly why which parameter was chosen, and discuss the potential caveats of this experiment.

6. In Fig. 4H, the authors activate two MBONs and measure the odour response of PAM- β '2a. With a low sample size and an almost statistically significant increase in odour response, they simply state that no effect was observed. I think this point deserves more discussion, given that one of the activated MBONs was actually supposed to decrease responses (the opposite of the observed trend), and the other neuron was supposed to be downstream of PAM- β '2a and therefore should have no effect. How do the authors explain this result?

7. To activate Chrimson, the authors use 3 different wavelengths in different experiments, and only two of them are mentioned in the methods section. Please explain.

8. Please give the exact sample sizes of each experimental condition, not just a range. This is particularly important when the sample sizes are strongly different (e.g. 4-13).
9. Page 4: please give a reference for the statement that *Drosophila* are capable of reversing cue-outcome associations.
10. When discussing the results of the first screen, please be consistent in describing the observed effects as increased/decreased CS+ responses in comparison to CS- responses. I think always mentioning your reference (the CS- response) is important. Please check this also throughout the text.
11. Page 12: The authors refer to previously reported rewarding or punishing valences of MBONs by Aso et al. 2014. This is very misleading. These MBONs receive input from reward- or punishment-signalling DANs, and they are approach- or avoidance-promoting. Whether they may have rewarding or punishing properties themselves, was not studied by Aso et al.
12. Page 24: The measurement of CS+ avoidance is described as the fraction of flies in the CS- quadrants. Mathematically, this would be: $(\text{number of flies at CS-})/(\text{total number of flies})$. However, this is not the formula the authors use.
13. Page 25: I do not understand the sentence ending with “with no illuminated quadrants 30s after”.
14. Also Page 25: Do I understand the sentence beginning with “This was repeated, often with the same group of flies ...” correct that some groups of flies were used several times within one experiment, whereas others were used only once? This may be a critical confound of the data. Please specify precisely how you used flies repeatedly, and for which experimental conditions.
15. Page 26: The reference to Fig. 5H,J is wrong.

Reviewer #1:

McCurdy et al investigate the neural circuitry underlying aversive learning and reversal learning. By combining imaging, behavioral assays and optogenetic approaches the authors provide novel insights into how changes in contingencies of learned cues relate to learning dependent plasticity, to the activity of the dopaminergic system and to the valence coding output system within the mushroom body of fruit flies. The findings are of general importance and will raise interest beyond the fly learning and memory field. As positive as I am about the data presented, as much I disagree with many of the interpretation of the results and consider some of them for overstated. With all the respect for the work, the proposed model is not backed up by the data and therefore likely to be wrong. In fact, below I will challenge most of their claims point by point. I would like to emphasize that I do not question the quality of the data but its interpretation given the current state of experiments. Most importantly, the authors clearly under-appreciate the importance of their findings in relation to reversal learning. They neglect that, to my knowledge, they provide the first comprehensive data set on reversal learning related plasticity in the fly. Rather than solely focusing on the extinction-like omission of electric shock situation during reversal, the manuscript would massively improve if the authors would acknowledge that extinction and reversal learning are potentially different processes. Further, the authors should revisit their literature citations because some of them are inadequate. In am aware that, due to limited access to experimental set-ups in these special times, it is hard to add additional data but I will have to point out where crucial controls are missing. I support the communication of the data but not in the current conceptual setting nor with the proposed model. The authors must substantially re-work the manuscript to incorporate the outlined criticisms. I will address the issues point by point, hoping that they will be helpful, and I am looking forward to reading the authors response.

We thank Reviewer 1 for their detailed suggestions for improvement. The alternate interpretations suggested motivated us to think harder about our model, and perform substantial additional experiments and analyses. We performed all experiments proposed, and revised the interpretation of our work as suggested. We also refined our terminology and highlighted the differences between extinction and reversal learning.

First (1-4) I will go through the major claims one by one as they are stated in the manuscript, then address other major concerns followed by minor issues.

1) “We establish that PAM- β '2a DANs encode shock withdrawal during reversal as rewarding.” The data presented in Fig 1 C-Fii shows that the PAM β '2a responses to the CS+ decreases during aversive training and increases again during reversal learning. From the traces presented it becomes clear that the PAM β '2a responses during reversal return to the baseline of the first naïve-odor response (Fig 1 Fi and Fii). Thus, at no point in the entire protocol are the responses in these dopamine neurons elevated above the naïve odor response. Thus, the claim that PAM β '2a neurons encode the shock withdrawal and provides a rewarding teaching signal is not valid. In general, in all the imaging data, for all neurons investigated, the authors should visualize the effects of reversal trials 6 and 7 by normalizing and plotting it in the same panel then acq. trials 1-5 to avoid misinterpretations.

This is an interesting and important point. It is true that PAM- β '2a CS+ odor response does not increase beyond its naïve CS+ odor response during reversal. We think this makes sense, considering how the “rewarding” stimulus during reversal is the lack of a stimulus. The absence of electric shock is *relatively* rewarding when compared to the presence of electric shock. However, the absence of electric shock is no more rewarding in *absolute* terms than when no shock is ever presented. We would hypothesize that an increase in PAM- β '2a CS+ odor response relative to its naïve CS+ odor response would only occur when the odor is paired with an actual reward such as sugar. Hence, we view the increase in PAM- β '2a CS+ odor response during reversal to be a *relative* reward signal. This is consistent with our model, as it explains why the CS+ odor response initially decreases during acquisition.

We have edited our Discussion section to more clearly articulate this interpretation on Page 19:

It is important to note that the increase in PAM- β '2a CS+ odor response does not exceed the initial naïve odor response. This is consistent with our model in which PAM- β '2a encodes a relative reward signal, i.e., that shock omission is relatively more rewarding than the presence of shock, but the absence of aversive stimuli is not inherently rewarding in absolute terms. This also explains why a decrease in CS+ odor response occurs during acquisition (Fig. 1g), even though PAM- β '2a activity is not necessary for aversive memory acquisition (Supplemental Fig. 2a).

As suggested, we have also replotted all imaging data from reversal trials to be normalized to acquisition trial #1, to more easily compare the changes in odor response during reversal relative to the initial odor response.

Further, the authors test if activating or inhibiting PAM β '2a neurons during the CS+ during the two reversal trials changes behavior. The observed results are consistent with their claim. However, there is an alternative explanation equally valid given the presented controls: it might be that PAM β '2a activity, independent of reversal learning, provides a reward-like teaching signal and that the absence of the activity assigns a negative value to the odor similar to other PAM neurons (PAM γ 3, Yamagata et al 2016). To control for this alternative explanation, one has to repeat the same reversal protocol, at least in terms of number of trials, to test if pairing an odor with the activity of the tested neurons changes behavior. The experiments presented by the authors don't do that, e.g. the use one instead of two trials in the activation experiment or block throughout single aversive training trial in the inhibition experiment which does not control if the inhibition of the neurons provides a teaching signal. Therefore, the presented controls are invalid to exclude that alteration of PAM β '2a activity during two trials generates a teaching signal. Without the correct controls the experimental results from the reversal learning are inconclusive. Thus, since there is no increase in PAM β '2a activity coinciding with the CS+ during shock withdrawal in reversal, and the behavioral experiments being inconclusive, the claim is not supported.

We agree that the control behavioral experiments we provided could be expanded upon, and have conducted the suggested additional control experiments (Fig. 2c,d). As the reviewer predicted, we indeed found that manipulating PAM- β '2a activity in the presence of an odor can bidirectionally alter the valence associated with the odor: activating PAM- β '2a during an odor increases attraction to the odor, and silencing PAM- β '2a during the odor increases avoidance of

the odor. This supports the suggested interpretation that PAM- β '2a provides a reward-like teaching signal, independent of reversal learning.

These additional findings are consistent with the ability for PAM- β '2a to cause changes in valence associated with an odor, and consequently lead to changes in odor preference. It is true that these behavioral changes would also be observed by manipulating other PAM subsets (e.g. PAM- γ 3, as the reviewer points out). This is why we are convinced that our imaging results in **Fig 1** are crucial for identifying which neurons actually change their odor response during reversal. We established that PAM- β '2a is the *only* DAN that changes its odor response during reversal. Thus our behavioral data, when combined with the imaging data, supports our model of PAM- β '2a being the DAN that drives the changes in odor preference during reversal learning, as well as during naïve odor pairing.

2) “This dopaminergic reward signal extinguishes CS+-shock association, and hence reduces CS+ avoidance, by depressing KC synapses onto avoidance-encoding MBON- γ 5 β '2a.” Related to 1) it is not clear if the PAM β '2a provides a reward like teaching signal during reversal. The authors show that activating PAM β '2a artificially in coincidence with an odor leads to a depression of the MBON- γ 5 β '2a drive for that odor. However, as outlined in 1) there is no increase in PAM β '2a during reversal learning. Instead, the data shows that whenever PAM β '2a is decreased MBON- γ 5 β '2a responses are enhanced suggesting a direct modulation of KC-MBON- γ 5 β '2a synapse efficiency by the activity of PAM β '2a without plasticity. In fact, it has been frequently reported that PAM β '2a activity can modulate odor driven behavior in a context or state dependent manner (Lewis et al 2015 Senapati et al 2019, Scalpen et al 2020). In fact, the state-dependent control involves the axis from γ 2 α '1 MBON and PPL1 neuron similar to the finding of the authors and supports the idea of a cross-talk between the aversive/state controlling dopamine pathways and the MBON- γ 5 β '2a. The authors should consider these alternative interpretations of their findings. A valuable additional data set would be, to repeat the experiment presented in Figure 2G but look at the odor response during PAM β '2a activation to test if its activity can directly inhibit the odor drive of the MBON.

This is an important alternative possibility for explaining the opposite changes in odor response between PAM- β '2a and MBON- γ 5 β '2a. To address this, we performed the experiment suggested by the reviewer, and found that PAM- β '2a activation during odor response does not directly inhibit MBON- γ 5 β '2a odor response (**Supplemental Fig. S3e**). This supports our model that activation of PAM- β '2a in the presence of an odor causes a decrease in MBON- γ 5 β '2a odor response via synaptic plasticity.

In addition to this new reviewer-suggested experiment, this alternative explanation also would not explain the decrease in MBON- γ 5 β '2a odor response in the functional connectivity experiment (**Fig. 3f-h**). In that experiment, PAM- β '2a is not artificially activated in the ‘post’ phase, and thus the decrease in MBON- γ 5 β '2a odor response cannot be due to direct inhibition from PAM- β '2a, and is more likely due to changes in synaptic plasticity.

3) “We further demonstrate that approach-encoding MBON- γ 2 α '1 is an upstream element that activates PAM- β '2a when shock is withdrawn” I would suggest to the authors to rephrase the conclusion. They show that the MBON activates the PAM- β '2 which is supported by connectome data and by previously published data. Therefore, the authors

demonstrated that activity in the MBON drives PAM- β '2 neurons. However, this is not exclusively during shock withdrawal. In fact, during withdrawal it drives the MBON similarly strong than during first naïve response.

In this statement, we did not mean to imply that the activation of PAM- β '2a by MBON- γ 2 α '1 only occurs during reversal learning. Indeed, the odor response of MBON- γ 2 α '1 very likely activates PAM- β '2a also under naïve conditions.

We have edited the sentence to clarify (Page 6):

We further demonstrate that approach-encoding MBON- γ 2 α '1 is an excitatory upstream element of PAM- β '2a which causes the changes in PAM- β '2a CS+ odor response during acquisition and reversal.

We also address this concern more explicitly in the Discussion section (Page 21):

Both MBON- γ 2 α '1 and PAM- β '2a respond to naïve odors, so it is likely that MBON- γ 2 α '1 contributes to the naïve odor response in PAM- β '2a, resulting in a particular valence associated with the odor. They both also decrease CS+ odor response during acquisition and increase during reversal. Our findings support the hypothesis that this relative increase in CS+ odor response during reversal skews the valence associated with the CS+ odor to decrease CS+ odor avoidance.

4) “Finally, we show that PPL1- γ 2 α '1 shock-responsive DANs relieve synaptic depression of KC-MBON- γ 2 α '1 synapses as a consequence of shock withdrawal to increase CS+ odor activation of MBON- γ 2 α '1” The authors do not show that PPL1- γ 2 α '1 relieve synaptic depression. Instead, the authors show that activating PPL1- γ 2 α '1 neurons in coincidence with CS+ presentation during reversal phase boosts CS+ aversion. This is in line with previous findings that artificial activating PPL1- γ 2 α '1 neurons can substitute for electric shock punishment and depress KC- γ 2 α '1 MBON synapses (e.g. Berry et al 2018). Thus, assigning additional negative valence to the CS+ during the reversal leads to a stronger CS+ avoidance. These unsurprising findings does not allow the conclusion that PPL1- γ 2 α '1 activity is reduced during reversal. In addition, the claim is not backed up by the imaging data presented in Figure 1 and there for is not valid.

The reviewer is correct in pointing out that we did not see changes in PPL1- γ 2 α '1 odor response during acquisition or reversal. What we were referring to when we wrote that PPL1- γ 2 α '1 “relieves synaptic depression of KC-MBON- γ 2 α '1 synapses as a consequence of shock withdrawal” is the fact that during reversal, PPL1- γ 2 α '1 does not respond to electric shock, because the shock is no longer presented during CS+. Hence PPL1- γ 2 α '1 activity during CS+ presentation is less during reversal compared to acquisition. We argue that this relatively less PPL1- γ 2 α '1 activity during CS+ during reversal is what relieves synaptic depression of the KC-MBON- γ 2 α '1 synapses, leading to an increase in MBON- γ 2 α '1 odor response. To clarify in accordance with the reviewer suggestion, we have edited the text as follows (Page 6):

Finally, we show that the absence of shock response by PPL1- γ 2 α '1 shock-responsive DANs during reversal learning when CS+ is presented without shock relieves synaptic

depression of KC-MBON- γ 2 α '1 synapses to increase CS+ odor activation of MBON- γ 2 α '1.

We agree that our findings regarding the changes in odor response and behavior caused by PPL1- γ 2 α '1 manipulation are consistent with PPL1- γ 2 α '1's established role as a punishment-encoding DAN. However, we believe that these data are usefully included in the manuscript as support for our model. While it is true that activating other PPL1 neurons could have the same effect on behavior, these data reinforce our model, as if we did not get these expected results, that would invalidate our model.

Importantly, we do not mean to imply that PPL1- γ 2 α '1 is *only* involved in signaling shock omission during reversal learning, as it is clearly involved in other processes, including the acquisition of aversive memories. We also do not mean to imply that *only* manipulating PPL1- γ 2 α '1 can result in the changes in neural activity and behavior observed in **Fig. 6**. Indeed, manipulating other punishment-encoding DANs, e.g. PPL1- γ 1ped, would likely have the same effect. But considered in combination with our imaging studies and the extensive data supporting the other three nodes in our circuit (MBON- γ 2 α '1, PAM- β '2a and PAM- γ 5 β '2a), it is reasonable to propose PPL1- γ 2 α '1 as the most upstream first node in the circuit.

To clarify this point, we have added the following text to the Discussion (Page 21):

Here we demonstrate a role for MBON- γ 2 α '1 and PPL1- γ 2 α '1 in encoding shock omission as reward during reversal learning, in addition to their known roles in aversive memory formation.

5) The omission situation during a classical extinction procedure (trained cue in absence of the unconditioned stimulus/reinforcement) is similar to parts of the reversal learning when the contingency of the two odors switches. However, in the reversal learning the relationship between the stimuli is more complex and multiple different learning processes might take place. Thus, it cannot be concluded that extinction is an isolated and independent building block of reversal learning. Applied manipulations during the omission of shock in the reversal training might affect other aspects of reversal learning. The authors should avoid to claim that they show that the investigated circuits are involved in extinction. Further, it should be emphasized that omission learning during reversal training might substantially differ from extinction protocols. The presented work, for the first time, reveal a circuit mechanism related to reversal learning in the fly, that is very exciting (see also Berry et al 2018 Fig 4). However, they should make the distinction to a classical extinction experiment very clear and avoid calling the reversal learning linked omission extinction to prevent confusion.

We agree that extinction learning and reversal learning are similar albeit distinct processes, and appreciate the suggestions for how to clarify our use of terminology. While both extinction learning and reversal learning involve the formation of a new association between CS+ and shock omission, we of course agree that the mechanisms underlying the formation of this association are very likely different in the context of an extinction learning paradigm versus a reversal learning paradigm. Indeed, our model of how this new association is formed applies only to reversal learning, as the increase in PAM- β '2a CS+ odor response during reversal is

contingent on the increase in MBON- $\gamma 2\alpha'1$ CS+ odor response, which, as in Berry et al 2018, occurs much more rapidly during reversal learning than extinction learning. (Further evidence of differences between extinction learning and reversal learning is presented later in point #8 of this review.) We have edited the text in the Introduction to clarify this (Page 5):

Here we use a combination of functional imaging, optogenetics, and behavioral approaches in Drosophila to answer these key issues in the biology of reversal learning, specifically the neural mechanisms underlying the formation of the association between CS+ and shock omission.

We have also minimized the use of the term “extinction” when describing the changes in CS+ odor response and behavior during reversal learning, and replaced it with terms like “CS+ reversal” or “formation of association between CS+ and shock omission”.

I would encourage the authors to discuss the findings on reversal learning in comparison to the findings in extinction and emphasize the overlaps and the differences.

This is a great suggestion for both informing a broader audience and communicating carefully to experts in the field. We have thus included additional explanations in the Discussion section:

*The involvement of avoidance-encoding MBON- $\gamma 5\beta'2a$ in reversal learning is consistent with its similar role in extinction learning (Felsenberg et al., 2018). In both extinction and reversal learning, MBON- $\gamma 5\beta'2a$ CS+ odor response increases during acquisition and decreases when CS+ is no longer presented with shock. In the case of extinction learning, the authors identified that the increase in MBON- $\gamma 5\beta'2a$ CS+ odor response during acquisition causes an increase in PAM- $\gamma 5$ CS+ odor response, which could then induce synaptic depression at KC-MBON- $\gamma 5\beta'2a$ synapses to reduce MBON- $\gamma 5\beta'2a$ CS+ odor response during extinction. More recent work has identified a specific subset of PAM- $\gamma 5$ neurons, the lower commissure PAM- $\gamma 5(fb)$ neurons, as necessary for memory extinction, and uncovered recurrent connections from MBON- $\gamma 5\beta'2a$ to those PAM neurons as a likely mechanism underlying aversive memory extinction (Otto et al., 2020). Interestingly, we did not observe any changes in odor response in PAM- $\gamma 5(fb)$ neurons during acquisition or reversal (**Supplemental Fig. S1d-f**). Likewise, unlike MBON- $\gamma 5\beta'2a$ which changes its CS+ odor response both during extinction and reversal learning, we and others have uncovered some MBONs (e.g., MBON- $\gamma 2\alpha'1$ and MBON- $\beta'2mp$) which change their CS+ odor response during extinction learning and not reversal learning (Berry et al., 2018; Felsenberg et al., 2018), providing additional evidence that distinct neural circuits underlie extinction and reversal learning. (Page 20)*

...
*Taken together, our findings support the model of parallel opposing memories, first demonstrated in Drosophila extinction learning in (Felsenberg et al., 2018), as the basis of reversal learning. In this model, the $\gamma 1ped$ microcircuit encodes the originally acquired aversive memory (Hige et al., 2015; Perisse et al., 2016; Felsenberg et al., 2018). Indeed, we found that MBON- $\gamma 1ped$ does not increase its CS+ odor response during reversal (**Fig. 4e**), even after 6 reversal trials (data not shown), demonstrating the persistence of the original memory stored in the $\gamma 1ped$ microcircuit. MBON- $\gamma 5\beta'2a$ on*

the other hand is responsible for forming the CS+-no shock memory via PAM- γ 5(fb) in extinction learning, and is the last node in the circuit for forming the CS+-no shock memory via the γ 2 α '1- β '2 relay network we uncover here in reversal learning. MBON- γ 5 β '2a is poised to integrate the initial aversive memory with the new no-shock memory via the direct feed-forward inhibitory connection from MBON- γ 1ped (Felsenberg et al., 2018). (Page 22)

6) The authors write that extinction is the unlearning of an association. That statement is incorrect, classically extinction is regarded as a new learning but of an opposing memory, a CS-NoUs memory. There might be states in which extinction can lead to an unlearning, however, they are the exception rather than the rule.

We have replaced the term 'unlearning' with 'updating' in the Summary on Page 2. We have also edited the manuscript so that use of the term "extinction" is reserved only for extinction learning. We have replaced "extinction of the aversive CS+ memory" in the context of reversal learning with "formation of the association between CS+ and shock omission". Some examples from the edited manuscript of this are:

In Drosophila, as in mammals, reward-encoding DANs are thought to underlie the extinction of aversive memories via the formation of a parallel association between the CS+ cue and the omission of punishment (Felsenberg et al., 2018; Otto et al., 2020). However, those studies are of extinction learning, and it is less clear whether a similar process occurs during reversal learning. (Page 5)

Here we use a combination of functional imaging, optogenetics, and behavioral approaches in Drosophila to answer these key issues in the biology of reversal learning, specifically the neural mechanisms underlying the formation of the association between CS+ and shock omission. (Page 5)

Here we combine functional imaging and optogenetic interrogation to provide extensive evidence for a lateral relay connecting the γ 2 α '1 MB microcircuit to the γ 5 β '2a microcircuit to encode punishment omission as reward during reversal learning. (Page 19)

7) Throughout the manuscript it is claimed that it is not known how the dopaminergic reward pathway is driven during extinction of memories, e.g.: "...but it remains unknown in any model organism or learning paradigm how DAN signals are generated in response to withdrawal of negative reinforcement, nor how the absence of negative reinforcement is encoded as positive reinforcement." However, in Felsenberg et al 2017, 2018 and Otto et al 2020 evidence is presented that the changes in the mushroom body output network implemented during initial learning drives the dopamine system during memory extinction. In particular for aversive extinction it is shown that MBONs, that are necessary for extinction learning and that have an learning induced enhanced CS+ odor drive during odor re-exposure (extinction training), form functional connections to those dopamine neurons that drive extinction learning. Thus, the combined conclusion of Felsenberg et al and Otto et al is that the dopaminergic teaching signal during extinction is generated by learning induced changes in the output pathways, in the fly. Given the cumulative evidence for this conclusion it appears odd that the authors call this

conclusion speculative and the mechanisms as unknown. The authors have to change the text accordingly.

We thank the reviewer for pointing this out. We have substantially edited our manuscript to carefully acknowledge previous work on these neural circuits, especially specific dopaminergic subsets, involved in extinction learning. Please see the text from our edited manuscript quoted to address point #5 earlier, from Pages 20 and 22 of the Discussion section, where we elaborate on this.

8) The authors seem to be unaware that Otto et al in 2020 present behavioral data that supports the conclusion that extinction of aversive memories is mediate by specific $\gamma 5$ PAM neurons, $\gamma 5n$ but not $\gamma 5b$ neurons or PAM $\beta'2a$ neurons. Instead the authors call the conclusion that $\gamma 5$ PAM neurons are involved in extinction speculative. Concerning their own findings, the authors should consider that the driver line they use MB315C-GAL4 does not label all $\gamma 5$ PAM neurons but in fact those neurons which are not involved in aversive memory extinction (Otto et al 2020). Thus, the authors should adjust the text accordingly and tune down their claims about $\gamma 5$ PAM neurons.

We thank the reviewer for pointing us to the very recent work by Otto et al. The discovery of the specific PAM- $\gamma 5$ neurons revealed in that work is an important finding that we have now included in the revised Discussion section, as presented earlier (Pages 20 and 22).

It would massively strengthen the presented work if the authors would image the remaining $\gamma 5$ neurons using additional tools to make comparative claims between extinction learning and reversal learning.

As requested, we performed the reversal learning imaging experiment using the PAM- $\gamma 5$ line used in Otto et al.. Consistent with the reviewer's recommendation to make clear that extinction learning and reversal learning are distinct learning processes, we did not observe any changes in PAM- $\gamma 5(fb)$ odor responses during reversal learning (**Supplemental Fig. S1d-f**). We thank the reviewer for the suggestion of this experiment, which highlights the distinct mechanisms underlying extinction learning and reversal learning.

These findings are discussed in the Results section (Page 8):

*Although we did not see any change in odor response in PAM- $\gamma 5$ using our particular split-GAL4 line (MB315C-GAL4), we tested an additional PAM- $\gamma 5$ line which targets different PAM- $\gamma 5$ neurons to those used in the original screen, as these particular neurons (PAM- $\gamma 5(fb)$) were recently discovered to be involved in the extinction of aversive memories during extinction learning (Otto et al., 2020). We found no changes in odor response during acquisition or reversal, suggesting distinct mechanisms underlying extinction versus reversal learning (**Supplementary Fig. S1d-f**).*

9) The authors make assumptions about the behavioral assay they use. To improve clarity and the improve the justifications of their conclusions, the authors should test these assumptions to verify them or state clearly why they make these assumptions. The presented behavioral data includes no evidence that the avoidance to the CS+ is reduced after reversal training. All tests include a choice between CS- and CS+. Thus, the test

result after reversal, more flies in the CS+, could result merely from an increased aversion to the CS-. Thus, the authors should test if the CS+ response is reduced after reversal training when compared to a naïve odor. Further, it would improve clarity if the authors could put their behavioral data from Sup-Fig 1D in the main figure and make it very clear that a CS+ of avoidance of zero actually reflects a balanced avoidance to CS- and CS+ rather than the absence of a CS+ memory.

This is an important point and we thank the reviewer for this helpful suggestion. We have performed this experiment and our results demonstrate that the decrease in CS+ avoidance during reversal is due to a combination of the formation of the new association between CS+ and shock omission as well as the formation of the new CS- with shock association (**Fig. 2a-c**). Thanks to this experiment, we can compute the relative contribution of each memory to resulting behavior: approximately one-third of the decrease in CS+ avoidance is attributable to the new association between CS+ and shock omission, and two-thirds is due to the new association between CS- and shock.

We have also moved **Supplemental Fig. 1D** to the main figure (**Fig. 2a-c**), as suggested.

10) The training paradigm used during imaging and behavioral experiments differs. This has to be mentioned clearly in the text and possible implications should be mentioned. As it is presented there is no evidence that the protocol under the scope would lead to a change in the behavior of the animal in the behavioral assay. I strongly recommend, if possible, adding behavioral data which, as closely as technically possible, reassemble the protocol used under the scope, e.g. trial numbers or stimulation length.

The differences in training paradigm between imaging and behavioral experiments are in part due to the differences in setup. For example, the stimulus durations of both the odor and electric shock are shorter in the imaging experiments versus the behavioral experiments (5s and 100ms versus 20s and 1.5s). This is because in the imaging experiments, the fly is head-fixed and receives both the odor and electric shock in the exact same place, since it cannot move around. In the behavioral experiments, flies are trained en masse and they can and do jump around in the shock tube to briefly escape the electric shock.

To further validate our imaging setup, we performed the experiment suggested by the reviewer (Fig. 1c). We translated the imaging training paradigm to the behavioral training paradigm as much as possible: 5s odor pulses with 55s air in between, for 5 trials, with 2 reversal trials after. The only difference is the shock duration, which is necessary as discussed above. Flies were able to acquire and then reverse associations using this paradigm.

Additionally, we performed a hybrid experiment in which we conditioned each fly in the imaging setup (without cuticle dissection or head-fixing), and after 5 acquisition trials, placed the fly in the quadrant arena (Supplemental Fig. S1a). We observed a difference in MCH odor preference, depending on which odor the fly was conditioned to associate electric shock with. This suggests that shock conditions of the imaging setup are sufficient to form behaviorally relevant associations.

These new suggested experiments reinforce validity of that the protocol used in the imaging

experiments, and that the changes in neural activity observed in the imaging rig are responsible for the changes in behavior observed in the behavioral experiments.

We also now discuss these differences explicitly in the Results section:

Although the experimental protocol used in the imaging and behavioral experiments differ slightly, our data demonstrate that both setups lead to behavioral change, specifically an avoidance of CS+ after acquisition, and a decrease in avoidance after reversal (Fig. 1c, Supplemental Fig. S1a, Fig. 2a-c).

Minor points:

11) a reinforcer is a stimulus that increases a certain behavior during operant conditioning, though more frequently used in the fly learning literature the authors might want consider to use the term unconditioned stimulus to avoid confusions.

To address this clarification of learning theory terminology, we have replaced throughout the manuscript the term ‘reinforcer’ with ‘unconditioned stimulus’, ‘aversive stimuli’ or ‘punishment’.

12) negative and positive reinforcement are not equivalent to electric shock punishment and food reward, e.g. negative reinforcement is strengthening a behavior by removing a unpleasant consequence during operant conditioning. Thus, the authors might want reconsider the usage of these terms.

In accordance with this suggestion, we have replaced throughout the manuscript the term ‘positive/negative reinforcement’ with ‘rewarding/punishing stimuli’.

13) The authors should seriously double check their citations, e.g. Yamazaki et al 2018 and Berry et al 2018 do not use reward learning but are cited for PPL1 neurons being involved in the acquisition of appetitive memory. Further, they miss to cite Burke et al 2012, Huetteroth et al 2015 and Yamagata et al 2014 as references for PPL1 and PAM DANs and Perisse et al 2016 and Lewis et al 2015 for avoidance and approach coding output neurons.

We have edited the manuscript to correctly reflect these citations, and added the additional ones suggested.

14) All figures with average traces should have scale bars.

Scale bars are already included in each figure where average traces are presented, under the schematic and image of each neuron’s expression pattern. We have now shifted their positions to be more easily seen.

15) The range of the axis should be the same for the similar kind of experiments, at least within one figure.

We agree and have changed all axes to be similar for similar experiments.

16) In Sup Fig 1, there seems to be an error in the color coding. If not, then the effects presented are not matching the findings presented in Fig 1. Since the experiments are done to verify that the effects can be observed with the reversed odor sequence the authors have to mention the differences more clearly. This issue needs clarification and could become a matter of major concern. Especially the reversal learning has different dynamics for the CS-. The authors have to mention these differences very clearly in the main text.

We appreciate the reviewer's keen observation, and acknowledge that the results are not immediately obvious. This is due to the different naïve odor responses to MCH and OCT by PAM- β '2a. We consistently observed that the naïve odor response to MCH is much higher than OCT. As such, the decrease and subsequent increase in CS+ odor response (MCH) in Figure 1 is visually obvious. However, because the naïve odor response to OCT is low, its decrease and subsequent increase in Supplemental Figure 1, where OCT is CS+, is not visually striking. However, it is statistically significant.

Analyzing it a different way, the change in OCT CS+ odor response during acquisition in **Supplemental Fig. 1b** is a 92% decrease (mean \pm SEM = 1.64 ± 0.41 in trial 1, 0.12 ± 0.12 in trial 5), while the change in MCH CS- response is only 32% (13.75 ± 2.33 in trial 1, 9.31 ± 1.58 in trial 5). For comparison, the change in MCH CS+ odor response during acquisition in **Fig 1g** is a 57% decrease (14.64 ± 2.63 in trial 1, 6.20 ± 3.71 in trial 5). Thus, irrespective of which odor is used for which CS, there is a significant decrease in CS+ odor response and no significant change in CS- odor response during acquisition.

We have added the following text to describe these findings more clearly (Page 8):

Note that although the magnitudes of PAM- β '2a of naïve odor responses to OCT and MCH are different, using either odor as CS+ led to a decrease in CS+ odor response during acquisition of more than 50% and an increase during reversal, with no significant changes in CS- odor response during acquisition.

17) The authors should consider to discuss the feed-forward inhibition from MBON γ 1ped to the horizontal tip MBONS including MBON γ 5 β '2a in more depth.

This is an important topic to discuss, given the known role of γ 1ped microcircuit in forming and storing the original aversive memory. We now mention this twice:

This PAM- β '2a-mediated increase in MBON- γ 5 β '2a CS+ odor response is in addition to the known increase in MBON- γ 5 β '2a CS+ odor response caused by disinhibition of MBON- γ 1ped (Hige et al., 2015; Perisse et al., 2016; Felsenberg et al., 2018). (Page 19)

...

MBON- γ 5 β '2a is poised to integrate the initial aversive memory with the new no-shock memory via the direct feed-forward inhibitory connection from MBON- γ 1ped (Felsenberg et al., 2018). (Page 22)

18) The authors should use the prime symbol rather than the apostrophe when describing the respective neurons.

We have edited all text and figures accordingly.

Reviewer #2:

In this manuscript, McCurdy et al provide a circuit framework through which dopamine neurons encode the omission of aversive outcomes that drive reversal of cue-outcome contingencies in Drosophila. The authors present an excellent set of in vivo functional imaging, optogenetic manipulation and behavioral analysis along with synaptic reconstructions from electron microscopy data. They demonstrate that PAM-β'2a dopaminergic neurons (DANs) encode shock withdrawal during reversal as potentially rewarding. This dopaminergic reward signal plays a role in the extinction of shock-paired conditioned stimulus (CS+) shock association thus reducing CS+ avoidance by depressing KC synapses onto avoidance-encoding MBON-γ5β'2a. Further, they show approach-encoding MBON-γ2α'1 acts as an upstream modulator that activates PAMβ'2a when shock is withdrawn and PPL1-γ2α'1 shock-responsive DANs relieve synaptic depression of KC-MBON-γ2α'1 synapses to increase CS+ odor activation. Together, these findings provide a mechanistic circuit explanation through which dopamine neurons provide an indirect relay between compartments in a memory structure to encode withdrawal of a punishment outcome to an appetitive outcome during reversal learning. This is an important contribution to our understanding of the fundamental mechanisms underlying memory formation.

I'd like to commend the authors on putting together such a cohesive and interesting paper. The writing is easy to follow, and data analyzed and presented in a rigorous and aesthetically clear way. Appropriate controls are present and provided in the supplemental data. Furthermore, the authors eloquently discuss the current literature, clearly outline the new information their data adds to the field, and thoughtfully explain their data using comparisons to circuit motifs in other animals, but without over-reaching. I really enjoyed reading the paper and thank the authors for their beautiful work. Recommendations listed below are minor concerns and intended only as suggestions to improve the rigor and readability of this important piece of work.

We are grateful to this reviewer for their enthusiastic comments and suggestions for improvement, all of which have been incorporated in the revised manuscript.

1) Although extinction has historically been presented as unlearning of an association, the authors may want to consider updating their definition of extinction in the introduction. Recent data suggests that extinction is not the unlearning of an association, but occurs as a parallel memory trace. I recommend redefining in a way that is most consistent with the evidence.

This is a very important point, and we have edited our description of extinction accordingly:

In Drosophila, as in mammals, reward-encoding DANs are thought to underlie the extinction of aversive memories via the formation of a parallel association between the CS+ cue and the omission of punishment (Felsenberg et al., 2018; Otto et al., 2020).
(Page 5)

...

Taken together, our findings support the model of parallel opposing memories, first demonstrated in Drosophila extinction learning (Felsenberg et al., 2018), as the basis of reversal learning. (Page 22)

2) In Figure 1B, the training protocol depicted is not accompanied by data demonstrating that memory is formed with this protocol. Could the authors cite other work that shows this protocol produces memory or provide this behavioral data?

We agree that it is critical to provide evidence that the training protocol for the imaging experiments indeed leads to memory formation as reflected behaviorally. To validate behavioral relevance of our imaging results, we used two approaches.

First, we translated the imaging setup to the behavioral setup: 5s odor pulses with 55s air in between, for 5 trials, with 2 reversal trials after. The only difference being the shock duration. (This difference is necessary because in the imaging setup, the fly is tethered to the electric shock and cannot escape, whereas in the behavioral setup, flies can and do jump to temporarily avoid the electric shock.) Nonetheless, flies are able to acquire and then reverse the associations. We have included this data in **Fig. 1c**.

Additionally, we performed a hybrid experiment in which we conditioned each fly in the imaging setup (without cuticle dissection or head-fixing), and after 5 acquisition trials, placed the fly in the quadrant arena. We observed difference in MCH odor preference, as quantified by percentage of time spent in MCH versus OCT quadrants, depending on which odor the fly was conditioned to associate electric shock with. This data is included in **Supplemental Fig. 1a**.

These additional experiments demonstrate that the protocol used in the imaging experiments leads to behaviorally relevant changes in neural activity, and that the changes in neural activity observed are responsible for the changes in behavior observed in the behavioral experiments.

3) In order to ensure other people could reproduce this work, please include more detail on how ‘difference in mean odor response $\Delta R/R$ ’ (as in Fig 1C) is calculated in the methods.

We have updated our Methods section to provide more details on how $\Delta R/R$ is calculated:

Mean odor response was defined as the mean $\Delta R/R_0$ during the first four seconds after odor onset ($t = 0-4s$).

...

In screen data, “difference in mean odor response” for acquisition is calculated as the subtraction of mean odor response on the 1st acquisition trial from mean odor response on the 5th acquisition trial. “Difference in mean odor response” for reversal is calculated as the subtraction of mean odor response on the 1st reversal trial from the mean odor response on the 2nd reversal trial. In both cases, a positive value indicates an increase in odor response relative to the first acquisition/reversal trial. In data for individual neurons, “odor response relative to trial #1” is the difference in mean odor response

relative to the mean odor response on the first acquisition trial. Positive values indicate an increase in odor response.

4) Please provide justification in the methods or results for why the behavior protocols for the optogenetics experiments differ with respect to timing of the stimulus, inter-trial stimulus, and number of repetitions/trials from the Ca²⁺ imaging experiments.

The differences in training paradigm between imaging and behavioral experiments are in part due to the differences in setup. For example, the stimulus durations of both the odor and electric shock are shorter in the imaging experiments versus the behavioral experiments (5s and 100ms versus 20s and 1.5s). This is because in the former, the fly is head-fixed and receives both the odor and electric shock in the exact same place, since it cannot move around. In the behavioral experiments, flies are trained *en masse* and they can jump in the shock tube to briefly escape the electric shock.

However, we provide additional behavioral experiments (mentioned earlier in point #2), and hope that these additional experiments provide sufficient evidence that even though there are differences between the imaging and behavioral experiments, they both reliably induce the formation and reversal of memories, establishing that changes in neural activity we observe are responsible for the changes in behavior in the behavioral experiments.

We also discuss these differences explicitly in the Results section:

Although the experimental protocol used in the imaging and behavioral experiments differ slightly, our data demonstrate that both setups lead to behavioral change, specifically an avoidance of CS+ after acquisition, and a decrease in avoidance after reversal (Fig. 1c, Supplemental Fig. S1a, Fig. 2a-c).

5) According to FlyLight, R66C08 is not specific to just the MBON-y5B'2a neuron, which is highly relevant for the optogenetic experiments conducted in Figure 2D-F. Please list off-target expression as detailed in FlyLight in the table in the methods section. Similar information should be provided for R39A05, R25D01-LexA and VT1211-LexA, which all look like they express in behaviorally relevant areas outside of their respective MBON targets.

We agree that off-target neurons could potentially influence fly behavior. We have added a list of off-target expression in the Table 1, as suggested (Page 32).

6) The authors might be interested in connectivity present between MBON α '2 and MBON B'2mp as this lends support to their functional data in Figure 3 (Aso et al 2014 closeness of fluorescent images in registered brains, Scaplen et al 2020 trans-tango connectivity, Xu et al 2020 bioRxiv latest annotation of EM hemibrain shows synapses).

This is a potentially important neural connection. However, the changes in MBON- α '2 odor responses we observed in our screen using MCH as CS+ did not generalize to the context where the CS+ and CS- odors are switched. This intriguing connection thus requires future investigation to probe its role.

We have mentioned this explicitly in our revised Results section and included the suggested references (Page 13):

However, this finding did not generalize, as no changes in CS+ odor response were observed in the reciprocal odor experiment (Supplemental Fig. S4a). We thus excluded this MBON from further analyses, despite connectivity between MBON- α '2 and MBON- β '2a (Aso et al., 2014a; Otto et al., 2020; Scaplen et al., 2020; Xu et al., 2020).

7) Please include justification in the methods / results for not including data with optogenetic activation of MBON-B'2mp (R39A05) for Figure 4A,B 100% duty cycle and 100% intensity and for Figure 4E 50% intensity.

Regarding Figures 4A,B, we were unable to complete all conditions fully in those experiments prior to submission due to the pandemic. We have since performed those experiments, and have included the data (Fig. 5a-c). The data provide interesting information about the nonlinear effects of activation and inhibition of these neurons on behavior.

Regarding Figure 4E, we did not include the final datapoint for MBON- β '2mp because we observed that at 50% red light intensity, flies expressing Chrimson in MBON- β '2mp had motor defects, leading to a decreased ability to move away from the red light. The resulting preference index was about -0.10, ostensibly indicating only a mild aversion to the red light. Thus we did not include the data as we were concerned it would mislead readers. However, upon further consideration, we believe that it would be informative to provide this explanation, which we have now included in the figure legend (Page 44). This is especially relevant given the reviewer's comment about off-target expression (point #5).

Note that the data for MBON- β '2mp at the highest light intensity is not shown, because these flies exhibit locomotor deficits at this light intensity, so their red light preference may not accurately reflect valence associated with MBON activation.

8) For Figure 5: Have the authors looked at whether there are connections between the PPL1- γ 2a'1 neuron and PAM B'2a neurons? If there are synapses between these neurons, the imaging and behavioral effects the authors see could potentially be attributed to these connections rather than going through the MBONs as the authors propose. I recommend either demonstrating the connections between these DANs don't exist (or aren't extensive depending on thresholding in EM dataset) or acknowledging this possibility in the discussion.

This is an important alternate hypothesis to consider. We performed the suggested additional electron microscopy analysis and found that there are in fact no direct connections between PPL1- γ 2a'1 and PAM- β '2a in the hemibrain dataset (Supplemental Fig. S7b).

9) It would be helpful to see the p values for the not significant results if they are close to significance in the supplement (for example in Supplemental Figure 2B the right plot looks potentially interesting)

In accordance with this suggestion, we have compiled a list of every test statistic as a Supplemental Table. To answer the reviewer's question specifically, the paired t-test result is $p = 0.57$ for +590nm condition, and $p = 0.10$ for the -590nm condition.

10) Supplemental Figure 3A-C the far right plots are confusing because only one ns is included when both blue and red are ns. I recommend either not including * when data is not significant or including both blue and red ns when data are not significant.

We have updated all imaging figures according to this recommendation.

11) Supplemental Figure 5: it would be helpful if the plots were labelled acquisition and reversal.

Those plots are actually not of acquisition and reversal, they are of mean and difference of shock response, respectively. We have made the y-axes and labels of these plots larger to clarify this.

12) The authors might want to update bioRxiv citations in case papers were published.

We have updated all the relevant citations accordingly.

Reviewer #3:

This article by McCurdy et al. investigates the role of a neuronal circuit consisting of four neuron types for the acquisition and the reversal of an associative odour-electric shock memory in *Drosophila*. Combining Ca-imaging and optogenetic activation of neurons, the authors describe this circuit in high detail and complexity, and suggest a 4-step pathway how electric shock information (or the omission of it) can tune this system to eventually make the animal avoiding or approaching the trained odour. Although this is a highly interesting research that provides important insights into the learning circuits of *Drosophila*, I have several major and minor concerns that have to be addressed before publication.

We thank Reviewer 3 for their detailed suggestions. Alternate interpretations suggested have motivated us to think hard about our model, and perform additional experiments and analyses. We have performed all experiments proposed, and have refined the interpretation of our work as suggested. We have also edited our terminology and highlighted the differences between extinction and reversal learning.

Major:

1. In the introduction, the authors try to define and extinction learning and reversal learning, and in the discussion they clearly separate between neurons involved in either of those learning types. In the result section, however, they mix up these phenomena, and sometimes use "reversal", "extinction", "shock omission" or "shock withdrawal" for the same observation. Sometimes they write "extinction that happens during reversal". This makes it difficult to follow their arguments. I think one origin of this problem is that technically the authors perform a reversal learning paradigm, i.e. they swap the contingencies of odours and shock (1. phase A+/B, 2. phase A/B+), but they discuss and focus only on the effects of the "missing" shock for odour A, and thus on a phenomenon

that is very similar to extinction learning. A complete account of reversal learning should include also what happens to odour B. I strongly advise to clarify the used terminology, and to re-consider what is the topic the authors want to investigate here. Specifically, I recommend not to use “withdrawal” because of its connotations of “taking away something” – that’s not what the authors do, they rather omit an expected shock.

We have substantially edited the manuscript to clarify the distinction between reversal and extinction. Specifically, we avoid using the term “extinction” when describing our results, replacing it with “the decrease in CS+ odor avoidance during reversal learning” or “the formation of the association between CS+ and shock omission”. We have also replaced the term “withdrawal” with “omission”, including in the title.

2. The authors repeatedly activate or silence a given neuron only during the presentations of the previous CS+ in the reversal training phase. Based on the observed changes in odour avoidance they conclude specific functions of these neurons for memory reversal. I want to offer an alternative perspective: all the effects the authors see in these experiments can be explained by rewarding or punishing properties of the neurons under study that may not be limited to reversal learning at all:

PAM-β'2a: This neuron is even published as rewarding neuron, as the authors mention in the discussion. Nevertheless, the authors claim that this neuron is not generally rewarding based on an one-training-trial experiment (Fig. S1). In contrast, in the reversal learning experiment (Fig. 1), they use two training trials – this can make a big difference for memory scores (compare also Fig. S1D and S1F). Therefore, this control experiment is not sufficient to rule out a rewarding function of the neuron. Furthermore, Yamagata et al. 2016 (PLOS Biology) showed that silencing a DAN can have the opposite reinforcing function as compared to activating it. That means, in the experiment shown in Fig. 1J the authors may actually provide a virtual punishment by silencing PAM-β'2a, which would explain the result.

We conducted the additional control experiments as per the reviewer’s suggestion (**Fig. 2c,d**). As the reviewer predicted, we indeed found that manipulating PAM-β'2a activity in the presence of an odor can bidirectionally alter the valence associated with the odor: activating PAM-β'2a during an odor increases attraction to the odor, and silencing PAM-β'2a during the odor increases avoidance of the odor. This confirms the reviewer’s explanation that PAM-β'2a provides a reward-like teaching signal that can operate not only in the context of reversal learning.

These findings are consistent with the ability of PAM-β'2a activity to cause changes in valence associated with an odor, and consequently lead to behavioral changes in odor preference. It is true that these behavioral changes could also be observed by manipulating other PAM subsets (e.g. PAM-γ3, as the reviewer points out). This is why our imaging results in **Fig 1** are crucial for revealing which neurons actually change their odor response during reversal. We identified that PAM-β'2a is the *only* DAN that changes its odor response during reversal. Thus our behavioral data, when combined with the imaging data, supports our model of PAM-β'2a driving the changes in odor preference.

MBON-γ5β'2a and MBON-γ2α'1: König et al. 2019 (Biol Letters) demonstrated that also activating and silencing of MBONs can have reinforcing effects, with opposite signs for

activating or silencing the neuron. That means also the authors' experiments (Fig. 2D-F, Fig. 4A-C) could be explained by reinforcing functions of the MBONs they use.

We have now incorporated this additional citation into the text. Again, it is true that activating or silencing either MBON in the presence of a naïve odor can alter behavior, due to their reinforcing effects. However, our behavioral experiments, when combined with the imaging experiments, provide evidence for the likely particular mechanism through which behavioral change occurs during reversal learning.

As suggested, we have highlighted that others have found that manipulating MBON activity during learning can alter odor preference (Page 14):

These effects of optogenetic manipulation of MBON- $\gamma 2\alpha'1$ and MBON- $\beta'2mp$ are again consistent with roles for both of these neurons in encoding shock omission during CS+ presentation in reversal trials. These findings are also consistent with how optogenetically altering MBON activity during learning can have reinforcing effects (König et al., 2019).

PPL1- $\gamma 2\alpha'1$: here the authors' data suggest that it signals electric shock punishments: it's activated by shock, and animals behave as if there would have been a shock when the neuron was activated. However, the authors conclude that it is "uniquely necessary" for "encoding of withdrawal of shock". This specific role is not supported by any data – the most parsimonious explanation is that this neuron is active when there is shock, and not active when there is no shock; activating it simulates a shock punishment, and therefore no reversal is observed.

Our rationale for claiming that PPL1- $\gamma 2\alpha'1$ is "uniquely necessary" for encoding shock omission is based on the finding that PPL1- $\gamma 2\alpha'1$ is the only DAN that synapses onto MBON- $\gamma 2\alpha'1$ (Berry et al., 2018). This, combined with PPL1- $\gamma 2\alpha'1$'s known punishment-encoding properties, makes it the likely cause of the changes in CS+ odor response in MBON- $\gamma 2\alpha'1$ during acquisition and reversal.

We do not mean to imply that PPL1- $\gamma 2\alpha'1$ is *only* involved in signaling shock omission during reversal learning, as it is clearly involved in other processes, including the acquisition of aversive memories. We also do not mean to imply that *only* manipulating PPL1- $\gamma 2\alpha'1$ can result in the changes in neural activity and behavior observed in **Fig. 6**. Manipulating other punishment-encoding DANs, e.g. PPL1- $\gamma 1ped$, could have the same effect. But when considered in combination with our imaging studies and the extensive data supporting the other three nodes in our circuit (MBON- $\gamma 2\alpha'1$, PAM- $\beta'2a$ and PAM- $\gamma 5\beta'2a$), PPL1- $\gamma 2\alpha'1$ is the most parsimonious first node in our circuit.

In accordance with the reviewer suggestion, we have toned down our claim, and have removed the term 'uniquely' from the sentence.

IF the authors want to make the claim that the effects they see in these experiments are specific for memory reversal, they need to do proper control experiments by activating or silencing every neuron under study with the very same parameters as in their reversal experiment – just omitting the first training phase. Alternatively, the authors should

discuss the alternative explanations that I mentioned, and tone down their conclusions from the behavioural experiments.

We do not wish to claim that these neurons cause those effects *only* during reversal learning, or that *only* these neurons can cause the observed changes in neural activity or behavior. Our claim is that when combining the optogenetic experiments (which reveals which neurons are capable of causing changes in behavior) with the imaging data (which reveals which neurons *actually* change their activity during the behavior) that we derive the most parsimonious model of reversal learning.

Acknowledging the importance of clarifying that we do not believe that the neurons identified in this reversal learning circuit are *only* involved in reversal learning, we have edited the text to make this clear. For example:

Here we demonstrate a role for MBON- γ 2 α '1 and PPL1- γ 2 α '1 in encoding shock omission as reward during reversal learning, in addition to their known roles in aversive memory formation. Both MBON- γ 2 α '1 and PAM- β '2a respond to naïve odors, so it is likely that MBON- γ 2 α '1 contributes to the naïve odor response in PAM- β '2a, resulting in a particular valence associated with the odor. However, activation of MBON- γ 2 α '1 in response to CS+ odor during reversal learning causes a relative increase in PAM- β '2a odor response, which skews the valence associated with the CS+ odor to decrease CS+ odor avoidance. (Page 21)

And as mentioned earlier, we put the control experiments for PAM- β '2a in the main figure (**Fig. 2d,e**) to make clear that manipulating PAM- β '2a changes odor preference in contexts other than reversal learning.

3. In Fig. 5B-E, the authors activate the PPL1- γ 2 α '1, the first neuron in the chain, and image MBON- γ 5 β '2a, the last neuron in the chain. First, they want to show that without activating PPL1- γ 2 α '1 odour responses of MBON- γ 5 β '2a increase upon acquisition, and decrease after reversal (Fig. 5D). This result is based on a sample size of 3, and not statistical significant. Even if an analogous experiment has been done before (with another genotype!), this sample size is just too low to allow any meaningful conclusions. Then, they show that with activation of MBON- γ 5 β '2a, the reduction of odour response does not occur (Fig. 5E). Here, the decisive statistical test is missing – between “post-acq” and “post-rev”. The authors should add the respective test in both Fig. 5D and 5E. Furthermore, they should increase the sample size for the experiment of Fig. 5D, or find another, statistically valid way to make their point.

We have performed the additional experiments to increase the n , performed the appropriate statistical test, and included an additional experiment using the reciprocal odor in the supplement, all of which further support our original claims (**Supplemental Fig. S6c**).

Minor:

4. For the screen results (e.g. Fig. 1C-D), the black lines and individual data points are difficult to discriminate. Also, what is the meaning of the stippled vertical line in Fig. 1D?

We have decreased the optical transparency of the individual datapoints for improved visual clarity of the figure. The stippled vertical line was to separate the PAM neurons from the PPL1 neurons, but we agree this seems unnecessarily confusing; we have now removed it.

5. For many of the Ca-imaging experiments, the authors always use one given odour chemical as CS+, and the other one as CS-. Only for some of the experiments, they performed the reciprocal experiment as supplement. This is problematic as any differential effects for CS+ and CS- can be confounded by differential responses of the neuron to the particular chemical substance rather than to the experimental manipulation. This problem gets particularly obvious for PAM- β '2a (Fig. 1/S1) because this neuron responds drastically different to the two odours. The authors should clearly state and discuss this potential confounding effect.

We agree that reciprocal odor experiments are necessary for fullest interpretation of our imaging data. We have since conducted all additional imaging experiments to include the reciprocal odor data (**Supplemental Figs. S3a, S4a,b**).

6. I would suggest adding a display like in Fig. 5D-E to all Ca-Imaging results, i.e. the changes of odour responses after acquisition and reversal with a common reference. This would make clear that all the changes the authors observe during acquisition are reversed back to baseline after reversal. The way the figures 1-4 are designed at the moment obscures this observation by using a different reference for acquisition and reversal.

We suggested, we have replotted the reversal phase of all our imaging data to normalize to a common reference (odor response on acquisition trial 1).

Along the same lines, I suggest that the authors discuss this observation more clearly because it is actually a very interesting finding. Extinction is often regarded not as the “un-learning” of a previous memory, but as a new, additional memory that suppresses the first one. The observation that in the circuit under study the original memory trace seems to be lost during reversal is therefore very interesting – and seems to be in contrast to what happens in other circuits: MBON- γ 1ped, for example, does not restore the reduced CS+ response after reversal, but rather seems to keep the original memory trace (Fig. 3E).

This is an important point. Our results, especially that of MBON- γ 1ped, complement what was observed in Felsenberg et al., 2018, in which two parallel memories coexist in different circuits. This finding is more visually obvious thanks to the replotting suggestion in the earlier point. We have now included a more thorough discussion of this finding:

Taken together, our findings support the model of parallel opposing memories, first demonstrated in Drosophila extinction learning (Felsenberg et al., 2018), as the basis of reversal learning. In this model, the γ 1ped microcircuit encodes the originally acquired aversive memory (Hige et al., 2015; Perisse et al., 2016; Felsenberg et al., 2018). Indeed, we found that MBON- γ 1ped did not increase its CS+ odor response during reversal (Fig. 4e). This was true even after 6 reversal trials (data not shown), demonstrating the persistence of the original memory stored in the γ 1ped microcircuit. MBON- γ 5 β '2a on the other hand is responsible for forming the association between CS+ and shock

omission via PAM- γ 5(fb) in extinction learning, and is the last node in the circuit for forming the CS+-no shock memory via the γ 2 α '1- β '2 relay network we uncover here in reversal learning. MBON- γ 5 β '2a is poised to integrate the initial aversive memory with the new no-shock memory via the direct feed-forward inhibitory connection from MBON- γ 1ped (Felsenberg et al., 2018).

7. I feel that the experiment in Fig. S1D deserves to be emphasized much more. This is the single experiment in the whole study showing that the reversal learning paradigm the authors use is actually working as intended, and it is not even mentioned with a single sentence in the main text! The authors should at least clearly refer to this result, or may even consider moving it to Figure 1.

We agree that this is an important experiment for validating our interpretations. We have now moved this to a main figure (**Fig. 2a,b**), and have performed additional behavioral experiments to provide further evidence validity of our interpretations (**Fig. 1c, 2c**).

8. I have a hard time to follow the rationale and conclusions of Fig. 4A-E: First, I would ask the authors to explain the term “duty cycle” that is unknown to me and is also not used anywhere else in the study.

We have clarified what we mean by “duty cycle” in the Methods section (Page 28):

Unless otherwise stated, red and green light were set to 50% and 100% max intensity respectively; both were pulsed at 50Hz with a duty cycle of 50%, i.e. 10ms light on, 10ms light off. A duty cycle of 100% indicates that the lights were constitutively on.

Second, the authors conclude from Fig. 4B that activation of the two MBONs have opposite effects on odour avoidance. However, contemplating the lowest intensity, for both neurons a reduction of avoidance is observed (although not significant in one case); for the highest intensity, this effect seems to be reversed for one of the MBONs – yet this is solely due to a change in the control group, whereas the experimental group behaves the same.

We agree that these optogenetic behavioral experiments did not provide results that are easily interpretable. We discuss the possible reasons for this in the Results section (Page 14):

The complex relationship between degree of light exposure and behavioral effects in these optogenetic experiments could be due to (1) photobiophysics and membrane biophysics of Chrimson and GtACR1-expressed in MBON- γ 2 α '1 and MBON- β '2mp, (2) nonlinear network effects in the context of recurrent MB connectivity (Aso et al., 2014a), and/or (3) superimposed non-linear intrinsic effects of red and green light on behavior.

There are many potential reasons for why we observe one particular valence associated with activation at one particular light intensity/duty cycle relative to genetic controls, but the opposite valence using a different light intensity/duty cycle.

That said, we do not discount the effects of light intensity independent of Chrimson expression, i.e. the behavioral effects observed in the empty-splitGAL4 flies. These data indicate that higher

red light intensity/duty cycle is intrinsically aversive independent of activating optogenetic actuators, hence pairing it with CS+ odor increases CS+ odor avoidance. So although the experimental group (MBON- $\gamma 2\alpha'1$) does not exhibit much change in CS+ avoidance across red light intensities/duty cycles in absolute terms, the important comparison is between the experimental group and control group at each light intensity/duty cycle.

Also, the other MBON is not tested under these conditions, so we don't know which effects it would have here.

We have performed the additional behavioral experiments for MBON- $\beta'2mp$ to complete the dataset (**Fig. 5b,c**). We found that optogenetic activation of MBON- $\beta'2mp$ using the lowest light intensity and duty cycle impairs CS+ reversal, while optogenetic inhibition using the lowest two light intensities and duty cycles enhance CS+ reversal, consistent with MBON- $\beta'2mp$ driving avoidance-promoting behaviors.

Third, I understand that Fig. 4E demonstrates that light intensities can matter for behavior. But why do the authors use completely different intensity levels here (1-50% versus 50-100% in the previous experiment)? The authors should state more clearly why which parameter was chosen, and discuss the potential caveats of this experiment.

It is difficult to compare the red light intensities used in **Fig. 5b** and **Fig. 5e**, for a number of reasons. First, the LEDs used in **Fig. 5b** are 617nm, while the ones used in **Fig. 5e** are 627nm. Second, for experiments in **Fig. 5b**, the LED board is positioned directly under the shock tube, which is translucent, and it is unclear what final light intensity that reaches the flies in the tube after being diffused by the shock tube. For experiments in **Fig. 5e**, the LED is positioned under the quadrant arena, which has a 3mm thick diffuser with IR absorption film. It is thus difficult to compare the relative contribution of each material and their effects on final light intensity that reaches the flies.

We have updated the Results section and figure legends to clarify the experimental details:

*Although the light intensities used in these two experiments (**Fig. 5b** versus **Fig. 5e**) cannot be directly compared due to different materials used to diffuse the light, they provide evidence that opposing valences can be induced based on different light intensities. (Results, Page 14)*

...

*Note that the red light intensities used in **Fig. 5b** cannot be directly compared with those used in this experiment. (Figure legend, Page 43)*

9. In Fig. 4H, the authors activate two MBONs and measure the odour response of PAM- $\beta'2a$. With a low sample size and an almost statistically significant increase in odour response, they simply state that no effect was observed. I think this point deserves more discussion, given that one of the activated MBONs was actually supposed to decrease responses (the opposite of the observed trend), and the other neuron was supposed to be downstream of PAM- $\beta'2a$ and therefore should have no effect. How do the authors explain this result?

We appreciate the reviewer raising this question and encouraging us to think more about this finding. It is difficult to parse the relative contribution of each of the two MBONs (MBON- $\beta'2mp$ and MBON- $\gamma5\beta'2a$) on PAM- $\beta'2a$ activity, but we can speculate based on our and others' work.

One possible model is that MBON- $\beta'2mp$ and MBON- $\gamma5\beta'2a$ have an inhibitory and excitatory effect on PAM- $\beta'2a$, respectively. This would result in the small net change in activity observed in **Fig. 5h**. Likewise, others have found that activating these two MBONs leads to a delayed inhibition and excitatory rebound response in PAM- $\beta'2m$ and PAM- $\beta'2p$ (Felsenberg et al., 2018).

We used electron microscopy to determine connectivity between each MBON and PAM- $\beta'2a$ (**Supplemental Fig. S7a**). we identified 2 synapses between MBON- $\beta'2mp$ and PAM- $\beta'2a$, and 10 synapses between MBON- $\gamma5\beta'2a$ and PAM- $\beta'2a$. Otto et al. 2020 also identified relatively stronger connections between MBON- $\gamma5\beta'2a$ and PAM- $\beta'2a$ versus MBON- $\beta'2mp$ and PAM- $\beta'2a$. Although we cannot assume a linear relationship between number of synapses and strength of connectivity, we propose that the response we observed more likely came from MBON- $\gamma5\beta'2a$ activation than MBON- $\beta'2mp$ activation, making it less likely that the connection between MBON- $\beta'2mp$ and PAM- $\beta'2a$ is responsible for the increase in PAM- $\beta'2a$ odor response during reversal learning.

We have now included a discussion of these possibilities in the Discussion section (Page 22):

*Our model does not completely rule out a potential contribution from MBON- $\beta'2mp$ onto PAM- $\beta'2a$. Despite the fact that both MBON- $\beta'2mp$ and MBON- $\gamma5\beta'2a$ secrete glutamate (Aso et al., 2014a; Aso et al., 2014b), they could possibly cause inhibitory and excitatory responses in PAM- $\beta'2a$, respectively, leading to the non-significant increase in PAM- $\beta'2a$ activity observed in **Fig. 5h**, the former possibly via glutamate-gated chloride channels. However, electron microscopy analyses revealed relatively few connections between MBON- $\beta'2mp$ and PAM- $\beta'2a$, relative to the connections between MBON- $\gamma2\alpha'1$ and PAM- $\beta'2a$ (**Supplemental Fig. S7a**). We thus believe that the direct excitatory connection between MBON- $\gamma2\alpha'1$ and PAM- $\beta'2a$ is the most parsimonious model.*

10. To activate Chrimson, the authors use 3 different wavelengths in different experiments, and only two of them are mentioned in the methods section. Please explain.

We use 590nm for functional connectivity experiments, 617nm for behavioral optogenetic experiments in the shock tube, and 627nm for behavioral experiments in the quadrant arena. This is due to logistical reasons. For example, 590nm was necessary for functional connectivity experiments to be able to record GCaMP fluorescence while simultaneously activating Chrimson, given the filter cube installed. This information is now fully clarified in the Methods section (Pages 28,29).

11. Please give the exact sample sizes of each experimental condition, not just a range. This is particularly important when the sample sizes are strongly different (e.g. 4-13).

We now report in the figure legends the sample size for all behavioral experiments.

12. Page 4: please give a reference for the statement that *Drosophila* are capable of reversing cue-outcome associations.

We have edited the statement to include studies of reversal learning conducted in *Drosophila* (Page 3):

“*Drosophila* are capable of both forming and reversing cue-outcome associations (e.g., Wu et al., 2012; Ren et al., 2012; Mancini et al., 2019) , such as odor-shock associations...”

13. When discussing the results of the first screen, please be consistent in describing the observed effects as increased/decreased CS+ responses in comparison to CS- responses. I think always mentioning your reference (the CS- response) is important. Please check this also throughout the text.

We have edited our descriptions of observed effects accordingly. For example:

No changes in CS- odor response occur in any DANs during acquisition. (Page 8)

...

Out of 17 MBON subsets screened, MBON- γ 1ped and MBON- γ 5 β '2a showed a significant difference in changes in CS+ versus CS- odor response during acquisition (Fig. 4a), consistent with prior observations (Bouzaiane et al., 2015; Hige et al., 2015; Oswald et al., 2015; Perisse et al., 2016). MBON- α '2, MBON- β '2mp, MBON- γ 1ped, MBON- γ 2 α '1 and MBON- γ 5 β '2a changed CS+ odor response relative to CS- during reversal (Fig. 4b). (Page 13)

...

No changes in CS- odor response occur during acquisition or reversal, except for MBON- γ 1ped... (Page 13)

14. Page 12: The authors refer to previously reported rewarding or punishing valences of MBONs by Aso et al. 2014. This is very misleading. These MBONs receive input from reward- or punishment-signalling DANs, and they are approach- or avoidance-promoting. Whether they may have rewarding or punishing properties themselves, was not studied by Aso et al.

We have clarified this according to the suggestion on Page 14:

These effects during reversal parallel previously reported approach- and avoidance-promoting behaviors driven by MBON- γ 2 α '1 and MBON- β '2mp, respectively (Aso et al., 2014b).

15. Page 24: The measurement of CS+ avoidance is described as the fraction of flies in the CS- quadrants. Mathematically, this would be: (number of flies at CS-)/(total number of flies). However, this is not the formula the authors use.

We have replaced the term “fraction” with “proportion” to avoid confusion. We also explicitly provide the formula in the Methods section (Page 27):

CS+ avoidance was quantified as the proportion of flies in the CS- quadrants after being allowed to explore for 2min. Specifically, it was calculated as $[(\#flies\ in\ CS-\ quadrants) - (\#flies\ in\ CS+\ quadrants)] / (total\ \#flies)$; an avoidance index of 1 indicates that 100% of the flies chose CS- quadrants, and an avoidance index of 0 indicates that flies chose the two odors equally.

16. Page 25: I do not understand the sentence ending with “with no illuminated quadrants 30s after”.

We have edited that statement in Methods and figure legend for clarity:

For 30s, two diagonal quadrants were illuminated with 627nm red LEDs to activate Chrimson-containing neurons, after which, no quadrants were illuminated for 30s. This was then repeated with the other two quadrants. (Page 28)

...

After one minute of acclimation, two quadrants are illuminated with red light for 30 seconds to activate Chrimson-expressing neurons. After 30 seconds of darkness, the other two quadrants are illuminated. (Page 43)

17. Also Page 25: Do I understand the sentence beginning with “This was repeated, often with the same group of flies ...” correct that some groups of flies were used several times within one experiment, whereas others were used only once? This may be a critical confound of the data. Please specify precisely how you used flies repeatedly, and for which experimental conditions.

For the majority (90/108 = 83%) of data points in **Fig. 5e**, each group of flies was assayed at multiple intensities of red light for the innate quadrant preference assay. For the minority of data points, flies were only assayed at one light intensity. This is why there isn't an equal number of data points for each genotype across intensities. However, we combined the data because no differences in results were observed from flies that were tested at one light intensity versus flies that were tested at multiple light intensities.

Only data from empty-splitGAL4 and MBON- β '2mp at 1% and 20% intensity are a combination of single-test and multiple-test experiments. For example, in the empty-splitGAL4 condition, at 20% intensity, six data points are from multiple-test experiments, in which the group of flies was tested at 1%, 5% and 10% intensity prior to the 20% intensity experiment. This yielded a mean and sem of 0.21 ± 0.05 . Six data points are from single-test experiments, in which flies were only tested at 20% intensity. This yielded a mean and sem of 0.23 ± 0.08 . Likewise, for MBON- β '2mp at 20% intensity, seven multiple-test experiments and four single-test experiments yielded a mean and sem of -0.49 ± 0.09 and -0.54 ± 0.06 , respectively. And the data at 1% intensity are effectively identical whether they are single-test or multiple-test, because the experiment begins with the 1% intensity condition.

We now clarify this in the manuscript (Page 29):

This was then repeated with the same group of flies over a range of light intensities: 1%, 5%, 10%, 20% & 50%. For a minority of data points, flies were only tested at one light intensity. Since there were no differences in mean preference between these two datasets, they were combined.

18. Page 26: The reference to Fig. 5H,J is wrong.

This reference has been corrected.

Reviewer #1 (Remarks to the Author):

McCurdy et al improved the manuscript substantially by adding important and interesting new data. The work reveals new insights into circuit changes due to reversal learning which will have an impact beyond the fly learning and memory field. However, based on the presented data the authors must adjust their claims and clearly state what remains unknown. The data is very important to be communicated and tuning down the claims will not lower the impact of the work if the unknown is clearly stated and discussed. Further, alternative explanations for some of their findings must be considered. Unfortunately, my previous comment on the flawed citation of the published work has been carelessly neglected and needs to be revised beyond the given examples I added here. Below I will list my remaining concerns and those that arise from the new data.

The data is very important, and I strongly support its publication given that the issue below are addressed

1) The authors acknowledge that there is no increase in PAM $\beta'2a$ above the initial odor response during reversal but rather a return to baseline. However, they write sentences like: "Because PAM DANs are traditionally considered to encode reward (Liu et al., 2012; Das et al., 2014; Lin et al., 2014; Lewis et al., 2015), PAM- $\beta'2a$'s increase in CS+ odor response during reversal suggests that it could encode the rewarding value of shock omission."(page 9). Thus, the biggest concern is that they fail to state clear enough that PAM- $\beta'2a$ responses are not enhanced during reversal but rather recovery to baseline. This must be clearly stated in the results every time the phenomenon is mentioned to avoid misunderstandings. The authors are free to outline their model in the discussion. However, they should acknowledge that this is an assumption and that currently there is no published evidence that a sub-baseline odor response in reward-coding DANs can be rewarding for flies. The idea that it is a relative increase that provides the teaching signal should carefully laid out, e.g. how does odor specificity work in such a model?

2) Related to 1), the authors argue that the optogenetic manipulation during reversal learning strongly support a role of the PAM- $\beta'2a$. However, their added experiments show that the mere pairing of inhibition or activation of these PAMs assigns a value so that behavior is changed. The finding that the very same changes in behavior are observed when the same manipulations are applied during the reversal learning experiments argues against a reversal specific effect of the manipulation. In fact, it is likely that manipulation of any reward coding DAN would lead to the same phenotype, especially the $\gamma3$ -DANs. I strongly advice to seriously re-write this part and explicitly mentions the caveats of these experiments in the results (same is true for MBON and PPL1 activation/inhibition experiments). Again, I agree with the authors that the PAM- $\beta'2a$ is the best guess to provide a teaching signal due to the imaging experiments and that the optogenetic experiments do not contradict with this idea but it should be clear that currently the mechanism how it works, given the baseline responses, is unclear.

3) Concerning the optogenetic experiments added in Fig 2 d-e the authors should comment on the effect the different light seems to have on the behavior of the control group. The presented data suggests that pairing the CS+ with a red light assigns a negative value and pairing it with green light assigns a positive valence to the odor.

4) The outcome of the aversive conditioning experiment in Sup Fig 2 is meaningless based on the data in Fig 2 and is mentioned in a misleading context. "Given that optogenetic inhibition of PAM- $\beta'2a$ fails to interfere with aversive shock-odor memory acquisition (Supplemental Fig. S2a), these imaging and functional optogenetic results indicate a specific role for PAM- $\beta'2a$ in encoding the relative reward of punishment omission during reversal." In the way the inhibition is applied during the experiment both odors become more negative. However, what drives avoidance in the test is

the difference arising from the pairing of the CS+ with the shock. This shock-driven difference is the same in experimental and control flies.

5) The subtitle “PAM- β '2a decreases CS+ odor avoidance during reversal by depressing KC-to-MBON- γ 5 β '2a synapses” is an assumption that would belong to the discussion. The authors show that the MBON- γ 5 β '2a increases during aversive conditioning, which to my knowledge, is widely considered to be a network effect arising from disinhibition based on learning induced depression of the GABAergic MBON- γ 1ped (Perisse et al 2016, Felsenberg et al 2018). However, the authors propose that the depression of the MBON- γ 5 β '2a derives from plasticity established by the below baseline activation of PAM β '2a. The data seems not to support this claim: PAM- β '2a shows, at least on average, reduced firing in the 4th and 5th learning trial (Fig 1 g). Thus, plasticity driven by such a decrease should be evident in MBON- γ 5 β '2a in trial 5 and 6. However, facilitation in MBON- γ 5 β '2a responses emerge as early as in trial 2 and plateaus in trial 4-6 suggesting very little or no additional increase. This dynamic fits with the plasticity of MBON- γ 1ped responses. Thus, the disinhibition seems to be more plausible to explain the changes in MBON- γ 5 β '2a during acquisition and should be mentioned accordingly. In contrast, the MBON- γ 5 β '2a decreases in reversal learning is exciting, since the MBON- γ 1ped remains unchanged. This finding is the strongest argument for reversal specific plasticity provided by PAM- β '2a. Obviously, demonstrating that above baseline activation of PAM- β '2a is in principle potent to induce depression (Fig 2 f and g) can be used as an additional argument. The optogenetic experiments manipulating the MBON- γ 5 β '2a activity during reversal learning have no additional value to support the claim nor are the important for the conclusion of the manuscript. Since they might have the same caveats as the PAM activation experiments and the respective controls are missing, the authors should consider removing the data from the main figure or from the manuscript entirely.

6) I noticed that the authors write “Out of 17 MBON subsets screened, MBON- γ 1ped and MBON- γ 5 β '2a showed a significant difference in changes in CS+ versus CS- odor response during acquisition.” (page 13), but in Figure 4 c,d and f show differences between the CS- and CS+ during acquisition for other MBONs. Further the above comment conflicts with their later conclusion that, “During acquisition, decreased MBON- γ 2 α '1 CS+ odor response decreases excitation of PAM- β '2a, reflected in its decreased CS+ odor response.” Only one of the conclusions can be true. Since the mismatch seem to originate in the different analysis I would urge the authors to display all their recordings in the style they use in Fig1g, 3b or 4c at least in the supplements. Such figures would be an important resource of information for the field.

7) The authors should adjust the misleading wording concerning increased responses to MBONs: “During reversal, shock omission increases MBON- γ 2 α '1 CS+ odor response, thus increasing PAM- β '2a CS+ odor response.” (page 16). The authors should change it accordingly consistent with 1)

8) The authors added important behavioral data validating their training and reversal protocol. The addition of the PAM- γ 5(fb) imaging experiments adds extra value to the work allowing the authors to emphasize the differences between extinction and reversal learning. Uncovering the distinction and the overlaps of the two processes makes the work even more valuable to understand reversal learning. With the shifted emphasize of the manuscript the authors should include reversal learning in the title and consider to change the sub-headers of the manuscript

9) “It thus remains unknown in any model organism or learning paradigm how DAN signals are generated in response to the omission of unconditioned stimuli, or how the omission of punishment is encoded as rewarding.” (page 5). This sentence needs an addition: “...of unconditioned stimuli during reversal learning.” Otherwise it is not true due the findings in extinction learning in flies, rats and mouse.

10) Regarding the claim: “We further demonstrate that approach-encoding MBON- γ 2 α '1 is an excitatory upstream element of PAM- β '2a which causes the changes in PAM- β '2a CS+ odor response

during acquisition and reversal.” (page 6). The authors will surely agree that their data suggests but not demonstrates.

11) To further clarify the data presentation in Figure 1 d and 1 e, the authors should clearly identify what 0 means. Further, I would encourage the authors to mention in the text that the CS+ responses in PAM $\beta'2mp$ an PPL1- $\gamma1ped$ remain decreased/increased during reversal training. It is an important and meaningful finding which the authors might want to discuss.

12) Due to the known aversive teaching signal of the PPL1- $\gamma2\alpha'1$ the same criticism as for the PAM activation applies here. It is very likely that the authors will observe the very same phenotype for the experiment in 6a when they use the PPL1- $\gamma1pedc$ and potentially as well with PPL1- $\alpha3$. Thus, according to published data there is no reversal learning specific PPL1- $\gamma2\alpha'1$ function implied by these experiments and the authors should mention that caveat. Activating this PPL1- $\gamma2\alpha'1$ neuron during the first CS+ reversal probably leads to the maintenance of the aversive memory trace in the respective MBON, which in turn seems to prevent the plasticity in MBON- $\gamma5\beta'2a$ during reversal. Given the functional connectivity the authors establish this finding is another very interesting argument for the circuit that links the maintenance of the initial memory to reversal learning. However, the authors seem to favor the explanation that the absence of the shock reduces PPL1- $\gamma2\alpha'1$ activity which leads to facilitation of the underlying MBON. This seems somewhat arbitrary since there is no change in the PPL1- $\gamma2\alpha'1$ in their imaging data. Work on active forgetting and reversal learning, Berry et al 2018, suggests that a shock alone but not the omission of a shock signal establishes such plasticity. The authors should consider this alternative explanation.

13) Maybe there is an error in the following sentence: “Likewise, unlike MBON- $\gamma5\beta'2a$ which changes its CS+ odor response both during extinction and reversal learning, we and others have uncovered some MBONs (e.g., MBON- $\gamma2\alpha'1$ and MBON- $\beta'2mp$) which change their CS+ odor response during extinction learning and not reversal learning (Berry et al., 2018; Felsenberg et al., 2018), providing additional evidence that distinct neural circuits underlie extinction and reversal learning ...” (page 21). I think it should say “...which changes their CS+ odor response during reversal learning but not extinction learning...”

14) In the discussion (p.19) the authors write “This decreases PAM- $\beta'2a$ CS+ response, thus relieving synaptic depression of KC-MBON- $\gamma5\beta'2a$ synapses and increasing MBON- $\gamma5\beta'2a$ CS+ odor response.” At this stage of the protocol, in untrained animals, these synapses are not depressed and therefore can not be relieved from it. In fact, as argued above, the change in this MBON might come from disinhibition through the MBON- $\gamma1pedc$.

15) “MBON- $\gamma2\alpha'1$ makes connections with a variety of DAN subsets, and is involved in appetitive and aversive short-term memory (Berry et al., 2018; Yamazaki et al., 2018) and consolidation of appetitive memories (Felsenberg et al., 2017).” The citations are not correct. To my understanding Berry and Yamazaki show no evidence for $\gamma2\alpha'1$ -MBON in appetitive memory and Felsenberg et al 2017 does not show its involvement in consolidation. In fact, Yamazaki explicitly claims it is not involved in appetitive short-term memory but Felsenberg shows some evidence that it might be involved in 3h mid-term memory retrieval and in reconsolidation. The authors clearly took my advice to double check the citation too lightly and only focused on the example I gave. In their own interest the authors should double check ALL literature they cite. I will point out some of the obvious mistakes.

16) Double check Lewis et al 2015 for reward PAM-DANs and add Yamagata et al 2015 and Burke et al 2012 on page 9 in the line starting “Because PAM DANs...”

17) Page 10 “...we focused our attention on these two MBONs, which encode negative and positive valence respectively (Aso et al., 2014b).” Oswald et al 2015 and Aso et al 2014b both show that these MBONs promote aversion and should be cited accordingly.

18) Page 11 “This is similar to the decrease in CS+ odor response during extinction learning observed

in previous studies (Bouzaiane et al., 2015; Oswald et al., 2015; Felsenberg et al., 2018).” Nor Bouzaiane et al neither Oswald et al do a single extinction experiment.

19) Page 13 “MBON- γ 1ped decreases CS- odor response during reversal, while its CS+ odor response is unchanged (Fig. 4e), also consistent with prior observations (Felsenberg et al., 2018)”. This is unprecise, since Felsenberg et al did not do reversal learning. Maybe the authors want to add “...in extinction...”

20) Page 13 “We thus excluded this MBON from further analyses, despite connectivity between MBON- α ’2 and MBON- β ’2a (Aso et al., 2014a; Otto et al., 2020; Scaplen et al., 2020; Xu et al., 2020).” Either content or citation is wrong in this sentence, maybe meant to say PAM- β ’2a.

21) Page 14 “These findings are also consistent with how optogenetically altering MBON activity during learning can have reinforcing effects (König et al., 2019).” The quote is not precise. There is no learning in König et al but the one driven by altering MBON activity. In addition, the authors want add Ichinose et al 2015 and Ueno et al 2017 for MBON driven teaching signals.

22) The authors should provide an additional figure for Fig 1 g and Sup Fig 1 d with a range that allows to judge about differences in OCT responses. In the current setting readers will assume something is mi labeled.

Reviewer #2 (Remarks to the Author):

The authors have sufficiently addressed all of my concerns. Moreover, they did a commendable job in addressing the concerns of the other reviewers. I was impressed by the amount of new data added to this manuscript and believe it is an important contribution to the literature.

Reviewer #3 (Remarks to the Author):

This revision of the manuscript by McCurdy et al addressed all the issues I had raised, and overall improved the clarity and quality of the work. The authors also included several new experiments which for sure is not an easy task in these difficult times.

Overall, I am satisfied with the manuscript in its current state. It reports very interesting novel data, clearly describes the used techniques, and draws appropriate conclusions. There are, however, still a couple of minor points that should be addressed before publication.

1. The authors do not use the term “extinction” with respect to their own results anymore, which is very good. However, when they do discuss extinction learning (in the introduction and discussion), it may not become fully clear now to all readers what extinction learning actually is. I suggest that the authors at one point give a brief definition/explanation of extinction learning, and what is the principal difference to reversal learning. Ideally, this is done when extinction is mentioned the first time in the introduction.

2. In the discussion, the authors compare their results to what is known about extinction learning (mostly referring to Felsenberg et al.), and concluding that there are both some overlap and some differences in the involvement of specific neurons. In general, I totally agree to this conclusion, but I would like to raise awareness to the fact that in the past parametrical differences between studies could cause differences in which specific neurons were found to be necessary for a given task. I suggest that the authors (very briefly) comment on the possibility that some of the differences found between their work and the literature may be due to such different parameters rather than extinction versus reversal learning.

3. When describing the results of the screen in Fig. 1 (p. 8), the authors state that “no changes in CS-odor response occur ...”. I wonder on which statistical basis this statement was done. As far I can see, the authors always compare CS+ and CS- responses – this test does not allow concluding which of the two responses was changed. To make such a statement, the authors should include a test comparing the CS- response to the baseline.
4. For all behavioral experiments, the authors now state the exact sample size as requested. For some of the calcium imaging experiments, however, for example Fig. 1d, there is still only a range of sample sizes given. Please correct.
5. In Fig. 2c, I suggest to replace “no shock” with “mock” in the x-axis label, as it is done for all the calcium imaging experiments.
6. In Fig. S1, please add “ns” to all the figures without significant changes, just as it is done in all the other figures.
7. On p 12, the sentence starting with “Finally, direct activation of ...” can be misleading. In all cases, the neurons are activated directly. I think, “acute activation”, as contrast to an activation during a previous training trial, would make the point clearer.
8. In the methods section (p. 27), the authors replaced “fraction of flies” with “proportion”. However, I still think this is not correct. At least as I understand it, “proportion of flies in the CS- quadrants” would calculate as: “number of flies in CS- quadrants / total number of flies”, and would have a scale from 0 to 1. However, the authors calculate: “number of flies in CS- quadrants – number of flies in CS+ quadrants / total number of flies”, with a scale from -1 to 1. I suggest to just omit the “proportion”-part.

REVIEWER COMMENTS

Reviewer #1 (Remarks to the Author):

McCurdy et al improved the manuscript substantially by adding important and interesting new data. The work reveals new insights into circuit changes due to reversal learning which will have an impact beyond the fly learning and memory field. However, based on the presented data the authors must adjust their claims and clearly state what remains unknown. The data is very important to be communicated and tuning down the claims will not lower the impact of the work if the unknown is clearly stated and discussed. Further, alternative explanations for some of their findings must be considered. Unfortunately, my previous comment on the flawed citation of the published work has been carelessly neglected and needs to be revised beyond the given examples I added here. Below I will list my remaining concerns and those that arise from the new data.

The data is very important, and I strongly support its publication given that the issues below are addressed.

We are grateful to Reviewer 1 for their meticulous re-examination of our manuscript. We take their comments on our work very seriously, and have incorporated them all.

1) The authors acknowledge that there is no increase in PAM $\beta'2a$ above the initial odor response during reversal but rather a return to baseline. However, they write sentences like: "Because PAM DANs are traditionally considered to encode reward (Liu et al., 2012; Das et al., 2014; Lin et al., 2014; Lewis et al., 2015), PAM- $\beta'2a$'s increase in CS+ odor response during reversal suggests that it could encode the rewarding value of shock omission."(page 9). Thus, the biggest concern is that they fail to state clear enough that PAM- $\beta'2a$ responses are not enhanced during reversal but rather recovery to baseline. This must be clearly stated in the results every time the phenomenon is mentioned to avoid misunderstandings.

We have edited our Results section to more clearly articulate this interpretation:

This increase in CS+ odor response during reversal does not completely return to or exceed the initial CS+ odor response, which is consistent with a weakening but not complete abolishing of negative valence associated with the CS+ odor after a single reversal trial. (Page 8)

We have also edited our text in other areas to make it explicit that when we refer to the increase in PAM- $\beta'2a$ CS+ odor response during reversal, we are referring to an increase relative to its CS+ odor response after acquisition, and not relative to its initial pre-acquisition CS+ odor response:

PAM- $\beta'2a$'s increase in CS+ odor response relative to post-acquisition during reversal suggests that it could encode the rewarding value of shock omission. (Page 10)

During acquisition, PAM- $\beta'2a$ decreases CS+ odor response. When electric shock is omitted during reversal, PAM- $\beta'2a$ increases CS+ odor response relative to post-

acquisition, thus depressing KC-MBON- γ 5 β '2a synapses, decreasing MBON- γ 5 β '2a CS+ odor response, and thereby decreasing CS+ avoidance. (Page 13)

During reversal, shock omission increases MBON- γ 2 α '1 CS+ odor response, thus increasing PAM- β '2a CS+ odor response relative to post-acquisition levels. (Page 17)

The authors are free to outline their model in the discussion. However, they should acknowledge that this is an assumption and that currently there is no published evidence that a sub-baseline odor response in reward-coding DANs can be rewarding for flies.

We believe that it is reasonable to conclude that there is a positive correlation between PAM- β '2a activity and reward signaling. Our optogenetic experiments (Fig. 2d,e) suggest a linear relationship between PAM- β '2a activity levels and reward signaling, where lower levels are more aversive and higher levels are more rewarding. It is thus reasonable to conclude that an increase in PAM- β '2a activation corresponds to an increase in reward signaling, irrespective of the original baseline for PAM- β '2a activity. Hence, a parsimonious explanation is that a relative increase in PAM- β '2a odor response during reversal, from post-acquisition CS+ levels to post-reversal CS+ levels, i.e., from low levels to less low levels, indicates a relative increase in reward signaling due to shock omission, that reduces avoidance of the odor.

To clarify that these are interpretations of our findings and not naked factual conclusions, we have refined our wording as follows:

Even though PAM- β '2a CS+ odor responses during reversal do not exceed its initial CS+ odor response, our optogenetic experiments suggest a linear relationship between PAM- β '2a activity levels and reward signaling. As such, this sub-baseline increase in CS+ odor response during reversal is likely nonetheless an increase in rewarding signaling thus decreases avoidance of the CS+ odor. (Page 11)

The idea that it is a relative increase that provides the teaching signal should carefully laid out, e.g. how does odor specificity work in such a model?

Odor specificity comes from the fact that the increase in PAM- β '2a odor response relative to post-acquisition levels occurs specifically in response to the CS+ and not CS-. This parallels the changes in MBON- γ 2 α '1, which also increases specifically its CS+ odor response during reversal, relative to post-acquisition levels, and not CS-.

To make this more clear, we have refined our description in the Discussion to:

During reversal, the omission of electric shock during CS+ odor presentation decreases PPL1- γ 2 α '1 activation due to the lack of shock response; this coupled with the presence of shock during CS- odor presentation relieves depression of KC-MBON- γ 2 α '1 synapses that are activated by the CS+ odor. Consequent increase in MBON- γ 2 α '1 CS+ odor response is relayed by an excitatory synapse to PAM- β '2a, increasing specifically its CS+ odor response. (Page 20)

2) Related to 1), the authors argue that the optogenetic manipulation during reversal learning strongly support a role of the PAM-β'2a. However, their added experiments show that the mere paring of inhibition or activation of these PAMs assigns a value so that behavior is changed. The finding that the very same changes in behavior are observed when the same manipulations are applied during the reversal learning experiments argues against a reversal specific effect of the manipulation.

We do not claim that the manipulation of PAM-β'2a activity *only* affects reversal learning. Indeed, as we show in that same figure, manipulating PAM-β'2a in the presence of a naïve odor affects odor-driven behavior. We mean that in the context of reversal learning, PAM-β'2a activity is not necessary for the formation of the original aversive memory (see point #4 for more details on this), but is necessary specifically during the reversal phase to drive the reduced CS+ avoidance. Nonetheless, we have removed the word 'specific' from our description of the results:

*Given that optogenetic inhibition of PAM-β'2a fails to interfere with aversive shock-odor memory acquisition (**Supplemental Fig. S2a**), these imaging and functional optogenetic results indicate a ~~specific~~ role for PAM-β'2a in encoding the relative reward of punishment omission during reversal. (Page 11)*

In fact, it is likely that manipulation of any reward coding DAN would lead to the same phenotype, especially the γ3-DANs. I strongly advice to seriously re-write this part and explicitly mention the caveats of these experiments in the results (same is true for MBON and PPL1 activation/inhibition experiments). Again, I agree with the authors that the PAM-β'2a is the best guess to provide a teaching signal due to the imaging experiments and that the optogenetic experiments do not contradict with this idea but it should be clear that currently the mechanism how it works, given the baseline responses, is unclear.

To refine this point in the manuscript, we have added the following sentences:

*Although manipulating PAM DANs other than PAM-β'2a could also lead to the changes in behavior observed in **Fig. 2d-g**, our imaging data demonstrate that PAM-β'2a is the only DAN that exhibits a relative change in CS+ odor response during reversal, making PAM-β'2a the likely teaching signal that drives reversal learning. (Page 10)*

*Although manipulating other MBONs could also lead to the changes in behavior observed in **Fig. 3d,e**, our imaging data suggest that MBON-γ5β'2a is the likely postsynaptic partner of PAM-β'2a which changes its CS+ odor response to cause a reduction in CS+ avoidance. (Page 12)*

*Although manipulating other MBONs could also lead to the changes in behavior observed in **Fig. 5b,c**, our imaging data suggest that these two MBONs are the likely MBONs involved in driving reversal learning, given their changes in CS+ odor response during reversal. (Page 15)*

*Although manipulating DANs other than PPL1-γ2α'1 could also lead to the changes in behavior observed in **Fig. 6a**, the extensive imaging and behavioral data presented thus*

far strongly suggest that PPL1- γ 2 α '1 is the most likely DAN responsible for driving reversal learning. (Page 18)

3) Concerning the optogenetic experiments added in Fig 2 d-e the authors should comment on the effect the different light seems to have on the behavior of the control group. The presented data suggests that pairing the CS+ with a red light assigns a negative value and pairing it with green light assigns a positive valence to the odor.

We have added a sentence in the Results section to clarify this:

Thus, despite the mildly negative and positive valence associated with red and green light, respectively, PAM- β '2a activity is rewarding, and the lack of activity is punishing. (Page 10)

4) The outcome of the aversive conditioning experiment in Sup Fig 2 is meaningless based on the data in Fig 2 and is mentioned in a misleading context. "Given that optogenetic inhibition of PAM- β '2a fails to interfere with aversive shock-odor memory acquisition (Supplemental Fig. S2a), these imaging and functional optogenetic results indicate a specific role for PAM- β '2a in encoding the relative reward of punishment omission during reversal." In the way the inhibition is applied during the experiment both odors become more negative. However, what drives avoidance in the test is the difference arising from the pairing of the CS+ with the shock. This shock-driven difference is the same in experimental and control flies.

We respectfully disagree that our data and interpretation of Supplementary Fig. S2a are meaningless and misleading. Silencing neurons throughout a learning task, not just specifically during an odor, is a common method for determining the necessity of the neuron's role in a particular learning task. For example, silencing PAM neurons throughout appetitive learning impairs appetitive memory formation (e.g., Liu et al., 2012; Vogt et al., 2014; Yamagata et al., 2016), and silencing PPL1 neurons throughout aversive learning impairs aversive memory formation (e.g., Schwaerzel et al., 2003; Aso et al., 2012; Vogt et al., 2014). Our finding that silencing PAM- β '2a throughout aversive learning does not impair aversive learning demonstrates that PAM- β '2a activity is not necessary during aversive learning, which we believe is accurately summarized by the text "optogenetic inhibition of PAM- β '2a fails to interfere with aversive shock-odor memory acquisition" (Page 11).

In contrast, the data in Fig. 2 demonstrate the sufficiency of activating and silencing PAM- β '2a in forming an appetitive and aversive memory respectively, for both naïve and learned odors. This however does not tell us whether PAM- β '2a is actually involved in aversive learning. Hence we believe that both kinds of experiments are critical in understanding the role of PAM- β '2a in aversive memory acquisition and reversal.

5) The subtitle "PAM- β '2a decreases CS+ odor avoidance during reversal by depressing KC-to-MBON- γ 5 β '2a synapses" is an assumption that would belong to the discussion.

We have refined this subtitle in the Results and figure legend:

PAM- β '2a likely decreases CS+ odor avoidance during reversal by depressing KC-to-MBON- γ 5 β '2a synapses during reversal learning (Page 11)

PAM- β '2a DANs likely induce depression of KC-MBON- γ 5 β '2a synapses to decrease CS+ odor avoidance during reversal (Page 42)

The authors show that the MBON- γ 5 β '2a increases during aversive conditioning, which to my knowledge, is widely considered to be a network effect arising from disinhibition based on learning induced depression of the GABAergic MBON- γ 1ped (Perisse et al 2016, Felsenberg et al 2018). However, the authors propose that the depression of the MBON- γ 5 β '2a derives from plasticity established by the below baseline activation of PAM β '2a. The data seems not to support this claim: PAM- β '2a shows, at least on average, reduced firing in the 4th and 5th learning trial (Fig 1 g). Thus, plasticity driven by such a decrease should be evident in MBON- γ 5 β '2a in trial 5 and 6. However, facilitation in MBON- γ 5 β '2a responses emerge as early as in trial 2 and plateaus in trial 4-6 suggesting very little or no additional increase. This dynamic fits with the plasticity of MBON- γ 1ped responses. Thus, the disinhibition seems to be more plausible to explain the changes in MBON- γ 5 β '2a during acquisition and should be mentioned accordingly.

We agree that the primary cause of change in MBON- γ 5 β '2a CS+ odor response during acquisition is driven by the reduced CS+ odor response on MBON- γ 1ped. We explain the role of γ 1ped in aversive memory acquisition multiple times in the manuscript, and have now edited the first quote to further emphasize that the γ 1ped microcircuit is the primary cause of change:

During aversive memory acquisition, coincident activation of specific KCs by the CS+ odor and PPL1- γ 1ped by electric shock depresses KC-MBON- γ 1ped synapses, decreasing MBON- γ 1ped CS+ odor response, and disinhibiting MBON- γ 5 β '2a (Hige et al., 2015; Perisse et al., 2016; Felsenberg et al., 2018). Additionally, coincident activation of specific KCs by the CS+ odor and PPL1- γ 2 α '1 by electric shock depresses KC-MBON- γ 2 α '1 synapses, decreasing MBON- γ 2 α '1 CS+ odor response (Berry et al., 2018). This decreases PAM- β '2a CS+ response, thus relieving synaptic depression of KC-MBON- γ 5 β '2a synapses, and increasing MBON- γ 5 β '2a CS+ odor response. (Page 20)

This is also completely in line with our finding (elaborated earlier in point #4) that PAM- β '2a is not necessary for aversive memory acquisition, as any decreases in synaptic plasticity induced on KC-MBON- γ 5 β '2a synapses that may increase MBON- γ 5 β '2a CS+ odor response are likely small in comparison to the disinhibition of MBON- γ 5 β '2a by MBON- γ 1ped.

In contrast, the MBON- γ 5 β '2a decreases in reversal learning is exciting, since the MBON- γ 1ped remains unchanged. This finding is the strongest argument for reversal specific plasticity provided by PAM- β '2a. Obviously, demonstrating that above baseline activation of PAM- β '2a is in principle potent to induce depression (Fig 2 f and g) can be used as an additional argument. The optogenetic experiments manipulating the MBON- γ 5 β '2a activity during reversal learning have no additional value to support the claim nor are the important for the conclusion of the manuscript. Since they might have the same caveats as the PAM activation experiments and the respective controls are missing, the authors

should consider removing the data from the main figure or from the manuscript entirely.

We are convinced that the optogenetic behavioral experiments performed on MBON- $\gamma 5\beta'2a$ have value as they serve as proof-of-principle that altering MBON- $\gamma 5\beta'2a$ during CS+ odor response can cause a decrease in CS+ odor avoidance. This is important to demonstrate, because if this were not the case, it would completely invalidate our model. Furthermore, these neural manipulations of MBON- $\gamma 5\beta'2a$ have not been assessed previously in the context of reversal learning.

We conclude based on our imaging data that MBON- $\gamma 5\beta'2a$ is the likely post-synaptic target of PAM- $\beta'2a$ that causes the change in behavior (due to its changes in CS+ odor response), and our optogenetic behavioral experiments performed on MBON- $\gamma 5\beta'2a$ support this conclusion, independent of potential elaborations on this conclusion, e.g., (1) whether manipulating MBON- $\gamma 5\beta'2a$ activity causes changes in odor preference in contexts other than reversal learning and (2) whether manipulating other neurons can cause the same change in odor preference. We consider these data of value and reasonable for the readers of our manuscript to draw their own conclusions.

6) I noticed that the authors write “Out of 17 MBON subsets screened, MBON- $\gamma 1ped$ and MBON- $\gamma 5\beta'2a$ showed a significant difference in changes in CS+ versus CS- odor response during acquisition.” (page 13), but in Figure 4 c,d and f show differences between the CS- and CS+ during acquisition for other MBONs. Further the above comment conflicts with their later conclusion that, “During acquisition, decreased MBON- $\gamma 2\alpha'1$ CS+ odor response decreases excitation of PAM- $\beta'2a$, reflected in its decreased CS+ odor response.” Only one of the conclusions can be true. Since the mismatch seem to originate in the different analysis I would urge the authors to display all their recordings in the style they use in Fig1g, 3b or 4c at least in the supplements. Such figures would be an important resource of information for the field.

Both conclusions are true; the reviewer is correct to attribute the different results to different statistical tests. We believe that both methods are valid, so we show and describe both findings: conclusions about differences in changes in CS+ versus CS- odor responses (t-tests for analyzing the imaging screen data e.g., Fig 4a) and changes in CS+ or CS- odor responses relative to the first acquisition trial (repeated-measures two-way ANOVA for analyzing responses within a genotype e.g., Fig. 4c).

Plotting the changes in odor response on a trial-by-trial basis for all genotypes tested in the imaging screen would yield 27 additional sub-figures (14 DANs and 13 MBONs), which could look quite overwhelming, even as a supplemental figure. However, we agree with the reviewer that these results will be an important resource for the field. As such, we have updated our supplementary statistics document to include F and p values from each repeated-measures two-way ANOVA analysis for each genotype featured in both imaging screens. This provides information on main effects of odor and acquisition trial number and their interaction.

We have also added figures to the supplement of the mean odor responses on the first acquisition trial and first reversal trial for all genotypes (Supplemental Fig. S1b,c, S5a,b), so the actual magnitude of odor responses is known, not just the difference. Thus for each genotype, we

provide information on: mean odor response on first acquisition trial (Supplemental Fig. S1b, Fig. S5a), mean shock response on first acquisition trial (Supplemental Fig. S8a), difference in odor response during acquisition (Fig. 1d, Fig. 4a), difference in shock response during acquisition (Supplemental Fig. S8b), mean odor response on first reversal trial (Supplemental Fig. S1c, Fig. S4b), difference in odor response during reversal (Fig. 1e, Fig. 4b); underline indicates the new figures added. This provides the reader with information on naïve odor responses, as well as changes in odor and shock responses during acquisition and reversal. Additionally, all our raw data remains available upon request.

7) The authors should adjust the misleading wording concerning increased responses to MBONs: “During reversal, shock omission increases MBON- γ 2 α '1 CS+ odor response, thus increasing PAM- β '2a CS+ odor response.” (page 16). The authors should change it accordingly consistent with 1)

We have edited the sentence accordingly:

During reversal, shock omission increases MBON- γ 2 α '1 CS+ odor response, thus increasing PAM- β '2a CS+ odor response relative to post-acquisition levels. (Page 17)

8) The authors added important behavioral data validating their training and reversal protocol. The addition of the PAM- γ 5(fb) imaging experiments adds extra value to the work allowing the authors to emphasize the differences between extinction and reversal learning. Uncovering the distinction and the overlaps of the two processes makes the work even more valuable to understand reversal learning. With the shifted emphasize of the manuscript the authors should include reversal learning in the title and consider to change the sub-headers of the manuscript.

We have now changed the title of the manuscript to: “*Dopaminergic mechanism underlying reward-encoding of punishment omission during reversal learning in Drosophila*”. Additionally, we have included the term “reversal” or “reversal learning” in most of our headings in the Results section and figure legends.

9) “It thus remains unknown in any model organism or learning paradigm how DAN signals are generated in response to the omission of unconditioned stimuli, or how the omission of punishment is encoded as rewarding.” (page 5). This sentence needs an addition: “...of unconditioned stimuli during reversal learning.” Otherwise it is not true due the findings in extinction learning in flies, rats and mouse.

We have edited this sentence accordingly:

It thus remains unknown in any model organism how DAN signals are generated in response to the omission of unconditioned stimuli during reversal learning, or how the omission of punishment is encoded as rewarding. (Page 5)

10) Regarding the claim: “We further demonstrate that approach-encoding MBON- γ 2 α '1 is an excitatory upstream element of PAM- β '2a which causes the changes in PAM- β '2a CS+ odor response during acquisition and reversal.” (page 6). The authors will surely

agree that their data suggests but not demonstrates.

We have edited this sentence accordingly:

We further demonstrate that approach-encoding MBON- γ 2 α '1 is an excitatory upstream element of PAM- β '2a which likely causes the changes in PAM- β '2a CS+ odor response during acquisition and reversal. (Page 6)

11) To further clarify the data presentation in Figure 1 d and 1 e, the authors should clearly identify what 0 means.

We have changed the y-axis label to “difference in mean odor response relative to trial #1” and “difference in mean odor response relative to trial #6” for Figures 1d and 1e, respectively. We did the same for the MBON screen data in Figure 4. We also included in the figure legend:

Figure 1d: Positive values indicate increase in odor response during acquisition; zero indicates no change in odor response during acquisition. (Page 38)

Figure 1e: Positive values indicate increase in odor response on second reversal trial relative to first reversal trial; zero indicates no change in odor response. (Page 39)

Further, I would encourage the authors to mention in the text that the CS+ responses in PAM β '2mp and PPL1- γ 1ped remain decreased/increased during reversal training. It is an important and meaningful finding which the authors might want to discuss.

We have added the following:

No other DANs change their odor responses during reversal; PAM- β '2m and PPL1- γ 1ped retain their decreased and increased CS+ odor response, respectively. (Page 8)

We also now discuss this in the Discussion:

We did not observe any reversal-related changes in CS+ odor response in PAM- β '2m, despite there being a decrease during acquisition. We hypothesize that PAM- β '2m may eventually change its CS+ odor response but only after additional reversal trials. PAM- β '2a would thus detect and signal unexpected reward as it occurs, and PAM- β '2m may eventually be recruited to signal reward as evidence accumulates. Given that PAM- β '2m receives dense innervation from MBON- γ 2 α '1 (Aso et al., 2014a), this may occur via a similar mechanism as the one proposed for PAM- β '2a. (Page 21)

12) Due to the known aversive teaching signal of the PPL1- γ 2 α '1 the same criticism as for the PAM activation applies here. It is very likely that the authors will observe the very same phenotype for the experiment in 6a when they use the PPL1- γ 1pedc and potentially as well with PPL1- α 3. Thus, according to published data there is no reversal learning specific PPL1- γ 2 α '1 function implied by these experiments and the authors should mention that caveat.

We have added the sentence:

Although manipulating DANs other than PPL1- γ 2 α '1 could also lead to the changes in behavior observed in Fig. 6a, the extensive imaging and behavioral data presented thus far strongly suggest that PPL1- γ 2 α '1 is the most likely DAN responsible for driving reversal learning. (Page 18)

Activating this PPL1- γ 2 α '1 neuron during the first CS+ reversal probably leads to the maintenance of the aversive memory trace in the respective MBON, which in turn seems to prevent the plasticity in MBON- γ 5 β '2a during reversal. Given the functional connectivity the authors establish this finding is another very interesting argument for the circuit that links the maintenance of the initial memory to reversal learning. However, the authors seem to favor the explanation that the absence of the shock reduces PPL1- γ 2 α '1 activity which leads to facilitation of the underlying MBON. This seems somewhat arbitrary since there is no change in the PPL1- γ 2 α '1 in their imaging data. Work on active forgetting and reversal learning, Berry et al 2018, suggests that a shock alone but not the omission of a shock signal establishes such plasticity. The authors should consider this alternative explanation.

We agree with the reviewer's interpretation of the results. As seen in Berry et al. 2018, the omission of a shock signal during CS+ is not sufficient for establishing plasticity, it must be combined with the presence of shock during CS-. We now explicitly mention this:

During reversal, the omission of electric shock during CS+ odor presentation decreases PPL1- γ 2 α '1 activation due to the lack of shock response; this coupled with the presence of shock during CS- odor presentation relieves depression of KC-MBON- γ 2 α '1 synapses. (Page 20)

13) Maybe there is an error in the following sentence: "Likewise, unlike MBON- γ 5 β '2a which changes its CS+ odor response both during extinction and reversal learning, we and others have uncovered some MBONs (e.g., MBON- γ 2 α '1 and MBON- β '2mp) which change their CS+ odor response during extinction learning and not reversal learning (Berry et al., 2018; Felsenberg et al., 2018), providing additional evidence that distinct neural circuits underlie extinction and reversal learning ..." (page 21). I think it should say "...which changes their CS+ odor response during reversal learning but not extinction learning..."

The reviewer is correct; we have edited the manuscript accordingly.

14) In the discussion (p.19) the authors write "This decreases PAM- β '2a CS+ response, thus relieving synaptic depression of KC-MBON- γ 5 β '2a synapses and increasing MBON- γ 5 β '2a CS+ odor response." At this stage of the protocol, in untrained animals, these synapses are not depressed and therefore cannot be relieved from it. In fact, as argued above, the change in this MBON might come from disinhibition through the MBON- γ 1pedc.

Indeed, as mentioned earlier and discussed in the manuscript, we agree that the increase in MBON- γ 5 β '2a CS+ odor response is driven primarily by MBON- γ 1ped disinhibition. However, given the strong naïve odor responses in PAM- β '2a, like in many other PAMs (now presented in Supplemental Fig. S1b), we conclude that PAM- β '2a induces some amount of synaptic

depression even in the absence of electric shock. Coincident activation of odor-activating KCs and PAM- β '2a odor response thus likely drives synaptic depression and contributes to the low MBON- γ 5 β '2a odor response, in addition to the inhibition by MBON- γ 1ped. Nonetheless, we have removed that portion from the Discussion and Figure legends:

Additionally, coincident activation of specific KCs by the CS+ odor and PPL1- γ 2 α '1 by electric shock depresses KC-MBON- γ 2 α '1 synapses, decreasing MBON- γ 2 α '1 CS+ odor response (Berry et al., 2018) and decreasing PAM- β '2a CS+ response. (Page 20)

During acquisition, CS+ odor response of PAM- β '2a (green) decreases; MBON- γ 5 β '2a CS+ odor response decreases via feedforward disinhibition by the γ 1ped compartment. (Page 44)

15) “MBON- γ 2 α '1 makes connections with a variety of DAN subsets, and is involved in appetitive and aversive short-term memory (Berry et al., 2018; Yamazaki et al., 2018) and consolidation of appetitive memories (Felsenberg et al., 2017).” The citations are not correct. To my understanding Berry and Yamazaki show no evidence for γ 2 α '1-MBON in appetitive memory and Felsenberg et al 2017 does not show its involvement in consolidation. In fact, Yamazaki explicitly claims it is not involved in appetitive short-term memory but Felsenberg shows some evidence that it might be involved in 3h mid-term memory retrieval and in reconsolidation. The authors clearly took my advice to double check the citation too lightly and only focused on the example I gave. In their own interest the authors should double check ALL literature they cite. I will point out some of the obvious mistakes.

We greatly appreciate the reviewer’s attention to ensuring fullest possible clarity of our citations to the prior literature.

Berry et al., 2018 provides evidence that MBON- γ 2 α '1 is involved in short-term aversive memory (Figures 1E,F). Yamazaki et al., 2018 does in fact provide evidence for MBON- γ 2 α '1 in appetitive memory: in Figures 5C-E, inhibiting MBON- γ 2 α '1 during acquisition, consolidation or retrieval impairs 2-hr appetitive memory, but not aversive memory.

To clarify all of this, we have now changed the sentence to:

MBON- γ 2 α '1 makes connections with a variety of DAN subsets, and is involved in appetitive and aversive memories (Berry et al., 2018; Yamazaki et al., 2018) and reconsolidation of appetitive memories (Felsenberg et al., 2017). (Page 23)

16) Double check Lewis et al 2015 for reward PAM-DANs and add Yamagata et al 2015 and Burke et al 2012 on page 9 in the line starting “Because PAM DANs...”.

Lewis et al., 2015 found that optogenetically activating PAM- β '2 causes attraction to the light, demonstrating that PAM- β '2 activity signals reward, so we believe this citation is appropriate. We have also included the additional citations suggested by the reviewer.

17) Page 10 “...we focused our attention on these two MBONs, which encode negative and positive valence respectively (Aso et al., 2014b).” Oswald et al 2015 and Aso et al

2014b both show that these MBONs promote aversion and should be cited accordingly.

We have added this citation accordingly.

18) Page 11 “This is similar to the decrease in CS+ odor response during extinction learning observed in previous studies (Bouzaiane et al., 2015; Oswald et al., 2015; Felsenberg et al., 2018).” Nor Bouzaiane et al neither Oswald et al do a single extinction experiment.

We have edited this citation accordingly.

19) Page 13 “MBON- γ 1ped decreases CS- odor response during reversal, while its CS+ odor response is unchanged (Fig. 4e), also consistent with prior observations (Felsenberg et al., 2018)”. This is unprecise, since Felsenberg et al did not do reversal learning. Maybe the authors want to add “...in extinction...”

We have edited this accordingly:

MBON- γ 1ped decreases CS- odor response during reversal, while its CS+ odor response is unchanged (Fig. 4e), the latter being consistent with prior findings from extinction learning (Felsenberg et al., 2018). (Page 14)

20) Page 13 “We thus excluded this MBON from further analyses, despite connectivity between MBON- α '2 and MBON- β '2a (Aso et al., 2014a; Otto et al., 2020; Scaplen et al., 2020; Xu et al., 2020).” Either content or citation is wrong in this sentence, maybe meant to say PAM- β '2a.

We have corrected this typographical error.

21) Page 14 “These findings are also consistent with how optogenetically altering MBON activity during learning can have reinforcing effects (König et al., 2019).” The quote is not precise. There is no learning in König et al but the one driven by altering MBON activity. In addition, the authors want add Ichinose et al 2015 and Ueno et al 2017 for MBON driven teaching signals.

We believe that the reviewer is referring to Ueoka et al., 2017, not Ueno et al., 2017, and have correspondingly refined our discussion of this point to add the suggested additional citations:

These findings are also consistent with how altering MBON activity in the presence of an odor can have reinforcing effects that change corresponding behavior towards the odor (Ichinose et al., 2015; Ueoka et al., 2017; König et al., 2019). (Page 15)

22) The authors should provide an additional figure for Fig 1 g and Sup Fig 1 d with a range that allows to judge about differences in OCT responses. In the current setting readers will assume something is mislabeled.

We have now included a figure (Supplemental Fig. S1f) that plots the difference in odor responses during acquisition for all 4 conditions (the 2 acquisition experiments and 2 mock acquisition experiments). The graphs show a significant difference in MCH odor response when

MCH is used as CS+ versus mock conditioning, and a significant difference in OCT odor response when OCT is used as CS+ versus mock conditioning.

Reviewer #2 (Remarks to the Author):

The authors have sufficiently addressed all of my concerns. Moreover, they did a commendable job in addressing the concerns of the other reviewers. I was impressed by the amount of new data added to this manuscript and believe it is an important contribution to the literature.

We thank Reviewer 2 for their enthusiastic support for this manuscript.

Reviewer #3 (Remarks to the Author):

This revision of the manuscript by McCurdy at al addressed all the issues I had raised, and overall improved the clarity and quality of the work. The authors also included several new experiments which for sure is not an easy task in these difficult times. Overall, I am satisfied with the manuscript in its current state. It reports very interesting novel data, clearly describes the used techniques, and draws appropriate conclusions. There are, however, still a couple of minor points that should be addressed before publication.

We thank Reviewer 3 for their support, and for their past and present suggestions for improvement. We have modified our manuscript to incorporate all additional points raised here.

1. The authors do not use the term “extinction” with respect to their own results anymore, which is very good. However, when they do discuss extinction learning (in the introduction and discussion), it may not become fully clear now to all readers what extinction learning actually is. I suggest that the authors at one point give a brief definition/explanation of extinction learning, and what is the principal difference to reversal learning. Ideally, this is done when extinction is mentioned the first time in the introduction.

We now include a description of extinction learning when we first mention it in the Introduction, as suggested. (Added portion underlined.)

*In *Drosophila*, as in mammals, reward-encoding DANs are thought to underlie the extinction of aversive memories via the formation of a parallel association between the CS+ cue and the omission of punishment (Felsenberg et al., 2018; Otto et al., 2020). However, those studies are of extinction learning, which is identical to reversal learning except with no pairing of CS- with punishment; it is less clear whether a similar process occurs during reversal learning. (Page 5)*

2. In the discussion, the authors compare their results to what is known about extinction learning (mostly referring to Felsenberg et al.), and concluding that there are both some overlap and some differences in the involvement of specific neurons. In general, I totally agree to this conclusion, but I would like to raise awareness to the fact that in the past parametrical differences between studies could cause differences in which specific neurons were found to be necessary for a given task. I suggest that the authors (very

briefly) comment on the possibility that some of the differences found between their work and the literature may be due to such different parameters rather than extinction versus reversal learning.

We have now included a discussion of the methodological differences between Felsenberg et al. 2018 paper and ours:

*It is possible that distinct circuits are involved due to methodological differences between extinction learning performed by others and reversal learning presented here. For example, flies undergo extinction and imaging 30 and 60 minutes after acquisition, respectively, whereas in our experiments reversal occurs immediately after acquisition. Interestingly, using our experimental setup, we did not observe any changes in odor response in PAM- γ 5(fb) neurons during acquisition or reversal (**Supplemental Fig. S2a-c**), suggesting different PAM subsets are involved due to different kinds of learning.*
(Page 22)

3. When describing the results of the screen in Fig. 1 (p. 8), the authors state that “no changes in CS- odor response occur ...”. I wonder on which statistical basis this statement was done. As far I can see, the authors always compare CS+ and CS- responses – this test does not allow concluding which of the two responses was changed. To make such a statement, the authors should include a test comparing the CS- response to the baseline.

It is true that the statistical tests shown in the imaging screen figures (e.g., Fig. 1d) are comparisons between changes in CS+ versus CS- odor response, and we agree that this is not appropriate for drawing conclusions about changes in odor response relative to each odor's baseline response. We have drawn this conclusion using repeated-measures two-way ANOVAs (using odor and trial number as variables as in e.g., Fig. 1g) for each genotype, comparing CS+ odor response on subsequent acquisition trials versus CS+ odor response on acquisition trial 1, and comparing CS- odor response on subsequent acquisition trials versus CS- odor response on acquisition trial 1, as the reviewer proposed. We have now included the results of these repeated-measures two-way ANOVAs for every genotype in our supplementary table of statistics.

4. For all behavioral experiments, the authors now state the exact sample size as requested. For some of the calcium imaging experiments, however, for example Fig. 1d, there is still only a range of sample sizes given. Please correct.

We included the exact sample sizes in the supplementary file where all statistical tests are reported. Additionally, we now report the exact sample size for both imaging screens in the Figure legends (Pages 39 and 45).

5. In Fig. 2c, I suggest to replace “no shock” with “mock” in the x-axis label, as it is done for all the calcium imaging experiments.

We have changed the label accordingly.

6. In Fig. S1, please add “ns” to all the figures without significant changes, just as it is done in all the other figures.

We have now added the appropriate statistics to all imaging figures.

7. On p 12, the sentence starting with “Finally, direct activation of ...” can be misleading. In all cases, the neurons are activated directly. I think, “acute activation”, as contrast to an activation during a previous training trial, would make the point clearer.

We have changed “direct” to “acute” accordingly.

8. In the methods section (p. 27), the authors replaced “fraction of flies” with “proportion”. However, I still think this is not correct. At least as I understand it, “proportion of flies in the CS- quadrants” would calculate as: “number of flies in CS- quadrants / total number of flies”, and would have a scale from 0 to 1. However, the authors calculate: “number of flies in CS- quadrants – number of flies in CS+ quadrants / total number of flies”, with a scale from -1 to 1. I suggest to just omit the “proportion”-part.

We have changed the wording accordingly:

CS+ avoidance was quantified as the number of flies in CS- quadrants minus the number of flies in CS+ quadrants, divided by total number of flies, after being allowed to explore for 2min; an avoidance index of 1 indicates that 100% of the flies chose CS- quadrants, and an avoidance index of 0 indicates that flies chose the two odors equally. (Page 29)

We have also changed the description of the red light preference experiments in line with this suggestion:

Red light preference index was calculated as the number of flies in red light quadrants minus flies in dark quadrants, divided by total number of flies; the values are then averaged to control for place preference. A preference index of 1 indicates that 100% of flies prefer red light quadrants, and a preference index of 0 indicates that flies chose the quadrants equally. (Page 31)

Reviewer #1 (Remarks to the Author):

I would like to thank the authors for carefully addressing my concerns by editing the text, clarifying concepts or correctly challenging my criticism. Further, I would like to apologize for my choice of words: none of the experiments are meaningless, on the contrary, all are valuable. After clarification provided by the authors, I see why they want to include the data. My reservation with regards to the interpretation of the activation and inhibition experiments remains but the authors addressed parts of it and I agree with their main conclusion. I would emphasize once more that the work presented is very important and will surely get the deserved attention. I recommend the manuscript for publications in its current form and I am looking forward to discussing the work on future occasions.

All the best

Johannes Felsenberg

Reviewer #3 (Remarks to the Author):

The authors have fully addressed all issues I raised. I support the acceptance of this manuscript without further revision.

REVIEWER COMMENTS

Reviewer #1 (Remarks to the Author):

I would like to thank the authors for carefully addressing my concerns by editing the text, clarifying concepts or correctly challenging my criticism. Further, I would like to apologize for my choice of words: none of the experiments are meaningless, on the contrary, all are valuable. After clarification provided by the authors, I see why they want to include the data. My reservation with regards to the interpretation of the activation and inhibition experiments remains but the authors addressed parts of it and I agree with their main conclusion. I would emphasize once more that the work presented is very important and will surely get the deserved attention. I recommend the manuscript for publications in its current form and I am looking forward to discussing the work on future occasions.

***All the best
Johannes Felsenberg***

We are deeply appreciative of Dr. Felsenberg for the time and effort he put into his reviews. His thorough examination of our manuscript and the resultant lively debates greatly strengthened our manuscript.

Reviewer #3 (Remarks to the Author):

The authors have fully adressed all issues I raised. I support the acceptance of this manuscript without further revision.

We thank Reviewer 3 for their helpful comments throughout this process.